# Model-free Posterior Sampling via Learning Rate Randomization

**Daniil Tiapkin**[1,2]    **Denis Belomestny**[3,2]    **Daniele Calandriello**[4]    **Éric Moulines**[1,5]
**Remi Munos**[4]    **Alexey Naumov**[2]    **Pierre Perrault**[6]    **Michal Valko**[4]    **Pierre Ménard**[7]
[1]CMAP, École Polytechnique    [2]HSE University    [3]Duisburg-Essen University
[4]Google DeepMind    [5]Mohamed Bin Zayed University of AI, UAE    [6]IDEMIA    [7]ENS Lyon
{daniil.tiapkin,eric.moulines}@polytechnique.edu
denis.belomestny@uni-due.de
{dcalandriello,munos,valkom}@google.com    anaumov@hse.ru
pierre.perrault@outlook.com    pierre.menard@ens-lyon.fr

## Abstract

In this paper, we introduce Randomized Q-learning (`RandQL`), a novel randomized model-free algorithm for regret minimization in episodic Markov Decision Processes (MDPs). To the best of our knowledge, `RandQL` is the first tractable model-free posterior sampling-based algorithm. We analyze the performance of `RandQL` in both tabular and non-tabular metric space settings. In tabular MDPs, `RandQL` achieves a regret bound of order $\widetilde{\mathcal{O}}(\sqrt{H^5SAT})$, where $H$ is the planning horizon, $S$ is the number of states, $A$ is the number of actions, and $T$ is the number of episodes. For a metric state-action space, `RandQL` enjoys a regret bound of order $\widetilde{\mathcal{O}}(H^{5/2}T^{(d_z+1)/(d_z+2)})$, where $d_z$ denotes the zooming dimension. Notably, `RandQL` achieves optimistic exploration *without using bonuses*, relying instead on a novel idea of *learning rate randomization*. Our empirical study shows that `RandQL` outperforms existing approaches on baseline exploration environments.

## 1   Introduction

In reinforcement learning (RL, Sutton and Barto 1998), an agent learns to interact with an unknown environment by acting, observing the next state, and receiving a reward. The agent's goal is to maximize the sum of the collected rewards. To achieve this, the agent can choose to use model-based or model-free algorithms. In model-based algorithms, the agent builds a model of the environment by inferring the reward function and the transition kernel that produces the next state. The agent then plans in this model to find the optimal policy. In contrast, model-free algorithms directly learn the optimal policy, which is the mapping of a state to an optimal action, or equivalently, the optimal Q-values, which are the mapping of a state-action pair to the expected return of an optimal policy starting by taking the given action at the given state.

Although empirical evidence suggests that model-based algorithms are more sample efficient than model-free algorithms [Deisenroth and Rasmussen, 2011, Schulman et al., 2015]; model-free approaches offer several advantages. These include smaller time and space complexity, the absence of a need to learn an explicit model, and often simpler algorithms. As a result, most of the recent breakthroughs in deep RL, such as those reported by Mnih et al. [2013], Schulman et al. [2015, 2017], Haarnoja et al. [2018], have been based on model-free algorithms, with a few notable exceptions, such as Schrittwieser et al. [2020], Hessel et al. [2021]. Many of these model-free algorithms [Mnih et al., 2013, Van Hasselt et al., 2016, Lillicrap et al., 2016] are rooted in the well-known Q-learning algorithm of Watkins and Dayan [1992]. Q-learning is an off-policy learning technique where the agent follows a behavioral policy while simultaneously incrementally learning the optimal Q-values

by combining asynchronous dynamic programming and stochastic approximation. Until recently, little was known about the sample complexity of Q-learning in the setting where the agent has no access to a simulator allowing to sample an arbitrary state-action pair. In this work, we consider such challenging setting where the environment is modelled by an episodic Markov Decision Process (MDP) of horizon $H$. After $T$ episodes, the performance of an agent is measured through regret which is the difference between the cumulative reward the agent could have obtained by acting optimally and what the agent really obtained during the interaction with the MDP.

This framework poses the famous exploration-exploitation dilemma where the agent must balance the need to try new state-action pairs to learn an optimal policy against exploiting the current observations to collect the rewards. One effective approach to resolving this dilemma is to adopt the principle of optimism in the face of uncertainty. In finite MDPs, this principle has been successfully implemented in the model-based algorithm using bonuses [Jaksch et al., 2010, Azar et al., 2017, Fruit et al., 2018, Dann et al., 2017, Zanette and Brunskill, 2019]. Specifically, the upper confidence bounds (UCBs) on the optimal Q-value are built by adding bonuses and then used for planning. Building on this approach, Jin et al. [2018] proposed the `OptQL` algorithm, which applies a similar bonus-based technique to Q-learning, achieving efficient exploration. Recently, Zhang et al. [2020] introduced a simple modification to `OptQL` that achieves optimal sample complexity, making it competitive with model-based algorithms.

Another class of methods for optimistic exploration is Bayesian-based approaches. An iconic example among this class is the posterior sampling for reinforcement learning (PSRL, Strens 2000, Osband et al. 2013) algorithm. This model-based algorithm maintains a *surrogate Bayesian model* of the MDP, for instance, a Dirichlet posterior on the transition probability distribution if the rewards are known. At each episode, a new MDP is sampled (i.e., a transition probability for each state-action pair) according to the posterior distribution of the Bayesian model. Then, the agent plans in this sampled MDP and uses the resulting policy to interact with the environment. Notably, an optimistic variant of PSRL, named optimistic posterior sampling for reinforcement learning (`OPSRL`, Agrawal and Jia, 2017, Tiapkin et al., 2022a) also enjoys an optimal sample complexity [Tiapkin et al., 2022a]. The random least square value iteration (`RLSVI`, Osband et al. [2013]) is another well-known model-based algorithm that leverages a Bayesian-based technique for exploration. Precisely, `RLSVI` directly sets a Gaussian prior on the optimal Q-values and then updates the associated posterior trough value iteration in a model [Osband et al., 2013, Russo, 2019]. A close variant of `RLSVI` proposed by Xiong et al. [2022], using a more sophisticated prior/posterior couple, is also proven to be near-optimal.

It is noteworthy that Bayesian-based exploration techniques have shown superior empirical performance compared to bonus-based exploration, at least in the tabular setting [Osband et al., 2013, Osband and Van Roy, 2017]. Furthermore, these techniques have also been successfully applied to the deep RL setting [Osband et al., 2016, Azizzadenesheli et al., 2018, Fortunato et al., 2018, Li et al., 2022, Sasso et al., 2023]. Finally, Bayesian methods allow for the incorporation of apriori information into exploration (e.g. by giving more weight to important states). However, most of the theoretical studies on Bayesian-based exploration have focused on model-based algorithms, raising the natural question of whether the PSRL approach can be extended to a provably efficient model-free algorithm that matches the good empirical performance of its model-based counterparts. Recently, Dann et al. [2021] proposed a model-free posterior sampling algorithm for structured MDPs, however, it is not computationally tractable. Therefore, a provably tractable model-free posterior sampling algorithm has remained a challenge.

In this paper, we aim to resolve this challenge. We propose the randomized Q-learning (`RandQL`) algorithm that achieves exploration without bonuses, relying instead on a novel idea of learning rate randomization. `RandQL` is a tractable model-free algorithm that updates an ensemble of Q-values via Q-learning with Beta distributed step-sizes. If tuned appropriately, the noise introduced by the random learning rates is similar to the one obtained by sampling from the posterior of the PSRL algorithm. Thus, one can see the ensemble of Q-values as posterior samples from the same induced posterior on the optimal Q-values as in PSRL. Then, `RandQL` chooses among these samples in the same optimistic fashion as `OPSRL`. We prove that for tabular MDPs, a staged version [Zhang et al., 2020] of `RandQL`, named `Staged-RandQL` enjoys the same regret bound as the `OptQL` algorithm, that is, $\widetilde{\mathcal{O}}(\sqrt{H^5 SAT})$ where $S$ is the number of states and $A$ the number of actions. Furthermore, we extend `Staged-RandQL` beyond the tabular setting into the `Net-Staged-RandQL` algorithm to deal with metric state-action spaces [Domingues et al., 2021c, Sinclair et al., 2019]. `Net-Staged-RandQL` operates similarly to `Staged-RandQL` but over a fixed discretization of the state-action space and

uses a specific prior tuning to handle the effect of discretization. We prove that `Net-Staged-RandQL` enjoys a regret bound of order $\widetilde{\mathcal{O}}(H^{5/2}T^{(d_c+1)/(d_c+2)})$, where $d_c$ denotes the covering dimension. This rate is of the same order as the one of `Adaptive-QL` by Sinclair et al. [2019, 2023], an adaptation of `OptQL` to metric state-action space and has a better dependence on the budget $T$ than one of the model-based kernel algorithms such that `Kernel-UCBVI` by Domingues et al. [2021c]. We also explain how to adapt `Net-Staged-RandQL` and its analysis to work with an adaptive discretization as by Sinclair et al. [2019, 2023]. Finally, we provide preliminary experiments to illustrate the good performance of `RandQL` against several baselines in finite and continuous environments.

We highlight our main contributions:

- The `RandQL` algorithm, a new tractable (provably efficient) model-free Q-learning adaptation of the `PSRL` algorithm that explores through randomization of the learning rates.
- A regret bound of order $\widetilde{\mathcal{O}}(\sqrt{H^5SAT})$ for a staged version of the `RandQL` algorithm in finite MDPs where $S$ is the number of states and $A$ the number of actions, $H$ the horizon and $T$ the budget.
- A regret bound of order $\widetilde{\mathcal{O}}(H^{5/2}T^{(d_c+1)/(d_c+2)})$ for an adaptation of `RandQL` to metric spaces where $d_c$ denotes the covering dimension.
- Adaptive version of metric space extension of `RandQL` algorithm that achieves a regret bound of order $\widetilde{\mathcal{O}}(H^{5/2}T^{(d_z+1)/(d_z+2)})$, where $d_z$ is a *zooming* dimension.
- Experiments in finite and continuous MDPs that show that `RandQL` is competitive with model-based and model-free baselines while keeping a low time-complexity.

## 2  Setting

We consider an episodic MDP $\big(\mathcal{S}, \mathcal{A}, H, \{p_h\}_{h\in[H]}, \{r_h\}_{h\in[H]}\big)$, where $\mathcal{S}$ is the set of states, $\mathcal{A}$ is the set of actions, $H$ is the number of steps in one episode, $p_h(s'|s,a)$ is the probability transition from state $s$ to state $s'$ upon taking action $a$ at step $h$, and $r_h(s,a) \in [0,1]$ is the bounded deterministic reward received after taking the action $a$ in state $s$ at step $h$. Note that we consider the general case of rewards and transition functions that are possibly non-stationary, i.e., that are allowed to depend on the decision step $h$ in the episode.

**Policy & value functions**   A *deterministic* policy $\pi$ is a collection of functions $\pi_h : \mathcal{S} \to \mathcal{A}$ for all $h \in [H]$, where every $\pi_h$ maps each state to a *single* action. The value functions of $\pi$, denoted by $V_h^\pi$, as well as the optimal value functions, denoted by $V_h^\star$ are given by the Bellman and the optimal Bellman equations,

$$
\begin{aligned}
Q_h^\pi(s,a) &= r_h(s,a) + p_h V_{h+1}^\pi(s,a) & V_h^\pi(s) &= \pi_h Q_h^\pi(s) \\
Q_h^\star(s,a) &= r_h(s,a) + p_h V_{h+1}^\star(s,a) & V_h^\star(s) &= \max_a Q_h^\star(s,a),
\end{aligned}
$$

where by definition, $V_{H+1}^\star \triangleq V_{H+1}^\pi \triangleq 0$. Furthermore, $p_h f > (s,a) \triangleq \mathbb{E}_{s'\sim p_h(\cdot|s,a)}[f(s')]$ denotes the expectation operator with respect to the transition probabilities $p_h$ and $\pi_h g(s) \triangleq g(s, \pi_h(s))$ denotes the composition with the policy $\pi$ at step $h$.

**Learning problem**   The agent, to which the transitions are *unknown* (the rewards are assumed to be known[1] for simplicity), interacts with the environment during $T$ episodes of length $H$, with a *fixed* initial state $s_1$.[2] Before each episode $t$ the agent selects a policy $\pi^t$ based only on the past observed transitions up to episode $t-1$. At each step $h \in [H]$ in episode $t$, the agent observes a state $s_h^t \in \mathcal{S}$, takes an action $\pi_h^t(s_h^t) = a_h^t \in \mathcal{A}$ and makes a transition to a new state $s_{h+1}^t$ according to the probability distribution $p_h(s_h^t, a_h^t)$ and receives a deterministic reward $r_h(s_h^t, a_h^t)$.

**Regret**   The quality of an agent is measured through its regret, that is the difference between what it could obtain (in expectation) by acting optimally and what it really gets,

$$
\mathfrak{R}^T \triangleq \sum_{t=1}^T V_1^\star(s_1) - V_1^{\pi^t}(s_1).
$$

---

[1] Our work can be extended without too much difficulty to the case of random rewards.

[2] As explained by Fiechter [1994] if the first state is sampled randomly as $s_1 \sim p$, we can simply add an artificial first state $s_{1'}$ such that for any action $a$, the transition probability is defined as the distribution $p_{1'}(s_{1'}, a) \triangleq p$.

**Additional notation** For $N \in \mathbb{N}_{++}$, we define the set $[N] \triangleq \{1, \dots, N\}$. We denote the uniform distribution over this set by $\mathrm{Unif}[N]$. We define the beta distribution with parameters $\alpha, \beta$ as $\mathrm{Beta}(\alpha, \beta)$. Appendix A references all the notation used.

## 3  Randomized Q-learning for Tabular Environments

In this section we assume that the state space $\mathcal{S}$ is finite of size $S$ as well as the action space $\mathcal{A}$ of size $A$. We first provide some intuitions for `RandQL` algorithm.

### 3.1  Concept

The main idea of `RandQL` is to perform the usual Q-learning updates but instead of adding bonuses to the targets as `OptQL` to drive exploration, `RandQL` injects noise into the updates of the Q-values through *noisy learning rates*. Precisely, for $J \in \mathbb{N}$, we maintain an ensemble of size $J$ of Q-values[3] $(\overline{Q}^{n,j})_{j \in [J]}$ updated with random independent Beta-distributed step-sizes $(w_{n,j})_{j \in [J]}$ where $w_{n,j} \sim \mathrm{Beta}(H, n)$. Then, policy Q-values $\overline{Q}^n$ are obtained by taking the maximum among the Q-values of the ensemble

$$\overline{Q}_h^{n+1,j}(s,a) = (1 - w_{n,j})\overline{Q}_h^{n,j}(s,a) + w_{n,j}[r_h(s,a) + \overline{V}_{h+1}^n(s_{h+1}^n)]$$
$$\overline{Q}_h^{n+1}(s,a) = \max_{j \in [J]} \overline{Q}_h^{n+1,j}(s,a), \quad \overline{V}_h^{n+1}(s) = \max_{a \in \mathcal{A}} \overline{Q}_h^{n+1}(s,a),$$

where $s_{h+1}^n$ stands for the next (in time) state after $n$-th visitation of $(s,a)$ at step $h$.

Note that the policy Q-values $\overline{Q}^n$ are designed to be upper confidence bound on the optimal Q-values. The policy used to interact with the environment is greedy with respect to the policy Q-values $\pi_h^n(s) \in \arg\max_a \overline{Q}_h^n(s,a)$. We provide a formal description of `RandQL` in Appendix B.

**Connection with** `OptQL`  We observe that the learning rates of `RandQL` are in expectation of the same order $\mathbb{E}[w_{n,j}] = H/(n+H)$ as the ones used by the `OptQL` algorithm. Thus, we can view our randomized Q-learning as a noisy version of the `OptQL` algorithm that doesn't use bonuses.

**Connection with** `PSRL`  If we unfold the recursive formula above we can express the Q-values $\overline{Q}^{n+1,j}$ as a weighted sum

$$\overline{Q}_h^{n+1,j}(s,a) = W_{n,j}^0 \overline{Q}_h^{1,j}(s,a) + \sum_{k=1}^n W_{n,j}^k [r_h(s,a) + \overline{V}_{h+1}^k(s_{h+1}^k)],$$

where we define $W_{n,j}^0 = \prod_{\ell=0}^{n-1}(1 - w_{\ell,j})$ and $W_{n,j}^k = w_{k-1,j} \prod_{\ell=k}^{n-1}(1 - w_{\ell,j})$.

To compare, we can unfold the corresponding formula for `PSRL` algorithm using the aggregation properties of the Dirichlet distribution (see e.g. Section 4 of Tiapkin et al. [2022b] or Appendix C)

$$\overline{Q}_h^{n+1}(s,a) = \widetilde{W}_n^0 \overline{Q}_h^1(s,a) + \sum_{k=1}^n \widetilde{W}_n^k [r_h(s,a) + \overline{V}_{h+1}^{n+1}(s_{h+1}^k)], \tag{1}$$

where weights $(\widetilde{W}_n^0, \dots, \widetilde{W}_n^n)$ follows Dirichlet distribution $\mathrm{Dir}(n_0, 1, \dots, 1)$ and $n_0$ is a weight for the prior distribution. In particular, one can represent these weights as partial products of other weights $w_n \sim \mathrm{Beta}(1, n + n_0)$. If we use (1) to construct a model-free algorithm, this would require recomputing the targets $r_h(s,a) + \overline{V}^{n+1}(s_{h+1}^k)$ in each iteration. To make algorithm more efficient and model-free, we approximate $\overline{V}^{n+1}$ by $\overline{V}^k$, and, as a result, obtain `RandQL` algorithm with weight distribution $w_{n,j} \sim \mathrm{Beta}(1, n + n_0)$.

Note that in expectation this algorithm is equivalent to `OptQL` with the uniform step-sizes which are known to be sub-optimal due to a high bias (see discussion in Section 3 of [Jin et al., 2018]). There are two known ways to overcome this sub-optimality for Q-learning: to introduce more aggressive

---

[3]We index the quantities by $n$ in this section where $n$ is the number of times the state-action pair $(s,a)$ is visited. In particular this is different from the global time $t$ since, in our setting, all the state- action pair are not visited at each episode. See Section 3.2 and Appendix B precise notations.

learning rates $w_{n,j} \sim \text{Beta}(H, n + n_0)$ leading to `RandQL` algorithm, or to use stage-dependent framework by Bai et al. [2019], Zhang et al. [2020] resulting in `Staged-RandQL` algorithm.

The aforementioned transition from `PSRL` to `RandQL` is similar to the transition from `UCBVI` [Azar et al., 2017] to Q-learning. To make `UCBVI` model-free, one has to to keep old targets in Q-values. This, however, introduces a bias that could be eliminated either by more aggressive step-size [Jin et al., 2018] or by splitting on stages [Bai et al., 2019]. Our algorithms (`RandQL` and `Staged-RandQL`) perform the similar tricks for `PSRL` and thus could be viewed as model-free versions of it. Additionally, `RandQL` shares some similarities with the `OPSRL` algorithm [Agrawal and Jia, 2017, Tiapkin et al., 2022a] in the way of introducing optimism (taking maximum over $J$ independent ensembles of Q-values). Let us also mention a close connection to the theory of Dirichlet processes in the proof of optimism for the case of metric spaces (see Remark 1 in Appendix E.4).

**Prior** As remarked above, in expectation, `RandQL` has a learning rate of the same order as `OptQL`. In particular, it implies that the first $(1 - 1/H)$ fraction of the the target will be forgotten exponentially fast in the estimation of the Q-values, see Jin et al. [2018], Ménard et al. [2021]. Thus we need to re-inject prior targets, as explained in Appendix B, in order to not forget too quickly the prior and thus replicate the same exploration mechanism as in the `PSRL` algorithm.

## 3.2 Algorithm

In this section, following Bai et al. [2019], Zhang et al. [2020], we present the `Staged-RandQL` algorithm a scheduled version of `RandQL` that is simpler to analyse. The main idea is that instead of using a carefully tuned learning rate to keep only the last $1/H$ fraction of the targets we split the learning of the Q-values in stages of exponentially increasing size with growth rate of order $1 + 1/H$. At a given stage, the estimate of the Q-value relies only on the targets within this stage and resets at the beginning of the next stage. Notice that the two procedures are almost equivalent. A detail description of `Staged-RandQL` is provided in Algorithm 1.

**Counts and stages** Let $n_h^t(s,a) \triangleq \sum_{i=1}^{t-1} \mathbb{1}\{(s_h^i, a_h^i) = (s,a)\}$ be the number of visits of state-action pair $(s,a)$ at step $h$ before episode $t$. We say that a triple $(s, a, h)$ belongs to the $k$-th stage at the beginning of episode $t$ if $n_h^t(s,a) \in [\sum_{i=0}^{k-1} e_i, \sum_{i=0}^{k} e_i)$. Here $e_k = \lfloor (1 + 1/H)^k \cdot H \rfloor$ is the length of the stage $k \geq 0$ and, by convention, $e_{-1} = 0$. Let $\widetilde{n}_h^t(s,a) \triangleq n_h^t(s,a) - \sum_{i=0}^{k-1} e_i$ be the number of visits of state-action pair $(s,a)$ at step $h$ during the current stage $k$.

**Temporary Q-values** At the beginning of a stage, let say time $t$, we initialize $J$ *temporary* Q-values as $\widetilde{Q}_h^{t,j}(s,a) = r_h(s,a) + r_0(H - h - 1)$ for $j \in [J]$ and $r_0$ some pseudo-reward. Then as long as $(s_h^t, a_h^t, h)$ remains within a stage we update recursively the *temporary* Q-values

$$\widetilde{Q}_h^{t+1,j}(s,a) = \begin{cases} (1 - w_{j,\widetilde{n}})\widetilde{Q}_h^{t,j}(s,a) + w_{j,\widetilde{n}}[r_h(s,a) + \overline{V}_{h+1}^t(s_{h+1}^t)], & (s,a) = (s_h^t, a_h^t) \\ \widetilde{Q}_h^{t,j}(s,a) & \text{otherwise,} \end{cases}$$

where $\widetilde{n} = \widetilde{n}_h^t(s,a)$ is the number of visits, $w_{j,\widetilde{n}}$ is a sequence of i.i.d. random variables $w_{j,\widetilde{n}} \sim \text{Beta}(1/\kappa, (\widetilde{n} + n_0)/\kappa)$ with $\kappa > 0$ being some posterior inflation coefficient and $n_0$ a number of pseudo-transitions.

**Policy Q-values** Next we define the policy Q-values that is updated at the end of a stage. Let say for state-action pair $(s,a)$ at step $h$ an stage ends at time $t$. This policy Q-values is then given by the maximum of temporary Q-values $\overline{Q}_h^{t+1} = \max_{j \in [J]} \widetilde{Q}_h^{t+1,j}(s,a)$. Then the policy Q-values is constant within a stage. The value used to defined the targets is $\overline{V}_h^{t+1}(s) = \max_{a \in \mathcal{A}} \overline{Q}_h^{t+1}(s,a)$. The policy used to interact with the environment is greedy with respect to the policy Q-values $\pi_h^{t+1}(s) \in \arg\max_{a \in \mathcal{A}} \overline{Q}_h^{t+1}(s,a)$ (we break ties arbitrarily).

## 3.3 Regret bound

We fix $\delta \in (0, 1)$ and the number of posterior samples $J \triangleq \lceil c_J \cdot \log(2SAHT/\delta) \rceil$, where $c_J = 1/\log(2/(1 + \Phi(1)))$ and $\Phi(\cdot)$ is the cumulative distribution function (CDF) of a normal distribution. Note that $J$ has a logarithmic dependence on $S, A, H, T$, and $1/\delta$.

We now state the regret bound of `Staged-RandQL` with a full proof in Appendix D.

---

**Algorithm 1** Tabular `Staged-RandQL`

---

1: **Input:** inflation coefficient $\kappa$, $J$ ensemble size, number of prior transitions $n_0$, prior reward $r_0$.
2: **Initialize:** $\overline{V}_h(s) = \overline{Q}_h(s,a) = \widetilde{Q}_h^j(s,a) = r(s,a) + r_0(H - h - 1)$, initialize counters $\widetilde{n}_h(s,a) = 0$ for $j, h, s, a \in [J] \times [H] \times \mathcal{S} \times \mathcal{A}$ and stage $q_h(s,a) = 0$.
3: **for** $t \in [T]$ **do**
4:     **for** $h \in [H]$ **do**
5:         Play $a_h \in \arg\max_a \overline{Q}_h(s_h, a)$.
6:         Observe reward and next state $s_{h+1} \sim p_h(s_h, a_h)$.
7:         Sample learning rates $w_j \sim \text{Beta}(1/\kappa, (\widetilde{n} + n_0)/\kappa)$ for $\widetilde{n} = \widetilde{n}_h(s_h, a_h)$.
8:         Update temporary $Q$-values for all $j \in [J]$

$$\widetilde{Q}_h^j(s_h, a_h) := (1 - w_j)\widetilde{Q}_h^j(s_h, a_h) + w_j\big(r_h(s_h, a_h) + \overline{V}_{h+1}(s_{h+1})\big).$$

9:         Update counter $\widetilde{n}_h(s_h, a_h) := \widetilde{n}_h(s_h, a_h) + 1$
10:         **if** $\widetilde{n}_h(s_h, a_h) = \lfloor (1 + 1/H)^q H \rfloor$ for $q = q_h(s_h, a_h)$ being the current stage **then**
11:             Update policy $Q$-values $\overline{Q}_h(s_h, a_h) := \max_{j \in [J]} \widetilde{Q}_h^j(s_h, a_h)$.
12:             Update value function $\overline{V}_h(s_h) := \max_{a \in \mathcal{A}} \overline{Q}_h(s_h, a)$
13:             Reset temporary $Q$-values $\widetilde{Q}_h^j(s_h, a_h) := r_h(s_h, a_h) + r_0(H - h - 1)$.
14:             Reset counter $\widetilde{n}_h(s_h, a_h) := 0$ and change stage $q_h(s_h, a_h) := q_h(s_h, a_h) + 1$.
15:         **end if**
16:     **end for**
17: **end for**

---

**Theorem 1.** *Consider a parameter $\delta \in (0, 1)$. Let $\kappa \triangleq 2(\log(8SAH/\delta) + 3\log(\mathrm{e}\pi(2T+1)))$, $n_0 \triangleq \lceil \kappa(c_0 + \log_{17/16}(T)) \rceil$, $r_0 \triangleq 2$, where $c_0$ is an absolute constant defined in* (5); *see Appendix D.3. Then for* `Staged-RandQL`, *with probability at least $1 - \delta$,*

$$\mathfrak{R}^T = \widetilde{\mathcal{O}}\Big(\sqrt{H^5 SAT} + H^3 SA\Big).$$

**Discussion** The regret bound of Theorem 1 coincides (up to a logarithmic factor) with the bound of the `OptQL` algorithm with Hoeffding-type bonuses from Jin et al. [2018]. Up to a $H$ factor, our regret matches the information-theoretic lower bound $\Omega(\sqrt{H^3 SAT})$ [Jin et al., 2018, Domingues et al., 2021b]. This bound could be achieved (up to logarithmic terms) in model-free algorithms by using Bernstein-type bonuses and variance reduction [Zhang et al., 2020]. We keep these refinements for future research as the main focus of our paper is on the novel randomization technique and its use to construct computationally tractable model-free algorithms.

**Computational complexity** `Staged-RandQL` is a model-free algorithm, and thus gets the $\widetilde{\mathcal{O}}(HSA)$ space complexity as `OptQL`, recall that we set $J = \widetilde{\mathcal{O}}(1)$. The per-episode time-complexity is also similar and of order $\widetilde{\mathcal{O}}(H)$.

## 4 Randomized Q-learning for Metric Spaces

In this section we present a way to extend `RandQL` to general state-action spaces. We start from the simplest approach with predefined $\varepsilon$-net type discretization of the state-action space $\mathcal{S} \times \mathcal{A}$ (see Song and Sun 2019), and then discuss an adaptive version of the algorithm, similar to one presented by Sinclair et al. [2019].

### 4.1 Assumptions

To pose the first assumption, we start from a general definition of covering numbers.

**Definition 1** (Covering number and covering dimension). Let $(M, \rho)$ be a metric space. A set $\mathcal{M}$ of open balls of radius $\varepsilon$ is called an $\varepsilon$-cover of $M$ if $M \subseteq \bigcup_{B \in \mathcal{M}} B$. The cardinality of the minimal $\varepsilon$-cover is called covering number $N_\varepsilon$ of $(M, \rho)$. We denote the corresponding minimal $\varepsilon$-covering by $\mathcal{N}_\varepsilon$. A metric space $(M, \rho)$ has a covering dimension $d_c$ if $\forall \varepsilon > 0 : N_\varepsilon \leq C_N \varepsilon^{-d_c}$, where $C_N$ is a constant.

The last definition extends the definition of dimension beyond vector spaces. For example, is case of $M = [0, 1]^d$ the covering dimension of $M$ is equal to $d$. For more details and examples see e.g. Vershynin [2018, Section 4.2].

Next we are ready to introduce the first assumption.

**Assumption 1** (Metric Assumption). Spaces $\mathcal{S}$ and $\mathcal{A}$ are separable compact metric spaces with the corresponding metrics $\rho_{\mathcal{S}}$ and $\rho_{\mathcal{A}}$. The joint space $\mathcal{S} \times \mathcal{A}$ endowed with a product metric $\rho$ that satisfies $\rho((s, a), (s', a')) \leq \rho_{\mathcal{S}}(s, s') + \rho_{\mathcal{A}}(a, a')$. Moreover, the diameter of $\mathcal{S} \times \mathcal{A}$ is bounded by $d_{\max}$, and $\mathcal{S} \times \mathcal{A}$ has covering dimension $d_c$ with a constant $C_N$.

This assumption is, for example, satisfied for the finite state and action spaces endowed with discrete metrics $\rho_{\mathcal{S}}(s, s') = \mathbb{1}\{s \neq s'\}, \rho_{\mathcal{A}}(a, a') = \mathbb{1}\{a \neq a'\}$ with $d_c = 0, C_N = SA$ and $S$ and $A$ being the cardinalities of the state and action spaces respectively. The above assumption also holds in the case $\mathcal{S} \subseteq [0, 1]^{d_{\mathcal{S}}}$ and $\mathcal{A} \subseteq [0, 1]^{d_{\mathcal{A}}}$ with $d_c = d_{\mathcal{S}} + d_{\mathcal{A}}$.

The next two assumptions describe the regularity conditions of transition kernel and rewards.

**Assumption 2** (Reparametrization Assumption). The Markov transition kernel could be represented as an iterated random function. In other words, there exists a measurable space $(\Xi, \mathcal{F}_{\Xi})$ and a measurable function $F_h \colon (\mathcal{S} \times \mathcal{A}) \times \Xi \to \mathcal{S}$, such that $s_{h+1} \sim p_h(s_h, a_h) \iff s_{h+1} = F_h(s_h, a_h, \xi_h)$ for a sequence of independent random variables $\{\xi_h\}_{h \in [H]}$.

This assumption is naturally satisfied for a large family of probabilistic model, see Kingma and Welling [2014]. Moreover, it has been utilized by the RL community both in theory [Ye and Zhou, 2015] and practice [Heess et al., 2015, Liu et al., 2018]. Essentially, this assumption holds for Markov transition kernels over a separable metric space, see Theorem 1.3.6 by Douc et al. [2018]. However, the function $F_h$ could be ill-behaved. To avoid this behaviour, we need the following assumption.

**Assumption 3** (Lipschitz Assumption). The function $F_h(\cdot, \xi_h)$ is $L_F$-Lipschitz in the first argument for almost every value of $\xi_h$. Additionally, the reward function $r_h \colon \mathcal{S} \times \mathcal{A} \to [0, 1]$ is $L_r$-Lipschitz.

This assumption is commonly used in studies of the Markov processes corresponding to iterated random functions, see Diaconis and Freedman [1999], Ghosh and Marecek [2022]. Moreover, this assumption holds for many cases of interest. As main example, it trivially holds in tabular and Lipschitz continuous deterministic MDPs [Ni et al., 2019]. Notably, this observation demonstrates that Assumption 3 does not necessitate Lipschitz continuity of the transition kernels in total variation distance, since deterministic Lipschitz MDPs are not continuous in that sense. Additionally, incorporation of an additive noise to deterministic Lipschitz MDPs will lead to Assumption 3 with $L_F = 1$.

Furthermore, it is possible to show that Assumption 3 implies other assumptions stated in the literature. For example, it implies that the transition kernel is Lipschitz continuous in 1-Wasserstein metric, and that $Q^\star$ and $V^\star$ are both Lipschitz continuous.

**Lemma 1.** *Let Assumption 1,2,3 hold. Then the transition kernels $p_h(s, a)$ are $L_F$-Lipschitz continuous in 1-Wasserstein distance*

$$\mathcal{W}_1(p_h(s, a), p_h(s', a')) \leq L_F \cdot \rho((s, a), (s', a')),$$

*where 1-Wasserstein distance between two probability measures on the metric space $(M, \rho)$ is defined as $\mathcal{W}_1(\nu, \eta) = \sup_{f \text{ is } 1-Lipschitz} \int_M f \mathrm{d}\nu - \int_M f \mathrm{d}\eta$.*

**Lemma 2.** *Let Assumption 1,2,3 hold. Then $Q_h^\star$ and $V_h^\star$ are Lipschitz continuous with Lipschitz constant $L_{V,h} \leq \sum_{h'=h}^{H} L_F^{h'-h} L_r$.*

The proof of these lemmas is postponed to Appendix E. For a more detailed exposition on 1-Wasserstein distance we refer to the book by Peyré and Cuturi [2019]. The first assumption was studied by Domingues et al. [2021c], Sinclair et al. [2023] in the setting of model-based algorithms in metric spaces. We are not aware of any natural examples of MDPs with a compact state-action space where the transition kernels are Lipschitz in $\mathcal{W}_1$ but fail to satisfy Assumption 3.

## 4.2 Algorithms

In this section, following Song and Sun [2019], we present `Net-Staged-RandQL` algorithm that combines a simple non-adaptive discretization and an idea of stages by Bai et al. [2019], Zhang et al. [2020].

We assume that we have an access to all Lipschitz constants $L_r, L_F, L_V \triangleq L_{V,1}$. Additionally, we have access to the oracle that computes $\varepsilon$-cover $\mathcal{N}_\varepsilon$ of the space $\mathcal{S} \times \mathcal{A}$ for any predefined $\varepsilon > 0$[4].

**Counts and stages** Let $n_h^t(B) \triangleq \sum_{i=1}^{t-1} \mathbb{1}\{(s_h^i, a_h^i) \in B\}$ be the number of visits of the ball $B \in \mathcal{N}_\varepsilon$ at step $h$ before episode $t$. Let $e_k = \lfloor (1 + 1/H)^k \cdot H \rceil$ be length of the stage $k \geq 0$ and, by convention, $e_{-1} = 0$. We say that $(B, h)$ belongs to the $k$-th stage at the beginning of episode $t$ if $n_h^t(B) \in [\sum_{i=0}^{k-1} e_i, \sum_{i=0}^{k} e_i)$. Let $\widetilde{n}_h^t(B) \triangleq n_h^t(s, a) - \sum_{i=0}^{k-1} e_i$ be the number of visits of the ball $B$ at step $h$ during the current stage $k$.

**Temporary Q-values** At the beginning of a stage, let say time $t$, we initialize $J$ *temporary* Q-values as $\widetilde{Q}_h^{t,j}(B) = r_0 H$ for $j \in [J]$ and $r_0$ some pseudo-reward. Then within a stage $k$ we update recursively the *temporary* Q-values

$$\widetilde{Q}_h^{t+1,j}(B) = \begin{cases} (1 - w_{j,\widetilde{n}})\widetilde{Q}_h^{t,j}(B) + w_{j,\widetilde{n}}[r_h(s_h^t, a_h^t) + \overline{V}_{h+1}^t(s_{h+1}^t)], & (s, a) = (s_h^t, a_h^t) \\ \widetilde{Q}_h^{1,j}(B) & \text{otherwise,} \end{cases}$$

where $\widetilde{n} = \widetilde{n}_h^t(B)$ is the number of visits, $w_{j,\widetilde{n}}$ is a sequence of i.i.d random variables $w_{j,\widetilde{n}} \sim \mathrm{Beta}(1/\kappa, (\widetilde{n} + n_0(k))/\kappa)$ with $\kappa > 0$ some posterior inflation coefficient and $n_0(k)$ a number of pseudo-transitions. The important difference between tabular and metric settings is the dependence on the pseudo-count $n_0(k)$ on $k$ in the latter case, since here the prior is used to eliminate the approximation error.

**Policy Q-values** Next, we define the policy Q-values that are updated at the end of a stage. Let us fix a ball $B$ at step $h$ and suppose that the currents stage ends at time $t$. Then the policy Q-values are given by the maximum of the temporary Q-values $\overline{Q}_h^{t+1}(B) = \max_{j \in [J]} \widetilde{Q}_h^{t+1,j}(B)$. The policy Q-values are constant within a stage. The value used to define the targets is computed on-flight using the formula $\overline{V}_h^t(s) = \max_{a \in \mathcal{A}} \overline{Q}_h^t(\psi_\varepsilon(s, a))$, where $\psi_\varepsilon : \mathcal{S} \times \mathcal{A} \to \mathcal{N}_\varepsilon$ is a quantization map, that assigns each state-action pair $(s, a)$ to a ball $B \ni (s, a)$. The policy used to interact with the environment is greedy with respect to the policy Q-values and also computed on-flight $\pi_h^t(s) \in \arg\max_{a \in \mathcal{A}} \overline{Q}_h^t(\psi_\varepsilon(s, a))$ (we break ties arbitrarily).

A detail description of `Net-Staged-RandQL` is provided in Algorithm 4 in Appendix E.2.

## 4.3 Regret Bound

We fix $\delta \in (0, 1)$, the discretization level $\varepsilon > 0$ and the number of posterior samples

$$J \triangleq \lceil \tilde{c}_J \cdot (\log(2C_N HT/\delta) + d_c \log(1/\varepsilon)) \rceil,$$

where $\tilde{c}_J = 1/\log(4/(3 + \Phi(1)))$ and $\Phi(\cdot)$ is the cumulative distribution function (CDF) of a normal distribution. Note that $J$ has a logarithmic dependence on $H, T, 1/\varepsilon$ and $1/\delta$. For the regret-optimal discretization level $\varepsilon = T^{-1/(d_c+2)}$, the number $J$ is almost independent of $d_c$ . Let us note that the role of prior in metric spaces is much higher than in the tabular setting. Another important difference is dependence of the prior count on the stage index. In particular, we have

$$n_0(k) = \left\lceil \widetilde{n}_0 + \kappa + \frac{\varepsilon L}{H - 1} \cdot (e_k + \widetilde{n}_0 + \kappa) \right\rceil, \qquad \widetilde{n}_0 = (c_0 + 1 + \log_{17/16}(T)) \cdot \kappa$$

where $c_0$ is an absolute constant defined in (5) ( see Appendix D.3), $\kappa$ is the posterior inflation coefficient and $L = L_r + (1 + L_F)L_V$ is a constant. We now state the regret bound of `Net-Staged-RandQL` with a full proof being postponed to Appendix E.

**Theorem 2.** *Suppose that $N_\varepsilon \leq C_N \varepsilon^{-d_c}$ for all $\varepsilon > 0$ and some constant $C_N > 0$. Consider a parameter $\delta \in (0, 1)$ and take an optimal level of discretization $\varepsilon = T^{-1/(d_c+2)}$. Let $\kappa \triangleq 2(\log(8HC_N/\delta) + d_c \log(1/\varepsilon) + 3\log(e\pi(2T+1)))$, $r_0 \triangleq 2$. Then it holds for `Net-Staged-RandQL`, with probability at least $1 - \delta$,*

$$\mathfrak{R}^T = \widetilde{\mathcal{O}}\left( H^{5/2} C_N^{1/2} T^{\frac{d_c+1}{d_c+2}} + H^3 C_N T^{\frac{d_c}{d_c+2}} + LT^{\frac{d_c+1}{d_c+2}} \right).$$

We can restore the regret bound in the tabular setting by letting $d_c = 0$ and $C_N = SA$, where $S$ is the cardinality of the state-space, and $A$ is the cardinality of the action-space.

---

[4]Remark that the simple greedy algorithm can generate $\varepsilon$-cover of size $N_{\varepsilon/2}$, that will not affect the asymptotic behavior of our regret bounds, see Song and Sun [2019].

**Discussion** From the point of view of instance-independent bounds, our algorithm achieves the same result as `Net-QL` [Song and Sun, 2019] and `Adaptive-QL` [Sinclair et al., 2019], that matches the lower bound $\Omega(HT^{\frac{d_c+1}{d_c+2}})$ by Sinclair et al. [2023] in dependence on budget $T$ and covering dimension $d_c$. Notably, as discussed by Sinclair et al. [2023], the model-based algorithm such as `Kernel-UCBVI` [Domingues et al., 2021c] does not achieves optimal dependence in $T$ due to hardness of the transition estimation problem.

**Computational complexity** For a fixed level of discretization $\varepsilon$, our algorithm has a space complexity of order $\widetilde{\mathcal{O}}(H\mathcal{N}_\varepsilon)$. Assuming that the computation of a quantization map $\psi_\varepsilon$ has $\widetilde{\mathcal{O}}(1)$ time complexity, we achieve a per-episode time complexity of $\widetilde{\mathcal{O}}(HA)$ for a finite action space and $\mathcal{O}(HN_\varepsilon)$ for an infinite action space in the worst case due to computation of $\arg\max_{a\in\mathcal{A}} \overline{Q}_h(\psi_\varepsilon(s,a))$. However, this can be improved to $\widetilde{\mathcal{O}}(H)$ if we consider adaptive discretization [Sinclair et al., 2019].

**Adaptive discretization** Additionally, we propose a way to combine `RandQL` with adaptive discretization by Cao and Krishnamurthy [2020], Sinclair et al. [2023]. This combination results in two algorithms: `Adaptive-RandQL` and `Adaptive-Staged-RandQL`. The second one could achieve the instance-dependent regret bound that scales with a *zooming* dimension, the instance-dependent measure of dimension. We will follow Sinclair et al. [2023] in the exposition of the required notation.

**Definition 2.** For any $(s,a) \in \mathcal{S} \times \mathcal{A}$, the stage-dependent sub-optimality gap is defined as $\mathrm{gap}_h(s,a) = V_h^\star(s) - Q_h^\star(s,a)$.

This quantity is widely used in the theoretical instance-dependent analysis of reinforcement learning and contextual bandit algorithms.

**Definition 3.** The near-optimal set of $\mathcal{S} \times \mathcal{A}$ for a given value $\varepsilon$ defined as $Z_h^\varepsilon = \{(s,a) \in \mathcal{S} \times \mathcal{A} \mid \mathrm{gap}_h(s,a) \leq (H+1)\varepsilon\}$.

The main insight of this definition is that essentially we are interested in a detailed discretization of the near-optimal set $Z_h^\varepsilon$ for small $\varepsilon$, whereas all other state-action pairs could be discretized in a more rough manner. Interestingly enough, $Z_h^\varepsilon$ could be a lower dimensional manifold, leading to the following definition.

**Definition 4.** The step-$h$ zooming dimension $d_{z,h}$ with a constant $C_{N,h}$ and a scaling factor $\rho > 0$ is given by
$$d_{z,h} = \inf\{d > 0 : \forall \varepsilon > 0 \; N_\varepsilon(Z_h^{\rho\cdot\varepsilon}) \leq C_{N,h}\varepsilon^{-d}\}.$$

Under some additional structural assumptions on $Q_h^\star$, it is possible to show that the zooming dimension could be significantly smaller than the covering dimension, see, e.g., Lemma 2.8 in Sinclair et al. [2023]. However, at the same time, it has been shown that $d_{z,h} \geq d_\mathcal{S} - 1$, where $d_\mathcal{S}$ is a covering dimension of the state space. Thus, the zooming dimension allows adaptation to a rich action space but not a rich state space.

Given this definition, it is possible to define define an adaptive algorithm `Adaptive-Staged-RandQL` that attains the following regret guarantees

**Theorem 3.** *Consider a parameter $\delta \in (0,1)$. For a value $\kappa$ that depends on $T, d_c$ ad $\delta$, for* `Adaptive-Staged-RandQL` *the following holds with probability at least $1 - \delta$,*

$$\mathfrak{R}^T = \widetilde{\mathcal{O}}\left(H^3 + H^{3/2}\sum_{h=1}^{H} T^{\frac{d_{z,h}+1}{d_{z,h}+2}}\right),$$

*where $d_{z,h}$ is the step-$h$ zooming dimension and we ignore all multiplicative factors in the covering dimension $d_c$, $\log(C_N)$, and Lipschitz constants.*

We refer to Appendix F to a formal statement and a proof.

# 5 Experiments

In this section we present the experiments we conducted for tabular environments using `rlberry` library [Domingues et al., 2021a]. We also provide experiments in non-tabular environment in Appendix I.

**Environment**  We use a grid-world environment with 100 states $(i, j) \in [10] \times [10]$ and 4 actions (left, right, up and down). The horizon is set to $H = 50$. When taking an action, the agent moves in the corresponding direction with probability $1 - \varepsilon$, and moves to a neighbor state at random with probability $\varepsilon = 0.2$. The agent starts at position $(1, 1)$. The reward equals to 1 at the state $(10, 10)$ and is zero elsewhere.

**Variations of randomized Q-learning**  For the tabular experiment we use the `RandQL` algorithm, described in Appendix B as it is the version of randomized Q-learning that is the closest to the baseline `OptQL`. Note that, we compare the different versions of randomized Q-learning in Appendix B.

**Baselines**  We compare `RandQL` algorithm to the following baselines: (i) `OptQL` [Jin et al., 2018] (ii) `UCBVI` [Azar et al., 2017] (iii) `Greedy-UCBVI`, a version of `UCBVI` using real–time dynamic programming [Efroni et al., 2019] (iv) `PSRL` [Osband et al., 2013] and (v) `RLSVI` [Russo, 2019]. For the hyper-parameters used for these baselines refer to Appendix I.

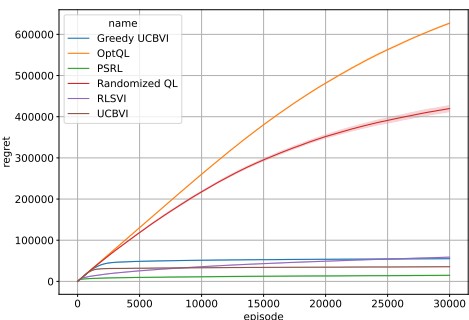

Figure 1: Regret curves of `RandQL` and baselines in a grid-world environment for $H = 50$ and transition noise $\varepsilon = 0.2$. The average is over 4 seeds.

**Results**  Figure 1 shows the result of the experiments. Overall, we see that `RandQL` outperforms `OptQL` algorithm on tabular environment, but still degrades in comparison to model-based approaches, that is usual for model-free algorithms in tabular environments. Indeed, using a model and backward induction allows new information to be more quickly propagated. But as counterpart, `RandQL` has a better time-complexity and space-complexity than model-based algorithm, see Table 2 in Appendix I.

## 6   Conclusion

This paper introduced the `RandQL` algorithm, a new model-free algorithm that achieves exploration without bonuses. It utilizes a novel idea of learning rate randomization, resulting in provable sample efficiency with regret of order $\widetilde{\mathcal{O}}(\sqrt{H^5 SAT})$ in the tabular case. We also extend `RandQL` to the case of metric state-action space by using proper discretization techniques. The proposed algorithms inherit the good empirical performance of model-based Bayesian algorithm such that `PSRL` while keeping the small space and time complexity of model-free algorithm. Our result rises following interesting open questions for a further research.

**Optimal rate for `RandQL`**  We conjecture that `RandQL` could get optimal regret in the tabular setting if coupled with variance reductions techniques as used by Zhang et al. [2020]. However, obtaining such improvements is not straightforward due to the intricate statistical dependencies involved in the analysis of `RandQL`.

**Beyond one-step learning**  We observe a large gap in the experiments between Q-learning type algorithm that do one-step planning and e.g. `UCBVI` algorithm that does full planning or `Greedy-UCBVI` that does one-step planning with full back-up (expectation under transition of the model) for all actions. Therefore, it would interesting to study also algorithms that range between these two extremes [Efroni et al., 2018, 2019].

## Acknowledgments

The work of D. Tiapkin, A. Naumov, and D. Belomestny were supported by the grant for research centers in the field of AI provided by the Analytical Center for the Government of the Russian Federation (ACRF) in accordance with the agreement on the provision of subsidies (identifier of the agreement 000000D730321P5Q0002) and the agreement with HSE University No. 70-2021-00139. E. Moulines received support from the grant ANR-19-CHIA-002 SCAI and parts of his work has been done under the auspices of Lagrange Center for maths and computing. P. Ménard acknowledges the Chaire SeqALO (ANR-20-CHIA-0020-01). This research was supported in part through computational resources of HPC facilities at HSE University.

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

# Appendix

## Table of Contents

# A   Notation

Table 1: Table of notation use throughout the paper for the tabular setting

| Notation | Meaning |
|---|---|
| $\mathcal{S}$ | state space of size $S$ |
| $\mathcal{A}$ | action space of size $A$ |
| $H$ | length of one episode |
| $T$ | number of episodes |
| $J$ | number of posterior samples |
| $r_h(s,a)$ | reward |
| $p_h(s'\|s,a)$ | probability transition |
| $Q_h^\pi(s,a)$ | Q-function of a given policy $\pi$ at step $h$ |
| $V_h^\pi(s)$ | V-function of a given policy $\pi$ at step $h$ |
| $Q_h^\star(s,a)$ | optimal Q-function at step $h$ |
| $V_h^\star(s)$ | optimal V-function at step $h$ |
| $\mathfrak{R}^T$ | regret |
| $n_0$ and $n_0(k)$ | number of pseudo-transitions |
| $s_0$ | optimistic pseudo-state |
| $r_0$ | pseudo-reward |
| $\kappa$ | posterior inflation parameter |
| $s_h^t$ | state that was visited at $h$ step during $t$ episode |
| $a_h^t$ | action that was picked at $h$ step during $t$ episode |
| $B_h^t$ | a ball that contains a pair $(s_h^t, a_h^t)$ |
| $n_h^t(s,a)$ | number of visits of state-action at the beginning of episode $t$ |
| | $n_h^t(s,a) = \sum_{k=1}^{t-1} \mathbb{1}\{(s_h^k, a_h^k) = (s,a)\}$ |
| $n_h^t(B)$ | number of visits of a ball $B$ at the beginning of episode $t$ |
| $e_k$ | length of $k$-th stage $e_k = \lfloor (1+1/H)^k H \rfloor$ for $k \geq 0$ and $e_{-1} = 0$ |
| $k_h^t(s,a)$ | index of stage previous to time $t$ at step $h$ and state-action pair $(s,a)$: |
| | $k_h^t(s,a) = \max\{k : n_h^t(s,a) \geq \sum_{i=0}^{k} e_i\}$ |
| $\widetilde{n}_h^t(s,a)$ | number of visits of state-action during the current stage: |
| | $\widetilde{n}_h^t(s,a) = n_h^t(s,a) - \sum_{i=0}^{k_h^t(s,a)} e_i$ |
| $\widetilde{n}_h^t(B)$ | number of visits of a ball $B$ during the current stage: |
| $\overline{V}_h^t(s)$ | upper approximation of the optimal V-value |
| $\overline{Q}_h^t(s,a)$ | upper approximation of the optimal Q-value |
| $\overline{Q}_h^t(B)$ | upper approximation of the optimal Q-value for all $(s,a) \in B$ |
| $\widetilde{Q}_h^{t,j}(s,a)$ | temporary estimate of the optimal Q-value |
| $\widetilde{Q}_h^{t,j}(B)$ | temporary estimate of the optimal Q-value for all $(s,a) \in B$ |
| $w_{n,j}$ | random learning rates |
| $\rho_\mathcal{S}, \rho_\mathcal{A}, \rho$ | metrics on $\mathcal{S}, \mathcal{A}$ and $\mathcal{S} \times \mathcal{A}$ correspondingly |
| $\mathcal{N}_\varepsilon$ | minimal $\varepsilon$-cover if $\mathcal{S} \times \mathcal{A}$ of size $N_\varepsilon$ |
| $d_c$ | covering dimension of space $\mathcal{S} \times \mathcal{A}$: $\forall \varepsilon > 0 : N_\varepsilon \leq C_N \varepsilon^{-d_c}$ |
| $d_{\max}$ | diameter of $\mathcal{S} \times \mathcal{A}$ |
| $F_h(s,a,\xi_h)$ | reparametrization function $s_{h+1} \sim p_h(s,a) \iff s_{h+1} = F_h(s,a,\xi_h)$ |
| $L_r, L_F$ | Lipschitz constants of rewards and reparametrization function |
| $L_V$ | Lipschitz constants of $Q_h^\star$ and $V_h^\star$ |

Let $(\mathsf{X}, \mathcal{X})$ be a measurable space and $\mathcal{P}(\mathsf{X})$ be the set of all probability measures on this space. For $p \in \mathcal{P}(\mathsf{X})$ we denote by $\mathbb{E}_p$ the expectation w.r.t. $p$. For random variable $\xi : \mathsf{X} \to \mathbb{R}$ notation $\xi \sim p$ means $\mathrm{Law}(\xi) = p$. We also write $\mathbb{E}_{\xi \sim p}$ instead of $\mathbb{E}_p$. For independent (resp. i.i.d.) random variables $\xi_\ell \overset{\mathrm{ind}}{\sim} p_\ell$ (resp. $\xi_\ell \overset{\mathrm{i.i.d}}{\sim} p$), $\ell = 1, \ldots, d$, we will write $\mathbb{E}_{\xi_\ell \overset{\mathrm{ind}}{\sim} p_\ell}$ (resp. $\mathbb{E}_{\xi_\ell \overset{\mathrm{i.i.d}}{\sim} p}$), to denote expectation w.r.t. product measure on $(\mathsf{X}^d, \mathcal{X}^{\otimes d})$. For any $x \in \mathsf{X}$ we denote $\delta_x$ a Dirac measure supported at point $x$.

For any $p, q \in \mathcal{P}(\mathsf{X})$ the Kullback-Leibler divergence $\mathrm{KL}(p, q)$ is given by

$$\mathrm{KL}(p, q) \triangleq \begin{cases} \mathbb{E}_p\left[\log \frac{\mathrm{d}p}{\mathrm{d}q}\right], & p \ll q, \\ +\infty, & \text{otherwise.} \end{cases}$$

For any $p \in \mathcal{P}(\mathsf{X})$ and $f: \mathsf{X} \to \mathbb{R}$, $pf = \mathbb{E}_p[f]$. In particular, for any $p \in \Delta_d$ and $f: \{0, \ldots, d\} \to \mathbb{R}$, $pf = \sum_{\ell=0}^d f(\ell)p(\ell)$. Define $\mathrm{Var}_p(f) = \mathbb{E}_{s' \sim p}\left[(f(s') - pf)^2\right] = p[f^2] - (pf)^2$. For any $(s, a) \in \mathcal{S}$, transition kernel $p(s, a) \in \mathcal{P}(\mathcal{S})$ and $f: \mathcal{S} \to \mathbb{R}$ define $pf(s, a) = \mathbb{E}_{p(s,a)}[f]$ and $\mathrm{Var}_p[f](s, a) = \mathrm{Var}_{p(s,a)}[f]$.

Let $(\mathsf{X}, \rho)$ be a metric space, then the 1-Wasserstein distance between $p, q \in \mathcal{P}(\mathsf{X})$ is defined as $\mathcal{W}_1(p, q) = \sup_{f \text{ is 1-Lipschitz}} \mathbb{E}_p[f] - \mathbb{E}_q[f]$.

We write $f(S, A, H, T) = \mathcal{O}(g(S, A, H, T, \delta))$ if there exist $S_0, A_0, H_0, T_0, \delta_0$ and constant $C_{f,g}$ such that for any $S \geq S_0, A \geq A_0, H \geq H_0, T \geq T_0, \delta < \delta_0, f(S, A, H, T, \delta) \leq C_{f,g} \cdot g(S, A, H, T, \delta)$. We write $f(S, A, H, T, \delta) = \widetilde{\mathcal{O}}(g(S, A, H, T, \delta))$ if $C_{f,g}$ in the previous definition is poly-logarithmic in $S, A, H, T, 1/\delta$.

For $\alpha, \beta > 0$, we define $\mathrm{Beta}(\alpha, \beta)$ as a beta distribution with parameters $\alpha, \beta$. For set $\mathsf{X}$ such that $|\mathsf{X}| < \infty$ define $\mathrm{Unif}(\mathsf{X})$ as a uniform distribution over this set. In particular, $\mathrm{Unif}[N]$ is a uniform distribution over a set $[N]$.

For a measure $p \in \mathcal{P}([0, b])$ supported on a segment $[0, b]$ (equipped with a Borel $\sigma$-algebra) and a number $\mu \in [0, b]$ we define

$$\mathcal{K}_{\inf}(p, \mu) \triangleq \inf\{\mathrm{KL}(p, q) : q \in \mathcal{P}([0, b]), p \ll q, \mathbb{E}_{X \sim q}[X] \geq \mu\}.$$

As the Kullback-Leibler divergence this quantity admits a variational formula by Lemma 18 of Garivier et al. [2018] up to rescaling for any $u \in (0, b)$

$$\mathcal{K}_{\inf}(p, \mu) = \max_{\lambda \in [0, 1/(b-\mu)]} \mathbb{E}_{X \sim p}[\log(1 - \lambda(X - \mu))].$$

# B  Description of `RandQL`

In this appendix we describe `RandQL` and `Sampled-RandQL` algorithms.

## B.1  `RandQL` algorithm

We recall that $n_h^t(s, a) = \sum_{i=1}^{t-1} \mathbb{1}\{(s_h^i, a_h^i) = (s, a)\}$ is the number of visits of state-action pair $(s, a)$ at step $h$ before episode $t$.

We start by initializing the ensemble of Q-values, the policy Q-values, and values to an optimistic value $\widetilde{Q}_h^{t,j}(s, a) = \overline{Q}_h^1(s, a) = \overline{V}_h^1(s, a) = r_h(s, a) + r_0(H - h)$ for all $(j, h, s, a) \in [J] \times [H] \times \mathcal{S} \times \mathcal{A}$ and $r_0 > 0$ some pseudo-rewards.

At episode $t$ we update the ensemble of Q-values as follows, denoting by $n = n_h^t(s, a)$ the count, $w_{j,n} \sim \mathrm{Beta}(H, n)$ the independent learning rates,

$$\widetilde{Q}_h^{t+1,j}(s, a) = \begin{cases} (1 - w_{j,n})\widetilde{Q}_h^{t,j}(s, a) + w_{j,n}\mathring{Q}_h^{t,j}(s, a), & (s, a) = (s_h^t, a_h^t) \\ \widetilde{Q}_h^{t,j}(s, a) & \text{otherwise,} \end{cases}$$

where we defined the target $\mathring{Q}_h^{t,j}(s, a)$ as a mixture between the usual target and some prior target with mixture coefficient $\mathring{w}_{n,j} \sim \mathrm{Beta}(n, n_0)$ and $n_0$ the number of prior samples,

$$\mathring{Q}_h^{t,j}(s, a) = \mathring{w}_{j,n}[r_h(s, a) + \overline{V}_{h+1}^t(s_{h+1}^t)] + (1 - \mathring{w}_{j,n})[r_h(s, a) + r_0(H - h - 1)].$$

It is important to note that in our approach, we need to re-inject prior targets to avoid forgetting their effects too quickly due to the aggressive learning rate. Indeed, the exponential decay of the prior effect can hurt exploration. We observe that the ensemble Q-value only averages uniformly over the last $1/H$ fraction of the targets, as the expected value of the learning rate is $\mathbb{E}[w_{j,n}] = H/(n + H)$. Since $\mathbb{E}[1 - \mathring{w}_{j,n}] = n_0(n + n_0)$ the weight put on the prior sample in expectation, when we unfold

the definition of $\widetilde{Q}_h^{t+1,j}$, is of order $H/n \cdot n/H \cdot n_0/(n+n_0) = n_0/(n+n_0)$, which is consistent with the usual prior forgetting in Bayesian learning. In `Staged-RandQL`, we avoid forgetting the prior too quickly by resetting the temporary Q-value to a prior value at the beginning of each stage.

The policy Q-values are obtained by taking the maximum among the ensemble of Q-values

$$\overline{Q}_h^{t+1}(s,a) = \max_{j \in [J]} \widetilde{Q}_h^{t+1,j}(s,a).$$

The policy is then greedy with respect to the policy Q-values $\pi_h^{t+1}(s) \in \arg\max_{a \in \mathcal{A}} \overline{Q}_h^{t+1}(s,a)$ and the value is $\overline{V}_h^{t+1}(s) = \max_{a \in \mathcal{A}} \overline{Q}_h^{t+1}(s,a)$. The complete `RandQL` procedure is detailed in Algorithm 2.

---

**Algorithm 2** `RandQL`

---

1: **Input:** $J$ ensemble size, number of prior transitions $n_0$, prior reward $r_0$.
2: **Initialize:** $\overline{V}_h(s) = \overline{Q}_h(s,a) = \widetilde{Q}_h^j(s,a) = r(s,a) + r_0(H-h)$, initialize counters $n_h(s,a) = 0$ for $h, s, a \in [H] \times \mathcal{S} \times \mathcal{A}$.
3: **for** $t \in [T]$ **do**
4:     **for** $h \in [H]$ **do**
5:         Play $a_h \in \arg\max_a \overline{Q}_h(s_h, a)$.
6:         Observe reward and next state $s_{h+1} \sim p_h(s_h, a_h)$.
7:         Sample $\mathring{w}_j \sim \text{Beta}(n, n_0)$ for $n = n_h(s_h, a_h)$.
8:         Build targets for all $j \in [J]$

$$\mathring{Q}_h^j = \mathring{w}_j[r_h(s_h, a_h) + \overline{V}_{h+1}(s_{h+1})] + (1 - \mathring{w}_j)[r_h(s_h, a_h) + r_0(H-h)].$$

9:         Sample learning rates $w_j \sim \text{Beta}(H, n)$.
10:        Update ensemble Q-functions for all $j \in [J]$

$$\widetilde{Q}_h^j(s_h, a_h) := (1 - w_j)\widetilde{Q}_h^j(s_h, a_h) + w_j \mathring{Q}_h^j.$$

11:        Update policy Q-function $\overline{Q}_h(s_h, a_h) := \max_{j \in [J]} \widetilde{Q}_h^j(s_h, a_h)$.
12:        Update value function $\overline{V}_h(s_h) := \max_{a \in \mathcal{A}} \overline{Q}_h(s_h, a)$.
13:     **end for**
14: **end for**

---

## B.2 `Sampled-RandQL` algorithm

To create an algorithm that is more similar to `PSRL`, it is possible to select a Q-value at random from the ensemble of Q-values, rather than using the maximum Q-value

$$\overline{Q}_h^t(s,a) = \widetilde{Q}_h^{t,j_t}(s,a) \qquad \text{with } j_t \sim \text{Unif}[J].$$

In this case we also need to update each Q-value in the ensemble with its corresponding target, see Osband and Van Roy [2015],

$$\mathring{Q}_h^{t,j}(s,a) = \mathring{w}_{j,n}[r_h(s,a) + \widetilde{V}_{h+1}^{t,j}(s_{h+1}^t)] + (1 - \mathring{w}_{j,n})[r_h(s,a) + r_0(H-h-1)]$$

where $\widetilde{V}_h^{t,j}(s) = \max_{a \in \mathcal{A}} \widetilde{Q}_h^{t,j}(s,a)$. We name this new procedure `Sampled-RandQL` and detail it in Algorithm 3.

**Algorithm 3** `Sampled-RandQL`

---

1: **Input:** $J$ ensemble size, number of prior transitions $n_0$, prior reward $r_0$.
2: **Initialize:** $\overline{V}_h(s) = \overline{Q}_h(s,a) = \widetilde{Q}_h^j(s,a) = r(s,a) + r_0(H-h)$, initialize counters $n_h(s,a) = 0$ for $h, s, a \in [H] \times \mathcal{S} \times \mathcal{A}$.
3: **for** $t \in [T]$ **do**
4:     Sample ensemble index $i \sim \mathrm{Unif}[J]$
5:     **for** $h \in [H]$ **do**
6:         Play $a_h \in \arg\max_a \overline{Q}_h(s_h, a)$.
7:         Observe reward and next state $s_{h+1} \sim p_h(s_h, a_h)$.
8:         Sample $\mathring{w}_j \sim \mathrm{Beta}(n, n_0)$ for $n = n_h(s_h, a_h)$.
9:         Build targets for all $j \in [J]$

$$\mathring{Q}_h^j = \mathring{w}_j[r_h(s_h, a_h) + \widetilde{V}_{h+1}^j(s_{h+1})] + (1 - \mathring{w}_j)[r_h(s_h, a_h) + r_0(H-h)].$$

10:       Sample learning rates $w_j \sim \mathrm{Beta}(H, n)$.
11:       Update ensemble $Q$-functions for all $j \in [J]$

$$\widetilde{Q}_h^j(s_h, a_h) := (1 - w_j)\widetilde{Q}_h^j(s_h, a_h) + w_j \mathring{Q}_h^j.$$

12:       Update value function $\overline{V}_h(s_h) := \max_{a \in \mathcal{A}} \widetilde{Q}_h^j(s_h, a)$ for all $j \in [J]$.
13:       Update policy $Q$-function $\overline{Q}_h(s_h, a_h) := \widetilde{Q}_h^i(s_h, a_h)$.
14:     **end for**
15: **end for**

---

## C   Weight Distribution in `RandQL`

In this section we study the joint distribution of weights over all targets in `RandQL` algorithm, described in details in Appendix B. To do it, we describe a very useful distribution, defined by Wong [1998].

**Definition 5.** We say that a random vector $(X_1, \ldots, X_n, X_{n+1})$ has a *generalized Dirichlet distribution* $\mathrm{GDir}(\alpha_1, \ldots, \alpha_n; \beta_1, \ldots, \beta_n)$ if $X_{n+1} = 1 - (X_1 + \ldots + X_n)$ and $(X_1, \ldots, X_n)$ it has the following density over the simplex $\{x_1, \ldots, x_n : x_1 + \ldots + x_n \le 1\}$,

$$p(x) = \prod_{i=1}^n \frac{1}{B(\alpha_i, \beta_i)} x_i^{\alpha_i - 1} (1 - x_1 - \ldots - x_i)^{\gamma_i}$$

for $x_1 + \ldots + x_n \le 1, x_j \ge 0$ for $j = 1, \ldots, n$, and $\gamma_j = \beta_j - \alpha_{j+1} - \beta_{j+1}$ for $j = 1, \ldots, n-1$ and $\gamma_n = \beta_n - 1$. If we set $x_{n+1} = 1 - (x_1 + \ldots + x_n)$ then we obtain a homogeneous formula

$$p(x) = \prod_{i=1}^n \frac{1}{B(\alpha_i, \beta_i)} x_i^{\alpha_i - 1} \left( \sum_{j=i+1}^{n+1} x_j \right)^{\gamma_i}$$

Alternative characterization of generalized Dirichlet distribution could be given using independent beta-distributed random variables $Z_1, \ldots, Z_n$ with $Z_i \sim \mathrm{Beta}(\alpha_i, \beta_i)$ as follows

$$X_1 = Z_1,$$

$$X_j = Z_j(1 - X_1 - \ldots - X_{j-1}) = Z_j \prod_{i=1}^{j-1}(1 - Z_i) \quad \text{for } j = 2, 3, \ldots, n$$

$$X_{n+1} = 1 - X_1 - \ldots - X_n = \prod_{i=1}^n (1 - Z_i)$$

Therefore, for `RandQL` algorithm without prior re-injection we have the following formula

$$\widetilde{Q}_h^{t,j}(s,a) = \sum_{i=0}^{n_h^t(s,a)} W_{j,n}^i \left( r_h(s_h^{\ell^i}, a_h^{\ell^i}) + \overline{V}_{h+1}^{\ell^i}(s_{h+1}^{\ell^i}) \right),$$

for $n = n_h^t(s, a)$ and weights are defined as follows

$$W_{j,n}^0 = \prod_{q=0}^{n-1}(1 - w_{j,q}), \quad W_{j,n}^i = w_{j,i-1} \cdot \prod_{q=i}^{n-1}(1 - w_{j,q}), \ i \geq 1.$$

And, moreover, we have that this vector of weights has the *generalized Dirichlet distribution*

$$(W_{n,j}^n, W_{n,j}^{n-1}, \ldots, W_{n,j}^1, W_{n,j}^0) \sim \mathrm{GDir}(H, H, \ldots, H; n + n_0, \ldots, n_0 + 1, n_0).$$

That is, weights generated by the RandQL procedure is an inverted generalized Dirichlet random vector, that induces additional similarities with a usual posterior sampling approaches. Notably, that for $H = 1$ we recover exactly usual Dirichlet distribution, as in the setting of Staged-RandQL.

In the setting of the analysis, the main feature of this distribution is asymmetry in attitude to the order of components. In particular, the expectation of the prior weight $W_{n,j}^0$ is $\prod_{i=1}^n \left(1 - \frac{H}{i+H}\right) \sim n^{-H}$ that leads to too rapid forgetting of the prior information.

# D  Proofs for Tabular algorithm

## D.1  Algorithm

In this section we describe in detail the tabular algorithms and the ways we will analyze them. We also provide some notations that will be used in the sequel.

Let $n_h^t(s,a)$ be the number of visits of $(s,a,h)$ (i.e., of the state-action pair $(s,a)$ at step $h$) at the beginning of episode $t$: $n_h^t(s,a) = \sum_{i=1}^{t-1} \mathbb{1}\{(s_h^i, a_h^i) = (s,a)\}$. In particular, $n_h^{T+1}(s,a)$ is the number of visits of $(s,a,h)$ after all episodes.

Let $e_k = \lfloor (1 + 1/H)^k \cdot H \rceil$ be the length of each stage for any $k \geq 0$ and, by convention, $e_{-1} = 0$. We will say that at the beginning of episode $t$ a triple $(s,a,h)$ is in $k$-th stage if $n_h^t(s,a) \in [\sum_{i=0}^{k-1} e_i, \sum_{i=0}^{k} e_i)$.

Let $\widetilde{n}_h^t(s,a)$ be the number of visits of state-action pair during the current stage at the beginning of episode $t$. Formally, it holds $\widetilde{n}_h^t(s,a) = n_h^t(s,a) - \sum_{i=0}^{k-1} e_i$, where $k$ is the index of current stage.

Let $\kappa > 0$ be the posterior inflation coefficient, $n_0$ be the number of prior transitions, and $J$ be the number of temporary $Q$-functions. Let $\widetilde{Q}_h^{t,j}$ be the $j$-th *temporary* Q-function and $\overline{Q}_h^t$ be the *policy* Q-function at the beginning of episode $t$. We initialize them as follows

$$\overline{Q}_h^1(s,a) = r_h(s,a) + r_0(H - h - 1), \quad \widetilde{Q}_h^{1,j}(s,a) = r_h(s,a) + r_0(H - h - 1),$$

We can treat this initialization as a setting prior over $n_0$ pseudo-transitions to artificial state $s_0$ with $r_0 > 1$ reward for each interaction.

For each transition we perform the following update of temporary Q-functions

$$\widetilde{Q}_h^{t+1/2,j}(s,a) = \begin{cases} (1 - w_{j,\widetilde{n}}^k) \cdot \widetilde{Q}_h^{t,j}(s,a) + w_{j,\widetilde{n}}^k[r_h(s,a) + \overline{V}_{h+1}^t(s_{h+1}^t)], & (s,a) = (s_h^t, a_h^t) \\ \widetilde{Q}_h^{t,j}(s,a) & \text{otherwise}, \end{cases} \tag{2}$$

where $\widetilde{n} = \widetilde{n}_h^t(s,a)$ is the number of visits of $(s,a,h)$ during the current stage at the beginning of episode $t$, $k$ is the index of the current stage, and $w_{j,\widetilde{n}}^k$ is a sequence of independent beta-distribution random variables $w_{j,\widetilde{n}}^k \sim \text{Beta}(1/\kappa, (\widetilde{n} + n_0)/\kappa)$. Here we slightly abuse the notation by dropping the dependence of weights $w_{j,\widetilde{n}}^k$ on the triple $(h,s,a)$ in order to simplify the exposition. In the case that the explicit dependence is required, we will call these weights as $w_{j,\widetilde{n}}^{k,h}(s,a)$.

Next we define the stage update as follows

$$\overline{Q}_h^{t+1}(s,a) = \begin{cases} \max_{j \in [J]} \widetilde{Q}_h^{t+1/2,j}(s,a) & \widetilde{n}_h^t(s,a) = \lfloor (1 + 1/H)^k H \rfloor \\ \overline{Q}_h^t(s,a) & \text{otherwise} \end{cases}$$

$$\widetilde{Q}_h^{t+1,j}(s,a) = \begin{cases} r_h(s,a) + r_0(H - h + 1) & \widetilde{n}_h^t(s,a) = \lfloor (1 + 1/H)^k H \rfloor \\ \widetilde{Q}_h^{t+1/2,j}(s,a) & \text{otherwise} \end{cases}$$

$$\overline{V}_h^{t+1}(s) = \max_{a \in \mathcal{A}} \overline{Q}_h^{t+1}(s,a)$$

$$\pi_h^{t+1}(s) \in \arg\max_{a \in \mathcal{A}} \overline{Q}_h^{t+1}(s,a),$$

where $k$ is the current stage. In other words, we update $\overline{Q}^{t+1}$ with temporary values of $\widetilde{Q}^{t+1/2,j}$, and then, if the change of stage is triggered, reinitialize $\widetilde{Q}_h^{t+1,j}(s,a)$ for all $j$. For episode $t$ we will call $k_h^t(s,a)$ the index of stage where $\overline{Q}_h^t(s,a)$ was updated (and $k_h^t(s,a) = -1$ if there was no update). For all $t$ we define $\tau_h^t(s,a) \leq t$ as an episode when the stage update happens. In other words, for any $t$ the following holds

$$\overline{Q}_h^{t+1}(s,a) = \max_{j \in [J]} \widetilde{Q}_h^{\tau_h^t(s,a)+1/2,j}(s,a),$$

where $\tau_h^t(s,a) = 0$ and $e_k = 0$ if there was no updates. To simplify the notation we will omit dependence on $(s,a,h)$ where it is deducible from the context.

To simplify the notation, we can extend the state space $\mathcal{S}$ by an additional state $s_0$ that will be purely technical and used in the proofs. This state has the prescribed value function $V_h^\star(s_0) = r_0(H - h)$ and could be treated as a absorbing pseudo-state with reward $r_0$.

We notice that in this case we use $e_k$ samples to compute $\widetilde{Q}^{\tau_h^t(s,a)+1/2,j}$ for $k = k_h^t(s, a)$. For this $k$ we define $\ell_{k,h}^i(s, a)$ as a time of $i$-th visit of state-action pair $(s, a)$ during $k$-th stage. Then we have the following decomposition

$$\widetilde{Q}_h^{\tau^t+1/2,j}(s,a) = r_h(s,a) + \sum_{i=0}^{e_k} W_{j,e_k,k}^i \overline{V}_{h+1}^{\ell^i}(s_{h+1}^{\ell^i}), \tag{3}$$

where we drop dependence on $k$ and $(s, a, h)$ in $\ell^i$ to simplify notations, and use the convention $s_{h+1}^{\ell_{k,h}^0(s,a)} = s_0$, and the following aggregated weights

$$W_{j,n,k}^0 = \prod_{q=0}^{n-1}(1 - w_{j,q}^k), \quad W_{j,n,k}^i = w_{j,i-1}^k \cdot \prod_{q=i}^{n-1}(1 - w_{j,q}^k), \; i \geq 1.$$

We will omit the dependent on the stage index $k$ when it is not needed for the statement. However, we notice that these vectors, for different stage $k$, will be independent.

By the properties of generalized Dirichlet distribution it is possible to show the following result

**Lemma 3.** *For any fixed $n > 0$, the random vector $(W_{j,n}^0, W_{j,n}^1, \ldots, W_{j,n}^n)$ has a Dirichlet distribution* $\mathrm{Dir}(n_0/\kappa, 1/\kappa, \ldots, 1/\kappa)$.

*Proof.* Using the Dirichlet random variate generation from marginal beta distributions, it is sufficient to prove that for all $i \in \{0, \ldots, n\}$, $W_{j,n,k}^{n-i} = (1 - W_{j,n,k}^n - \cdots - W_{j,n,k}^{n-i+1})w_{j,n-i-1}^k$, with the convention $w_{j,-1}^k = 1$. This is trivial for $i = 0$, as $W_{j,n,k}^n = w_{j,n-1}^k$. Now, if this is true for some $i$, then, for $i + 1 \in \{0, \ldots, n\}$, we have

$$
\begin{aligned}
W_{j,n,k}^{n-i-1} &= w_{j,n-i-2}^k \prod_{q=n-i-1}^{n-1}(1 - w_{j,q}^k) \\
&= w_{j,n-i-2}^k(1 - w_{j,n-i-1}^k)(1 - W_{j,n,k}^n - \cdots - W_{j,n,k}^{n-i+1}) \\
&= w_{j,n-i-2}^k(1 - W_{j,n,k}^n - \cdots - W_{j,n,k}^{n-i+1} - \underbrace{w_{j,n-i-1}^k(1 - W_{j,n,k}^n - \cdots - W_{j,n,k}^{n-i+1})}_{=W_{j,n,k}^{n-i}}),
\end{aligned}
$$

which finishes the proof. $\qquad\qquad\qquad\qquad\qquad\qquad\qquad\qquad\qquad\qquad\qquad\qquad\qquad\qquad\square$

Notably, the expression (3) shows a significant similarity between our method and OPSRL. It is the reason why we can call this method a model-free posterior sampling, where posterior sampling is performed over the model in a lazy and model-free fashion.

## D.2 Concentration

Let $\beta^\star \colon (0, 1) \times \mathbb{N} \to \mathbb{R}_+$ and $\beta^B, \beta^{\mathrm{conc}}, \beta \colon (0, 1) \to \mathbb{R}_+$ be some function defined later on in Lemma 4. We define the following favorable events

$$\mathcal{E}^{\star}(\delta) \triangleq \left\{ \forall t \in \mathbb{N}, \forall h \in [H], \forall (s,a) \in \mathcal{S} \times \mathcal{A}, k = k_h^t(s,a) : \right.$$

$$\left. \mathcal{K}_{\inf}\left( \frac{1}{e_k} \sum_{i=1}^{e_k} \delta_{V_{h+1}^{\star}(s_{h+1}^{\ell^i})}, p_h V_{h+1}^{\star}(s,a) \right) \leq \frac{\beta^{\star}(\delta, e_k)}{e_k} \right\},$$

$$\mathcal{E}^{B}(\delta) \triangleq \left\{ \forall t \in [T], \forall h \in [H], \forall (s,a) \in \mathcal{S} \times \mathcal{A}, \forall j \in [J], k = k_h^t(s,a) : \right.$$

$$\left| \sum_{i=0}^{e_k} (W_{j,e_k,k}^i - \mathbb{E}[W_{j,e_k,k}^i]) \overline{V}_{h+1}^{\ell^i}(s_{h+1}^{\ell^i}) \right| \leq 60\mathrm{e}^2 \sqrt{\frac{r_0^2 H^2 \kappa \beta^B(\delta)}{e_k + n_0 \kappa}}$$

$$\left. + 1200\mathrm{e} \frac{r_0 H \kappa \log(e_k + n_0 \kappa)(\beta^B(\delta))^2}{e_k + n_0 \kappa} \right\},$$

$$\mathcal{E}^{\mathrm{conc}}(\delta) \triangleq \left\{ \forall t \in [T], \forall h \in [H], \forall (s,a) \in \mathcal{S} \times \mathcal{A}, k = k_h^t(s,a) : \right.$$

$$\left| \frac{1}{e_k} \sum_{i=1}^{e_k} V_{h+1}^{\star}(s_{h+1}^{\ell_{k,h}^i(s,a)}) - p_h V_{h+1}^{\star}(s,a) \right| \leq \sqrt{\frac{2 r_0^2 H^2 \beta^{\mathrm{conc}}(\delta)}{e_k}} \right\}$$

$$\mathcal{E}(\delta) \triangleq \left\{ \sum_{t=1}^{T} \sum_{h=1}^{H} (1 + 1/H)^{H-h} \big| p_h [V_{h+1}^{\star} - V_{h+1}^{\pi_t}](s_h^t, a_h^t) - [V_{h+1}^{\star} - V_{h+1}^{\pi_t}](s_{h+1}^t) \big| \right.$$

$$\left. \leq 2\mathrm{e} r_0 H \sqrt{2HT\beta(\delta)}. \right\}.$$

We also introduce the intersection of these events, $\mathcal{G}(\delta) \triangleq \mathcal{E}^{\star}(\delta) \cap \mathcal{E}^{B}(\delta) \cap \mathcal{E}^{\mathrm{conc}}(\delta) \cap \mathcal{E}(\delta)$. We prove that for the right choice of the functions $\beta^{\star}, \beta^{\mathrm{KL}}, \beta^{\mathrm{conc}}, \beta, \beta^{\mathrm{Var}}$ the above events hold with high probability.

**Lemma 4.** *For any $\delta \in (0,1)$ and for the following choices of functions $\beta$,*

$$\beta^{\star}(\delta, n) \triangleq \log(8SAH/\delta) + 3\log(\mathrm{e}\pi(2n+1)),$$
$$\beta^{B}(\delta) \triangleq \log(8SAH/\delta) + \log(TJ),$$
$$\beta^{\mathrm{conc}}(\delta) \triangleq \log(8SAH/\delta) + \log(2T),$$
$$\beta(\delta) \triangleq \log(16/\delta),$$

*it holds that*

$$\mathbb{P}[\mathcal{E}^{\star}(\delta)] \geq 1 - \delta/8, \qquad \mathbb{P}[\mathcal{E}^{B}(\delta)] \geq 1 - \delta/8,$$
$$\mathbb{P}[\mathcal{E}^{\mathrm{conc}}(\delta)] \geq 1 - \delta/8, \qquad \mathbb{P}[\mathcal{E}(\delta)] \geq 1 - \delta/8.$$

*In particular, $\mathbb{P}[\mathcal{G}(\delta)] \geq 1 - \delta/2$.*

*Proof.* From the fact that $s_{h+1}^{\ell^i}$ are i.i.d. generated from $p_h(s,a)$, Theorem 4, and union bound $\mathcal{S} \times \mathcal{A} \times [H]$ it holds $\mathbb{P}[\mathcal{E}^{\star}(\delta)] \geq 1 - \delta/8$.

Next we fix all $t, h, s, a, j$, and denote $n = e_{k_h^t(s,a)}$. First, we define a filtration of $\sigma$-algebras $\mathcal{F}_\tau$ that is sigma-algebra generated by all random variables appeared untill the update (2) in the episode $t$ and step $h$, before newly generated random weights but after receiving new state $s_{h+1}^t$. Formally, we can define it as follows

$$\mathcal{F}_{t,h} = \sigma\left( \left\{ (s_{h'}^\tau, a_{h'}^\tau, w_{j,\widetilde{n}_{h'}^\tau}^{k_{h'}^\tau + 1, h'}(s_{h'}^\tau, a_{h'}^\tau)), \forall \tau < t, (h', j) \in [H] \times [J] \right\} \right.$$

$$\left. \cup \{ (s_{h'}^t, a_{h'}^t, s_{h'+1}^t), \forall h' \leq h \} \cup \{ w_{j,\widetilde{n}_{h'}^t}^{k_{h'}^t + 1, h'}(s_{h'}^t, a_{h'}^t), \forall h' < h, j \in [J] \} \right),$$

where we drop dependence on state-action pairs everywhere where it is deducible from the context.

Consider a sequence $\ell^1 < \ldots < \ell^n$ be an excursion of the state-action pair $(s,a)$ at the step $h$. Each $\ell^i$ is a stopping time w.r.t $\mathcal{F}_{t,h}$, so we can consider a stopped filtration (with a shift by 1 in indices) $\widetilde{\mathcal{F}}_{i-1} = \mathcal{F}_{\ell^i,h}$. In other words, this filtration at time-stamp $i-1$ contains all the information that is available just before generation of random weights for $i$-th update of temporary Q-functions inside the last stage. We notice that under this definition we have

$$\mathbb{E}[\overline{V}^{\ell_i}_{h+1}(s^{\ell_i}_{h+1})|\widetilde{\mathcal{F}}_{i-1}] = \overline{V}^{\ell_i}_{h+1}(s^{\ell_i}_{h+1}),$$

$$\mathbb{E}[W^i_{j,n,k}|\widetilde{\mathcal{F}}_{i-1}] = \mathbb{E}\left[w^k_{j,i-1}\prod_{\ell=i}^{n-1}(1-w^k_{j,\ell})|\mathcal{F}_{i-1}\right] = \mathbb{E}[W^i_{j,n,k}],$$

Next, we notice that the joint vector of weights follows the Dirichlet distribution, applying aggregation property and extending the filtration backward by adding fake transitions we can extend sum to $n + n_0$ summands defining $s^{\ell^{-i}}_{h+1} = s_0$

$$\sum_{i=0}^{n}\bigl(W^i_{j,n,k} - \mathbb{E}[W^i_{j,n,k}]\bigr)\overline{V}^{\ell^i}_{h+1}(s^{\ell^i}_{h+1}) = \sum_{q=-n_0+1}^{n}\left(\widetilde{W}_q - \mathbb{E}[\widetilde{W}_q]\right)\overline{V}^{\ell^q}_{h+1}(s^{\ell^q}_{h+1}).$$

Finally, we notice that marginals of Dirichlet random vector follow Beta distribution, therefore by Proposition 7 and union bound we conclude $\mathbb{P}[\mathcal{E}^B(\delta)] \geq 1 - \delta/8$.

To show that $\mathbb{P}(\mathcal{E}^{\mathrm{conc}}(\delta)) > 1 - \delta/8$, it is enough to apply Hoeffding inequality for a fixed number of samples $e_k$ used in empirical mean, and then use union bound of all possible values of $(s,a,h) \in \mathcal{S} \times \mathcal{A} \times [H]$ and $e_k \in [T]$.

Next, define the following sequence

$$Z_{t,h} \triangleq (1 + 1/H)^{H-h}\Bigl([V^\star_{h+1} - V^{\pi^t}_{h+1}](s^t_{h+1}) - p_h[V^\star_{h+1} - V^{\pi^t}_{h+1}](s^t_h, a^t_h)\Bigr), \quad t \in [T], h \in [H],$$

It is easy to see that these sequences form a martingale-difference w.r.t filtration $\mathcal{F}_{t,h} = \sigma\{\{(s^\ell_{h'}, a^\ell_{h'}, \pi^\ell), \ell < t, h' \in [H]\} \cup \{(s^t_{h'}, a^t_{h'}, \pi^t), h' \leq h\}\}$. Moreover, $|Z_{t,h}| \leq 2er_0H$ for all $t \in [T]$ and $h \in [H]$. Hence, the Azuma-Hoeffding inequality implies

$$\mathbb{P}\Bigl(\Bigl|\sum_{t=1}^{T}\sum_{h=1}^{H}Z_{t,h}\Bigr| > 2er_0H\sqrt{2tH \cdot \beta(\delta)}\Bigr) \leq 2\exp(-\beta(\delta)) = \delta/8,$$

therefore $\mathbb{P}[\mathcal{E}(\delta)] \geq 1 - \delta/8$. $\qquad\qquad\square$

### D.3 Optimism

In this section we prove that our estimate of $Q$-function $\overline{Q}^t_h(s,a)$ is optimistic, that is the event

$$\mathcal{E}_{\mathrm{opt}} \triangleq \Bigl\{\forall t \in [T], h \in [H], (s,a) \in \mathcal{S} \times \mathcal{A} : \overline{Q}^t_h(s,a) \geq Q^\star_h(s,a)\Bigr\}. \tag{4}$$

holds with high probability on the event $\mathcal{E}^\star(\delta)$.

Define constants

$$c_0 \triangleq \frac{8}{\pi}\left(\frac{4}{\sqrt{\log(17/16)}} + 8 + \frac{49 \cdot 4\sqrt{6}}{9}\right)^2 + 1. \tag{5}$$

and

$$c_J \triangleq \frac{1}{\log\left(\frac{2}{1+\Phi(1)}\right)}, \tag{6}$$

where $\Phi(\cdot)$ is a CDF of a normal distribution.

**Proposition 1.** Assume that $J = \lceil c_J \cdot \log(2SAHT/\delta) \rceil$, $\kappa = 2\beta^\star(\delta, T)$, $r_0 = 2$, and $n_0 = \lceil (c_0 + 1 + \log_{17/16}(T)) \cdot \kappa \rceil$. Then conditionally on $\mathcal{E}^\star(\delta)$ the event

$$\mathcal{E}_{\text{anticonc}} \triangleq \Bigg\{ \forall t \in [T], \ \forall h \in [H], \ \forall (s,a) \in \mathcal{S} \times \mathcal{A} :$$

$$\max_{j \in [J]} \Bigg\{ \sum_{i=0}^{e_k} W_{j,e_k,k}^i V_{h+1}^\star(s_{h+1}^{\ell_{t,h}^i(s,a)}) \Bigg\} \geq p_h V_{h+1}^\star(s,a), k = k_h^t(s,a) \Bigg\}$$

holds with probability at least $1 - \delta/2$.

*Proof.* Let us fix $t \in [T], h \in [H], (s,a) \in \mathcal{S} \times \mathcal{A}$, and $j \in [J]$. By Lemma 3, we have that the vector $(W_{j,e_k,k}^i)_{i=0,\ldots,e_k}$ has Dirichlet distribution. Note that $V_{h+1}^\star(s_{h+1}^{\ell^0}) = r_0(H - h - 1)$ is an upper bound on $V$-function and the weight of the first atom is $\alpha_0 \triangleq n_0/\kappa \geq c_0 + \log_{17/16}(T)$ for $c_0$ defined in (5). Define a measure $\bar{\nu}_{e_k} = \frac{n_0 - 1}{e_k + n_0 - 1} \delta_{V_{h+1}^\star(s_0)} + \sum_{i=1}^{e_k} \frac{1}{e_k + n_0 - 1} \delta_{V_{h+1}^\star(s_{h+1}^{\ell^i})}$. Since $p_h V_{h+1}^\star(s,a) \leq H - h - 1$, we can apply Lemma 10 with a fixed $\varepsilon = 1/2$ conditioned on independent samples $\{s_{h+1}^{\ell_i}\}_{i=1}^{e_k}$ from $p_h(s,a)$

$$\mathbb{P}\Bigg[ \sum_{i=0}^{e_k} W_{j,e_k,k}^i V_{h+1}^\star(s_{h+1}^{\ell_{t,h}^i(s,a)}) \geq p_h V_{h+1}^\star(s,a) \mid \{s_{h+1}^{\ell_i}\}_{i=1}^{e_k} \Bigg]$$

$$\geq \frac{1}{2}\Bigg( 1 - \Phi\Bigg( \sqrt{\frac{2(e_k + n_0 - \kappa)\mathcal{K}_{\inf}(\bar{\nu}_{e_k}, p_h V_{h+1}^\star(s,a))}{\kappa}} \Bigg) \Bigg), \tag{7}$$

where $\Phi$ is a CDF of a normal distribution. Combining Lemma 12 and the event $\mathcal{E}^\star(\delta)$

$$(e_k + n_0 - \kappa)\mathcal{K}_{\inf}(\bar{\nu}_{e_k}, p_h V_{h+1}^\star(s,a)) \leq e_k \mathcal{K}_{\inf}(\widehat{\nu}_{e_k}, p_h V_{h+1}^\star(s,a)) \leq \beta^\star(\delta, T),$$

where $\widehat{\nu}_{e_k} = \frac{1}{e_k} \sum_{i=1}^{e_k} \delta_{V_{h+1}^\star(s_{h+1}^{\ell^i})}$, and, as a corollary

$$\mathbb{P}\Bigg[ \sum_{i=0}^{e_k} W_{j,e_k,k}^i V_{h+1}^\star(s_{h+1}^{\ell_{t,h}^i(s,a)}) \geq p_h V_{h+1}^\star(s,a) \mid \mathcal{E}^\star(\delta), \{s_{h+1}^{\ell^i}\}_{i=1}^{e_k} \Bigg] \geq \frac{1}{2}\Bigg( 1 - \Phi\Bigg( \sqrt{\frac{2\beta^\star(\delta, T)}{\kappa}} \Bigg) \Bigg).$$

By taking $\kappa = 2\beta^\star(\delta, T)$ we have a constant probability of being optimistic

$$\mathbb{P}\Bigg( \sum_{i=0}^{e_k} W_{j,e_k,k}^i V_{h+1}^\star(s_{h+1}^{\ell_{t,h}^i(s,a)}) \geq p_h V_{h+1}^\star(s,a) \mid \mathcal{E}^\star(\delta) \Bigg) \geq \frac{1 - \Phi(1)}{2} \triangleq \gamma.$$

Next, using a choice $J = \lceil \log(2SAHT/\delta)/\log(1/(1-\gamma)) \rceil = \lceil c_J \cdot \log(2SAHT/\delta) \rceil$

$$\mathbb{P}\Bigg[ \max_{j \in [J]} \Bigg\{ \sum_{i=0}^{e_k} W_{j,e_k,k}^i V_{h+1}^\star(s_{h+1}^{\ell_{t,h}^i(s,a)}) \Bigg\} \geq p_h V_{h+1}^\star(s,a) \mid \mathcal{E}^\star(\delta) \Bigg] \geq 1 - (1-\gamma)^J \geq 1 - \frac{\delta}{2SAHT}.$$

By a union bound we conclude the statement. $\qquad \square$

Next we provide a connection between $\mathcal{E}^{\text{anticonc}}$ and $\mathcal{E}^{\text{opt}}$.

**Proposition 2.** It holds that $\mathcal{E}^{\text{opt}} \subseteq \mathcal{E}^{\text{anticonc}}$.

*Proof.* We proceed by a backward induction over $h$. Base of induction $h = H + 1$ is trivial. Next by Bellman equations for $\overline{Q}_h^t$ and $Q_h^\star$

$$[\overline{Q}_h^t - Q_h^\star](s,a) = \max_{j \in [J]} \Bigg\{ \sum_{i=0}^n W_{j,n}^i \overline{V}_{h+1}^{\ell^i}(s_{h+1}^{\ell^i}) \Bigg\} - p_h V_{h+1}^\star(s,a),$$

where $n = e_{k_h^t(s,a)}$ and we drop dependence on $k, t, h, s, a$ in $\ell^i$. By induction hypothesis we have $\overline{V}_{h+1}^{\ell^i}(s') \geq \overline{Q}_{h+1}^{\ell^i}(s', \pi^\star(s')) \geq Q_{h+1}^\star(s', \pi^\star(s')) = V_{h+1}^\star(s')$ for any $i$, thus

$$[\overline{Q}_h^t - Q_h^\star](s,a) \geq \max_{j \in [J]} \Bigg\{ \sum_{i=0}^n W_{j,n}^i V_{h+1}^\star(s_{h+1}^{\ell^i}) \Bigg\} - p_h V_{h+1}^\star(s,a).$$

By the definition of event $\mathcal{E}^{\text{anticonc}}(\delta)$ we conclude the statement. $\qquad \square$

**Proposition 3** (Optimism). *Assume that* $J = \lceil c_J \cdot \log(2SAHT/\delta) \rceil$, $\kappa = 2\beta^\star(\delta, T)$, $r_0 = 2$, *and* $n_0 = \lceil (c_0 + 1 + \log_{17/16}(T)) \cdot \kappa \rceil$, *where* $c_0$ *is defined in* (5) *and* $c_J$ *is defined in* (6). *Then* $\mathbb{P}(\mathcal{E}^{\mathrm{opt}} \mid \mathcal{E}^\star(\delta)) \geq 1 - \delta/2$.

## D.4 Regret Bound

Let us define the main event $\mathcal{G}'(\delta) = \mathcal{G}(\delta) \cap \mathcal{E}^{\mathrm{opt}}$. On this event we have the following corollary that connects `RandQL` with `OptQL` with Hoeffding bonuses.

Define the following quantity

$$\beta^{\max}(\delta) = \max\{\kappa, n_0/\kappa, \beta^B(\delta), \beta^{\mathrm{conc}}(\delta), \beta(\delta), \log(T + n_0)\} = \mathcal{O}(\log(SATH/\delta)).$$

**Corollary 1.** *Assume conditions of Proposition 3 hold. Let* $t \in [T], h \in [H], (s, a) \in \mathcal{S} \times \mathcal{A}$. *Define* $k = k_h^t(s, a)$ *and let* $\ell^1 < \ldots < \ell^{e_k}$ *be a excursions of* $(s, a, h)$ *until the previous stage. Then on the event* $\mathcal{G}'(\delta)$ *the following bound holds for* $k \geq 0$

$$0 \leq \overline{Q}_h^t(s, a) - Q_h^\star(s, a) \leq \frac{1}{n} \sum_{i=1}^n [\overline{V}_{h+1}^{\ell^i}(s_{h+1}^{\ell^i}) - V_{h+1}^\star(s_{h+1}^{\ell^i})] + \mathcal{B}_h^t(k),$$

*where*

$$\mathcal{B}_h^t(k) = 61\mathrm{e}^2 \frac{r_0 H(\beta^{\max}(\delta))}{\sqrt{e_k}} + 1201\mathrm{e}\frac{r_0 H(\beta^{\max}(\delta))^4}{e_k}.$$

*Proof.* The lower bound follows from the definition of the event $\mathcal{E}^{\mathrm{opt}}$. For the upper bound we first apply the decomposition for $\overline{Q}_h^t(s, a)$ and the definition of event $\mathcal{E}^B(\delta)$ from Lemma 4

$$\overline{Q}_h^t(s, a) = r_h(s, a) + \max_{j \in [J]} \left\{ \sum_{i=0}^{e_k} W_{j,e_k}^i \overline{V}_{h+1}^{\ell^i}(s_{h+1}^{\ell^i}) \right\}$$

$$\leq r_h(s, a) + \frac{1}{e_k + n_0} \sum_{i=1}^{e_k} \overline{V}_{h+1}^{\ell^i}(s_{h+1}^{\ell^i}) + \frac{n_0 \kappa \cdot r_0 H}{e_k + n_0} + 60\mathrm{e}^2 \sqrt{\frac{r_0^2 H^2 \kappa \beta^B(\delta)}{e_k + n_0}}$$

$$+ 1200\mathrm{e}\frac{r_0 H\kappa \log(e_k + n_0)(\beta^B(\delta))^2}{e_k + n_0}.$$

Then, by Bellman equations,

$$\overline{Q}_h^t(s, a) - Q_h^\star(s, a) \leq \frac{1}{e_k} \sum_{i=1}^{e_k} \left[ \overline{V}_{h+1}^{\ell^i} - V_{h+1}^\star \right](s_{h+1}^{\ell^i}) + \frac{1}{e_k} \sum_{i=1}^{e_k} \left[ V_{h+1}^\star(s_{h+1}^{\ell^i}) - p_h V_{h+1}^\star(s, a) \right]$$

$$+ (1200\mathrm{e} + 1)\frac{r_0 H(\beta^{\max}(\delta))^4}{e_k + n_0} + 60\mathrm{e}^2 \cdot \frac{r_0 H\beta^{\max}(\delta)}{\sqrt{e_k + n_0}}$$

By the definition of event $\mathcal{E}^{\mathrm{conc}}(\delta)$ we conclude the statement. $\square$

Let us define $\delta_h^t = \overline{V}_h^t(s_h^t) - V_h^{\pi^t}(s_h^t)$ and $\zeta_h^t = \overline{V}_h^t(s_h^t) - V_h^\star(s_h^t)$.

**Lemma 5.** *Assume conditions of Proposition 3 hold. Then on event* $\mathcal{G}'(\delta) = \mathcal{G}(\delta) \cap \mathcal{E}^{\mathrm{opt}}$, *where* $\mathcal{G}(\delta)$ *is defined in Lemma 4, the following upper bound on regret holds*

$$\mathfrak{R}^T \leq \mathrm{e}H \sum_{t=1}^T \sum_{h=1}^H \mathbb{1}\{k_h^t(s_h^t, a_h^t) = -1\} + \sum_{t=1}^T \sum_{h=1}^H (1 + 1/H)^{H-h}\xi_h^t + \mathrm{e}\sum_{t=1}^T \sum_{h=1}^H \mathcal{B}_h^t,$$

*where* $\xi_h^t = p_h[V_{h+1}^\star - V_{h+1}^{\pi^t}](s_h^t, a_h^t) - [V_{h+1}^\star - V_{h+1}^{\pi^t}](s_{h+1}^t)$ *and* $\mathcal{B}_h^t = \mathcal{B}_h^t(s_h^t, a_h^t) \cdot \mathbb{1}\{k_h^t(s_h^t, a_h^t) \geq 0\}$ *for* $\mathcal{B}_h^t$ *defined in Corollary 1.*

*Proof.* We notice that on the event $\mathcal{E}^{\mathrm{opt}}$ the following upper bound holds

$$\mathfrak{R}^T \leq \sum_{t=1}^T \delta_1^t. \tag{8}$$

Next we analyze $\delta_h^t$. By the choice of $a_h^t = \arg\max_{a \in \mathcal{A}} \overline{Q}_h^t(s_h^t, a)$, Corollary 1, and Bellman equations, we have

$$
\begin{aligned}
\delta_h^t &= \overline{V}_h^t(s_h^t) - V_h^{\pi^t}(s_h^t) = \overline{Q}_h^t(s_h^t, a_h^t) - Q_h^{\pi^t}(s_h^t, a_h^t) \\
&= \overline{Q}_h^t(s_h^t, a_h^t) - Q_h^\star(s_h^t, a_h^t) + Q_h^\star(s_h^t, a_h^t) - Q_h^{\pi^t}(s_h^t, a_h^t) \\
&\leq H\mathbb{1}\{N_h^t = 0\} + \mathbb{1}\{N_h^t > 0\}\left( \frac{1}{N_h^t} \sum_{i=1}^{N_h^t} \zeta_{h+1}^{\ell_{t,h}^i} + \mathcal{B}_h^t(s_h^t, a_h^t) + p_h[V_{h+1}^\star - V_{h+1}^{\pi^t}](s_h^t, a_h^t) \right).
\end{aligned}
$$

where $k_h^t = k_h^t(s_h^t, a_h^t)$, $N_h^t = e_{k_h^t}$, $\ell_{t,h}^i$ is episode of the $i$-th visitation of the state-action pair $(s_h^t, a_h^t)$ during the stage $k_h^t$, and additionally by the convention $0/0 = 0$. Let $\xi_h^t = p_h[V_{h+1}^\star - V_{h+1}^{\pi^t}](s_h^t, a_h^t) - [V_{h+1}^\star - V_{h+1}^{\pi^t}](s_{h+1}^t)$ be a martingale-difference sequence, and $\mathcal{B}_h^t = \mathcal{B}_h^t(s_h^t, a_h^t)\mathbb{1}\{N_h^t > 0\}$ then

$$
\delta_h^t \leq H\mathbb{1}\{N_h^t = 0\} + \frac{\mathbb{1}\{N_h^t > 0\}}{N_h^t} \sum_{i=1}^{N_h^t} \zeta_{h+1}^{\ell_{t,h}^i} - \zeta_{h+1}^t + \delta_{h+1}^t + \xi_h^t + \mathcal{B}_h^t.
$$

and, as a result

$$
\begin{aligned}
\sum_{t=1}^T \delta_h^t &\leq H \sum_{t=1}^T \mathbb{1}\{N_h^t = 0\} + \sum_{t=1}^T \frac{\mathbb{1}\{N_h^t > 0\}}{N_h^t} \sum_{i=1}^{N_h^t} \zeta_{h+1}^{\ell_{t,h}^i} \\
&\quad - \sum_{t=1}^T \zeta_{h+1}^t + \sum_{t=1}^T \delta_{h+1}^t + \sum_{t=1}^T \xi_h^t + \sum_{t=1}^T \mathcal{B}_h^t.
\end{aligned}
$$

Next we have to analyze the second term, following the approach by Zhang et al. [2020],

$$
\begin{aligned}
\sum_{t=1}^T \frac{\mathbb{1}\{N_h^t > 0\}}{N_h^t} \sum_{i=1}^{N_h^t} \zeta_{h+1}^{\ell_{t,h}^i} &= \sum_{q=1}^T \sum_{t=1}^T \frac{\mathbb{1}\{N_h^t > 0\}}{N_h^t} \sum_{i=1}^{N_h^t} \zeta_{h+1}^{\ell_{t,h}^i} \mathbb{1}\{\ell_{t,h}^i = q\} \\
&= \sum_{q=1}^T \zeta_{h+1}^q \cdot \sum_{t=1}^T \frac{\mathbb{1}\{k_h^t \geq 0\}}{N_h^t} \sum_{i=1}^{N_h^t} \mathbb{1}\{\ell_{t,h}^i = q\}.
\end{aligned}
$$

Notice that $\sum_{i=1}^{N_h^t} \mathbb{1}\{\ell_{t,h}^i = q\} \leq 1$ since all visitations are increasing in $i$, and, moreover, it turns to equality if and only if $(s_h^q, a_h^q) = (s_h^t, a_h^t)$ and this visitation happens in stage $k_h^t$, where $k_h^t$ is equal to the stage of episode $q$ with respect to $(s_h^q, a_h^q, h)$. Since the sum is over all the next episodes with respect to stage of $q$, we have that the number of non-zero elements in the sum over $t$ is bounded by $(1 + 1/H)N_h^t$. Thus

$$
\sum_{q=1}^T \zeta_{h+1}^q \cdot \sum_{t=1}^T \frac{\mathbb{1}\{k_h^t \geq 0\}}{N_h^t} \sum_{i=1}^{N_h^t} \mathbb{1}\{\ell_{t,h}^i = q\} \leq \left(1 + \frac{1}{H}\right) \sum_{q=1}^T \zeta_{h+1}^q.
$$

After a simple algebraic manipulations and using the fact that $\zeta_h^t \leq \delta_h^t$,

$$
\begin{aligned}
\sum_{t=1}^T \delta_h^t &\leq H \sum_{t=1}^T \mathbb{1}\{N_h^t = 0\} + \sum_{t=1}^T (1 + 1/H)\zeta_{h+1}^t - \sum_{t=1}^T \zeta_{h+1}^t + \sum_{t=1}^T \delta_{h+1}^t + \sum_{t=1}^T \xi_h^t + \sum_{t=1}^T \mathcal{B}_h^t \\
&\leq H \sum_{t=1}^T \mathbb{1}\{N_h^t = 0\} + \left(1 + \frac{1}{H}\right) \sum_{t=1}^T \delta_{h+1}^t + \sum_{t=1}^T \xi_h^t + \sum_{t=1}^T \mathcal{B}_h^t.
\end{aligned}
$$

By rolling out the upper bound on regret (8) and using inequality $(1 + 1/H)^{H-h} \leq e$ we have

$$
\mathfrak{R}^T \leq eH \sum_{t=1}^T \sum_{h=1}^H \mathbb{1}\{N_h^t = 0\} + \sum_{t=1}^T \sum_{h=1}^H (1 + 1/H)^{H-h} \xi_h^t + e \sum_{t=1}^T \sum_{h=1}^H \mathcal{B}_h^t.
$$

□

*Proof of Theorem 1.* First, we notice that the event $\mathcal{G}'(\delta)$ defined in Lemma 5, holds with probability at least $1 - \delta$ by Lemma 4 and Proposition 3. Thus, we may assume that $\mathcal{G}'(\delta)$ holds.

We start from the decomposition given by Lemma 5

$$\mathfrak{R}^T \leq \mathrm{e}H \sum_{t=1}^{T}\sum_{h=1}^{H} \mathbb{1}\{k_h^t(s_h^t, a_h^t) = -1\} + \sum_{t=1}^{T}\sum_{h=1}^{H}(1+1/H)^{H-h}\xi_h^t + \mathrm{e}\sum_{t=1}^{T}\sum_{h=1}^{H}\mathcal{B}_h^t.$$

The first term is upper bounded by $\mathrm{e}SAH^3$, since there is no more than $H$ visits of each state-action-step triple before the update for the first stage. The second term is bounded by $\widetilde{\mathcal{O}}(\sqrt{H^3 T})$ by a definition of the event $\mathcal{E}(\delta)$ in Lemma 4. To upper bound the last term we have to analyze the following sum

$$\sum_{t=1}^{T}\sum_{h=1}^{H} \frac{\mathbb{1}\{e_{k_h^t(s_h^t,a_h^t)} > 0\}}{\sqrt{e_{k_h^t(s_h^t,a_h^t)}}} \leq \sum_{(s,a,h)\in\mathcal{S}\times\mathcal{A}\times[H]} \sum_{k=0}^{k_h^{T+1}(s,a)} \frac{e_{k+1}}{\sqrt{e_k}},$$

where

$$e_k = \left\lfloor \left(1 + \frac{1}{H}\right)^k H \right\rfloor \Rightarrow \frac{e_{k+1}}{\sqrt{e_k}} \leq 2\sqrt{e_k},$$

therefore by Cauchy inequality

$$\sum_{k=0}^{k_h^{T+1}(s,a)} \frac{e_{k+1}}{\sqrt{e_k}} \leq 2 \sum_{k=0}^{k_h^{T+1}(s,a)} \sqrt{e_k} \leq 2\sqrt{k_h^{T+1}(s,a)}\sqrt{\sum_{k=0}^{k_h^{T+1}(s,a)} e_k} \leq 2\sqrt{\frac{\log(T)}{\log(1+1/H)}}\sqrt{n_h^{T+1}(s,a)},$$

where we used the definition of the previous stage $k_h^{T+1}(s,a)$

$$N_h^{T+1}(s,a) \geq \sum_{k=0}^{k_h^{T+1}(s,a)} e^k,$$

thus by Cauchy inequality and inequality $\log(1+1/H) \geq 1/(4H)$ for $H \geq 1$

$$\sum_{t=1}^{T}\sum_{h=1}^{H} \frac{\mathbb{1}\{e_{k_h^t(s_h^t,a_h^t)}>0\}}{\sqrt{e_{k_h^t(s_h^t,a_h^t)}}} \leq 2\sqrt{H\log(T)} \sum_{(s,a,h)\in\mathcal{S}\times\mathcal{A}\times[H]} \sqrt{N_h^{T+1}(s,a)+1}$$

$$\leq 4\sqrt{SAH^2 \log(T)}\sqrt{\sum_{(s,a,h)} (N_h^{T+1}(s,a)+1)}$$

$$\leq 4\sqrt{SAH^3 T \log(T)} + 4SAH^2 \log(T).$$

Using this upper bound, we have

$$\sum_{t=1}^{T}\sum_{h=1}^{H}\mathcal{B}_h^t = \widetilde{\mathcal{O}}\left(H\sum_{t=1}^{T}\sum_{h=1}^{H} \frac{\mathbb{1}\{e_{k_h^t(s_h^t,a_h^t)} > 0\}}{\sqrt{e_{k_h^t(s_h^t,a_h^t)}}}\right) = \widetilde{\mathcal{O}}\left(\sqrt{H^5 SAT} + SAH^3\right).$$

Combining this upper bound with the previous ones, we conclude the statement. $\square$

# E Proofs for Metric algorithm

## E.1 Assumptions

In this section we proof Lemma 2 and Lemma 1.

*Proof of Lemma 1.* By the dual formula for 1-Wasserstein distance (see e.g. Section 6 of Peyré and Cuturi [2019]) we have

$$\mathcal{W}_1(p_h(s,a), p_h(s',a')) = \sup_{f \text{ is } 1-\text{Lipchitz}} \{p_h f(s,a) - p_h f(s',a')\}.$$

By Assumption 2 we have

$$p_h f(s,a) - p_h f(s',a') = \mathbb{E}_{\xi_h}[f(F_h(s,a,\xi_h)) - f(F_h(s',a',\xi_h))] \leq L_F \rho((s,a),(s',a')).$$

$\square$

*Proof of Lemma 2.* Let us proceed by a backward induction over $h$. For $h = H+1$ we have $Q^\star_{H+1}(s,a) = V^\star_{H+1}(s) = 0$, therefore they are 0-Lipchitz.

Next we assume that have for any $h' > h$ the statement of Lemma 2 holds. Then by Bellman equations

$$|Q^\star_h(s,a) - Q^\star_h(s',a')| \leq |r_h(s,a) + r_h(s',a')| + |p_h V^\star_{h+1}(s,a) - p_h V^\star_{h+1}(s',a')|.$$

By Assumption 2 we can represent the action of the transition kernel as follows

$$p_h V^\star_{h+1}(s,a) - p_h V^\star_{h+1}(s',a') = \mathbb{E}_{\xi_h}\left[V^\star_{h+1}(F_h(s,a,\xi_h)) - V^\star_{h+1}(F_h(s',a',\xi_h)\right].$$

Since by induction hypothesis $V^\star_{h+1}$ is $\sum_{h'=h+1}^{H} L_F^{h'-h} L_r$-Lipchitz and $F_h(\cdot, \xi_h)$ is $L_F$-Lipchitz, therefore

$$|Q^\star_h(s,a) - Q^\star_h(s',a')| \leq \left(L_r + L_F \cdot \sum_{h'=h+1}^{H} L_F^{h'-h} L_r\right)\rho((s,a),(s',a'))$$

$$\leq \left(\sum_{h'=h}^{H} L_F^{h'-h} L_r\right)\rho((s,a),(s',a'))$$

To show that $V^\star_h$ is also Lipchitz, we have that there is some action $a^\star$ equal to $\pi^\star(s)$ or $\pi^\star(s')$, such that

$$|V^\star_h(s) - V^\star_h(s')| \leq |Q^\star_h(s,a^\star) - Q^\star_h(s',a^\star)| \leq L_{V,h} \cdot \rho((s,a^\star),(s',a^\star)) \leq L_{V,h} \cdot \rho_{\mathcal{S}}(s,s'),$$

where in the end we used the sub-additivity assumption on metric over joint space (see Assumption 1).

$\square$

## E.2 Algorithm

Next we describe a simple non-adaptive version of our algorithm that works with metric spaces. We assume that for any $\varepsilon > 0$ we can compute a minimal $\varepsilon$-cover of state-action space $\mathcal{N}_\varepsilon$.[5]

Next we will use the same notation but with state-action pairs replaces with balls from a fixed cover $\mathcal{N}_\varepsilon$. To unify the notation, we define $\psi_\varepsilon : \mathcal{S} \times \mathcal{A} \to \mathcal{N}_\varepsilon$ that maps any point $(s,a)$ to any ball from $\varepsilon$-cover that contains it.

For any $t, h$ we define $B_h^t = \psi_\varepsilon(s_h^t, a_h^t)$. Next, let $n_h^t(B)$ be a number of visits of ball $B$ before the episode $t$: $n_h^t(B) = \sum_{k=1}^{t-1} \mathbb{1}\{B_h^k = B\}$.

Let $e_k = \lfloor (1+1/H)^k \cdot H \rfloor$ be length of each stage for any $k \geq 0$ and, by convention, $e_{-1} = 0$. We will call that in the beginning of episode $t$ a pair $(B,h)$ is in $k$-th stage if $n_h^t(B) \in [\sum_{i=0}^{k-1} e_i, \sum_{i=0}^{k} e_i)$.

---

[5]Remark that the greedy algorithm can easily generate $\varepsilon$-cover of size $N_{\varepsilon/2}$, that will not affect the asymptotic behavior of regret bounds, see Song and Sun [2019].

**Algorithm 4** Metric `Net-Staged-RandQL`

---

1: **Input:** inflation coefficient $\kappa$, $J$ ensemble size, number of prior transitions $n_0(k)$, prior reward $r_0$, dicretization level $\varepsilon$.
2: **Initialize:** $\varepsilon$-net $\mathcal{N}_\varepsilon$, $\overline{Q}_h(B) = \widetilde{Q}_h^j(B) = r_0 H$, initialize counters $\widetilde{n}_h(B) = 0$ for $j, h, B \in [J] \times [H] \times \mathcal{N}_\varepsilon$, stage $q_h(B) = 0$, quantization map $\psi_\varepsilon : \mathcal{S} \times \mathcal{A} \to \mathcal{N}_\varepsilon$.
3: **for** $t \in [T]$ **do**
4:    **for** $h \in [H]$ **do**
5:       Play $a_h \in \arg\max_a \overline{Q}_h(\psi_\varepsilon(s_h, a))$ and define $B_h = \psi_\varepsilon(s_h, a_h)$.
6:       Observe reward and next state $s_{h+1} \sim p_h(s_h, a_h)$.
7:       Sample learning rates $w_j \sim \text{Beta}(1/\kappa, (\widetilde{n} + n_0(q_h(B_h)))/\kappa)$ for $\widetilde{n} = \widetilde{n}_h(B_h)$.
8:       Compute value function $\overline{V}_{h+1}(s_{h+1}) = \max_{a \in \mathcal{A}} \overline{Q}_{h+1}(\psi_\varepsilon(s_{h+1}, a))$.
9:       Update temporary $Q$-values for all $j \in [J]$

$$\widetilde{Q}_h^j(B) := (1 - w_j)\widetilde{Q}_h^j(B) + w_j\left(r_h(s_h, a_h) + \overline{V}_{h+1}(s_{h+1})\right).$$

10:       Update counter $\widetilde{n}_h(B_h) := \widetilde{n}_h(B_h) + 1$
11:       **if** $\widetilde{n}_h(B_h) = \lfloor (1 + 1/H)^q H \rfloor$ for $q = q_h(B_h)$ is the current stage **then**
12:          Update policy $Q$-values $\overline{Q}_h(B_h) := \max_{j \in [J]} \widetilde{Q}_h^j(B_h)$.
13:          Reset temporary $Q$-values $\widetilde{Q}_h^j(B_h) := r_0 H$.
14:          Reset counter $\widetilde{n}_h(B_h) := 0$ and change stage $k_h(B_h) := k_h(B_h) + 1$.
15:       **end if**
16:    **end for**
17: **end for**

---

Let $\widetilde{n}_h^t(B)$ be a number of visits of state-action pair during the current stage in the beginning of episode $t$. Formally, $\widetilde{n}_h^t(B) = n_h^t(B) - \sum_{i=0}^{k-1} e_i$, where $k$ is an index of current stage.

Define $\kappa > 0$ be a posterior inflation coefficient, $n_0$ is a number of pseudo-transitions, and $J$ as a number of temporary $Q$-functions. Let $\widetilde{Q}_h^{t,j}$ be a $j$-th *temporary* $Q$-value and $\overline{Q}_h^t$ be a *policy* $Q$-value at the beginning of episode $t$, defined over the $\varepsilon$-cover. We initialize them as follows

$$\overline{Q}_h^1(B) = r_0 H, \quad \widetilde{Q}_h^{1,j}(s, a) = r_0 H.$$

Additionally, we define to the value function as follows

$$\overline{V}_h^t(s) = \max_{a \in \mathcal{A}} \overline{Q}_h^t(\psi_\varepsilon(s, a)).$$

Notice that we cannot precomute it as in the tabular setting, however, it is possible to use its values in lazy fashion.

For each transition we preform the following update of temporary Q-values over balls $B \in \mathcal{N}_\varepsilon$

$$\widetilde{Q}_h^{t+1/2,j}(B) = \begin{cases} (1 - w_{j,\widetilde{n}}) \cdot \widetilde{Q}_h^{t,j}(B) + w_{j,\widetilde{n}}[r_h(s_h^t, a_h^t) + \overline{V}_{h+1}^t(s_{h+1}^t)], & B = B_h^t \\ \widetilde{Q}_h^{t,j}(B) & \text{otherwise,} \end{cases}$$

where $\widetilde{n} = \widetilde{n}_h^t(B)$ is the number of visits of $(B, h)$ in the beginning of episode $t$, and $w_{j,\widetilde{n}}$ is a sequence of independent beta-distribution random variables $w_{j,\widetilde{n}} \sim \text{Beta}(1/\kappa, (\widetilde{n} + n_0)/\kappa)$.

Next we define the stage update as follows

$$\overline{Q}_h^{t+1}(B) = \begin{cases} \max_{j \in [J]} \widetilde{Q}_h^{t+1/2,j}(B) & \widetilde{n}_h^t(B) = \lfloor (1 + 1/H)^k H \rfloor \\ \overline{Q}_h^t(B) & \text{otherwise} \end{cases}$$

$$\widetilde{Q}_h^{t+1,j}(B) = \begin{cases} r_0 H & n_h^t(B) \in \widetilde{n}_h^t(B) = \lfloor (1 + 1/H)^k H \rfloor \\ \widetilde{Q}_h^{t+1/2,j}(B) & \text{otherwise} \end{cases}$$

$$\overline{V}_h^{t+1}(s) = \min\{r_0(H - h), \max_{a \in \mathcal{A}} \overline{Q}_h^{t+1}(\psi_\varepsilon(s, a))\};$$

$$\pi_h^{t+1}(s) \in \arg\max_{a \in \mathcal{A}} \overline{Q}_h^{t+1}(\psi_\varepsilon(s, a)),$$

where $k$ is the current stage. A detailed description of the algorithm is presented in Algorithm 4.

For episode $t$ we will call $k_h^t(B)$ the index of stage where $\overline{Q}_h^t(B)$ were updated (and $k_h^t(B) = -1$ if there was no update). For all $t$ we define $\tau_h^t(B) \leq t$ as the episode when the stage update happens. In other words, for any $t$ the following holds

$$\overline{Q}_h^{t+1}(B) = \max_{j \in [J]} \widetilde{Q}_h^{\tau_h^t(B)+1/2,j}(B),$$

where $\tau_h^t(B) = 0$ and $e_k = 0$ if there was no updates. To simplify the notation we will omit dependence on $(s, a, h)$ where it is deducible from the context.

We notice that in this case we use $e_k$ samples to compute $\widetilde{Q}^{\tau_h^t(B)+1/2,j}$ for $k = k_h^t(s, a)$. For this $k$ we define $\ell_{k,h}^i(s, a)$ as the time of $i$-th visit of state-action pair $(s, a)$ during $k$-th stage. Then we have the following decomposition

$$\widetilde{Q}_h^{\tau^t+1/2,j}(B) = \sum_{i=0}^{e_k} W_{j,e_k}^i \left( r_h(s_h^{\ell^i}, a_h^{\ell^i}) + \overline{V}_{h+1}^{\ell^i}(s_{h+1}^{\ell^i}) \right), \tag{9}$$

where we drop dependence on $k$ and $(B, h)$ in $\ell^i$ to simplify notations, using the convention $r_h(s_h^{\ell^0}, a_h^{\ell^0}) = r_0$, $\overline{V}_{h+1}^{\ell^0}(s_{h+1}^{\ell^0}) = r_0(H - 1)$ and the following aggregated weights

$$W_{j,n}^0 = \prod_{q=0}^{n-1}(1 - w_{j,q}), \quad W_{j,n}^i = w_{j,i-1} \cdot \prod_{q=i}^{n-1}(1 - w_{j,q}), \ i \geq 1.$$

### E.3 Concentration

Let $\beta^\star \colon (0, 1) \times \mathbb{N} \times (0, d_{\max}) \to \mathbb{R}_+$ and $\beta^B, \beta^{\mathrm{conc}}, \beta \colon (0, 1) \times (0, d_{\max}) \to \mathbb{R}_+$ be some function defined later on in Lemma 6. We define the following favorable events

$$\mathcal{E}^\star(\delta, \varepsilon) \triangleq \left\{ \forall t \in \mathbb{N}, \forall h \in [H], \forall B \in \mathcal{N}_\varepsilon, k = k_h^t(B), (s, a) = \mathrm{center}(B) : \right.$$

$$\left. \mathcal{K}_{\inf}\left( \frac{1}{e_k} \sum_{i=1}^{e_k} \delta_{V_{h+1}^\star(F_h(s,a,\xi_{h+1}^{\ell^i}))}, p_h V_{h+1}^\star(s, a) \right) \leq \frac{\beta^\star(\delta, e_k, \varepsilon)}{e_k} \right\},$$

$$\mathcal{E}^B(\delta, \varepsilon) \triangleq \left\{ \forall t \in [T], \forall h \in [H], \forall B \in \mathcal{N}_\varepsilon, \forall j \in [J], k = k_h^t(B) : \right.$$

$$\left| \sum_{i=0}^{e_k} (W_{j,e_k,k}^i - \mathbb{E}[W_{j,e_k,k}^i]) \left( r_h(s_h^{\ell^i}, a_h^{\ell^i}) + \overline{V}_{h+1}^{\ell^i}(s_{h+1}^{\ell^i}) \right) \right|$$

$$\left. \leq 60\mathrm{e}^2 \sqrt{\frac{r_0^2 H^2 \kappa \beta^B(\delta, \varepsilon)}{e_k + n_0(k)}} + 1200\mathrm{e} \frac{r_0 H \kappa \log(e_k + n_0(k))(\beta^B(\delta, \varepsilon))^2}{e_k + n_0(k)} \right\},$$

$$\mathcal{E}^{\mathrm{conc}}(\delta, \varepsilon) \triangleq \left\{ \forall t \in [T], \forall h \in [H], \forall B \in \mathcal{N}_\varepsilon, k = k_h^t(B) : \right.$$

$$\left| \frac{1}{e_k} \sum_{i=1}^{e_k} V_{h+1}^\star(s_{h+1}^{\ell_{k,h}^i(B)}) - p_h V_{h+1}^\star(s_h^{\ell_{k,h}^i(B)}, a_h^{\ell_{k,h}^i(B)}) \right| \leq \sqrt{\frac{2r_0^2 H^2 \beta^{\mathrm{conc}}(\delta, \varepsilon)}{e_k}} \right\}$$

$$\mathcal{E}(\delta) \triangleq \left\{ \sum_{t=1}^{T} \sum_{h=1}^{H} (1 + 1/H)^{H-h} \left| p_h[V_{h+1}^\star - V_{h+1}^{\pi_t}](s_h^t, a_h^t) - [V_{h+1}^\star - V_{h+1}^{\pi_t}](s_{h+1}^t) \right| \right.$$

$$\left. \leq 2\mathrm{e}r_0 H \sqrt{2HT\beta(\delta)}. \right\}$$

We also introduce the intersection of these events, $\mathcal{G}(\delta) \triangleq \mathcal{E}^\star(\delta) \cap \mathcal{E}^B(\delta) \cap \mathcal{E}^{\mathrm{conc}}(\delta) \cap \mathcal{E}(\delta)$. We prove that for the right choice of the functions $\beta^\star, \beta^{\mathrm{KL}}, \beta^{\mathrm{conc}}, \beta, \beta^{\mathrm{Var}}$ the above events hold with high probability.

**Lemma 6.** *For any $\delta \in (0,1)$ and $\varepsilon \in (0, d_{\max})$ and for the following choices of functions $\beta$,*

$$\beta^\star(\delta, n, \varepsilon) \triangleq \log(8H/\delta) + \log(N_\varepsilon) + 3\log(e\pi(2n+1)),$$

$$\beta^B(\delta, \varepsilon) \triangleq \log(8H/\delta) + \log(N_\varepsilon) + \log(TJ),$$

$$\beta^{\mathrm{conc}}(\delta, \varepsilon) \triangleq \log(8H/\delta) + \log(N_\varepsilon) + \log(2T),$$

$$\beta(\delta) \triangleq \log(16/\delta),$$

*it holds that*

$$\mathbb{P}[\mathcal{E}^\star(\delta, \varepsilon)] \geq 1 - \delta/8, \qquad \mathbb{P}[\mathcal{E}^B(\delta, \varepsilon)] \geq 1 - \delta/8,$$

$$\mathbb{P}[\mathcal{E}^{\mathrm{conc}}(\delta, \varepsilon)] \geq 1 - \delta/8, \qquad \mathbb{P}[\mathcal{E}(\delta)] \geq 1 - \delta/8.$$

*In particular, $\mathbb{P}[\mathcal{G}(\delta)] \geq 1 - \delta/2$.*

*Proof.* Let us describe the changes from the similar statement in Lemma 4.

Regarding event $\mathcal{E}^\star(\delta, \varepsilon)$, for any fixed ball $B$ we have exactly the same structure of the problem thanks to Assumption 2 and a sequence of i.i.d. random variables $\xi_h^{\ell^i}$. Thus, Theorem 4 combined with a union bound over $B \in \mathcal{N}_\varepsilon$ and $H \in [H]$ concludes $\mathbb{P}(\mathcal{E}^\star(\delta, \varepsilon)) \geq 1 - \delta/8$.

The proof for the event $\mathcal{E}^B(\delta, \varepsilon)$ remains the almost the same, with two differences: the predictable weights slightly changed but the upper bound for them remain the same, and we have take a union bound not over all state-action pairs $(s, a) \in \mathcal{S} \times \mathcal{A}$ but all over balls $B \in \mathcal{N}_\varepsilon$.

To show that $\mathbb{P}(\mathcal{E}^{\mathrm{conc}}(\delta, \varepsilon)) \geq 1 - \delta/8$, let us fix $B \in \mathcal{N}_\varepsilon, h \in [H]$ and $e_k \in [T]$. Then we can define a filtration $\mathcal{F}_{t,h} = \sigma\{\{(s_{h'}^\ell, a_{h'}^\ell, \pi^\ell), \ell < t, h' \in [H]\} \cup \{(s_{h'}^t, a_{h'}^t, \pi^t), h' \leq h\}\}$ and, since $\ell_{k,h}^i(B)$ are stopping times for all $i = 1, \ldots, e_k$, we can define the stopped filtration $\widetilde{\mathcal{F}}_i = \mathcal{F}_{\ell^i, h}$. Then we notice that $X_i = V_{h+1}^\star(s_{h+1}^{\ell_{k,h}^i(B)}) - p_h V_{h+1}^\star(s_h^{\ell_{k,h}^i(B)}, a_h^{\ell_{k,h}^i(B)})$ forms a martingale-difference sequence with respect to $\widetilde{\mathcal{F}_{i,h}}$. Thus, by Azuma-Hoeffding inequality and a union bound we have $\mathbb{P}(\mathcal{E}^{\mathrm{conc}}(\delta, \varepsilon)) \geq 1 - \delta/8$.

The proof of $\mathbb{P}(\mathcal{E}(\delta)) \geq 1 - \delta/8$ remains exactly the same as in Lemma 4. $\qquad \square$

### E.4 Optimism

In this section we prove that our estimate of $Q$-function $\overline{Q}_h^t(s, a)$ is optimistic that is the event

$$\mathcal{E}_{\mathrm{opt}}(\varepsilon) \triangleq \left\{ \forall t \in [T], h \in [H], (s, a) \in \mathcal{S} \times \mathcal{A} : \overline{Q}_h^t(\psi_\varepsilon(s, a)) \geq Q_h^\star(s, a) \right\}. \tag{10}$$

holds with high probability on the event $\mathcal{E}^\star(\delta, \varepsilon)$.

Define constants

$$c_0 \triangleq \frac{8}{\pi} \left( \frac{4}{\sqrt{\log(17/16)}} + 8 + \frac{49 \cdot 4\sqrt{6}}{9} \right)^2 + 1. \tag{11}$$

and slightly another constant

$$\tilde{c}_J \triangleq \frac{1}{\log\left(\frac{4}{3+\Phi(1)}\right)}, \tag{12}$$

where $\Phi(\cdot)$ is a CDF of a normal distribution.

**Proposition 4.** Define a constant $L = L_r + L_V(1 + L_F)$. Assume that $J = \lceil \tilde{c}_J \cdot (\log(2HT/\delta) + \log(N_\varepsilon) \rceil$, $\kappa = 2\beta^\star(\delta, T, \varepsilon)$, $r_0 = 2$, and a prior count $n_0(k) = \lceil \tilde{n}_0 + \kappa + \frac{\varepsilon L}{H-1} \cdot (e_k + \tilde{n}_0 + \kappa) \rceil$ dependent on the stage $k$, where $\tilde{n}_0 = (c_0 + 1 + \log_{17/16}(T)) \cdot \kappa$.

Then on event $\mathcal{E}^\star(\delta, \varepsilon)$ the following event

$$\mathcal{E}_{\mathrm{anticonc}} \triangleq \left\{ \forall t \in [T] \; \forall h \in [H] \; \forall B \in \mathcal{N}_\varepsilon : \text{ for } k = k_h^t(B), (s,a) = \mathrm{center}(B) : \right.$$

$$\left. \max_{j \in [J]} \left\{ W_{j,e_k,k}^0 r_0(H-1) + \sum_{i=1}^{e_k} W_{j,e_k,k}^i V_{h+1}^\star\big(F_h(s,a,\xi_h^{\ell^i})\big) \right\} \geq p_h V_{h+1}^\star(s,a) + L\varepsilon \right\}$$

holds with probability at least $1 - \delta/2$.

**Remark 1.** We notice that the obtained result is connected to the theory of Dirichlet processes.

First, let us define the Dirichlet process, following Ferguson [1973]. The stochastic process $G$, indexed by elements $B$ of $\mathsf{X}$, is a Dirichlet Process with parameter $\nu$ ($G \sim \mathrm{DP}(\nu)$) if

$$G(B_1), \dots, G(B_d) \sim \mathrm{Dir}(\nu(B_1), \dots, \nu(B_d)),$$

for any measurable partition $(B_1, \dots, B_d)$ of $\mathsf{X}$.

Let $\widehat{P}_n = \frac{1}{n} \sum_{i=1}^n \delta_{Z_i}$ be an empirical measure of an i.i.d. sample $Z_1, \dots, Z_n \sim P$. Let $\nu$ be a finite (not necessarily probability) measure on $\mathsf{X}$ and $\widetilde{P}_n \sim \mathrm{DP}(\nu + n\widehat{P}_n)$. Then we have the following representation for the expectations of a function $f \colon \mathsf{X} \to \mathbb{R}$ over a sampled measure $\widetilde{P}_n$ (see Theorem 14.37 of Ghosal and Van der Vaart [2017] with $\sigma = 0$ for a proof)

$$\widetilde{P}_n f = V_n \cdot Qf + (1 - V_n) \sum_{i=1}^n W_i f(Z_i),$$

where $V_n \sim \mathrm{Beta}(|\nu|, n)$, $Q \sim \mathrm{DP}(\nu)$, and a vector $(W_1, \dots, W_n)$ follows uniform Dirichlet distribution $\mathrm{Dir}(1, \dots, 1)$. If we take $\nu = n_0 \cdot \delta_{Z_0}$ for some $Z_0 \in \mathsf{X}$ such that $f(Z_0) = r_0(H-1)$[6], then by a stick-breaking process representation of the Dirichlet distribution we have

$$\widetilde{P}_n f = \widetilde{W}_0 r_0(H-1) + \sum_{i=1}^n \widetilde{W}_1 f(Z_i), \quad (\widetilde{W}_0, \dots, \widetilde{W}_1) \sim \mathrm{Dir}(n_0, 1, \dots, 1).$$

By taking an appropriate $\mathsf{X}$ and $f$ we have that Proposition 4 could be interpret as a deriving a lower bound on the probability of $\mathbb{P}[\widetilde{P}_n f \geq Pf + \varepsilon L \mid \{Z_i\}_{i=1}^n]$.

*Proof.* First for all, let us fix $t \in [T], h \in [H]$ and $B \in \mathcal{N}_\varepsilon$ and, consequently, $k = k_h^t(B)$. Also, let fix $j \in [J]$. To simplify the notation in the sequel, define $X_0 = r_0(H-1)$ and $X_i = V_{h+1}^\star(F_h(s,a,\xi_h^{\ell^i}))$ for $i > 0$. Notice that $X_i$ for $i > 0$ is a sequence of i.i.d. random variables supported on $[0, H-h-1]$.

By Lemma 3 we have $(W_{j,e_k,k}^0, \dots, W_{j,e_k,k}^{e_k}) \sim \mathrm{Dir}(n_0(k)/\kappa, 1/\kappa, \dots, 1/\kappa)$. Then we use the aggregation property of Dirichlet distribution: there is a vector $(\widetilde{W}_j^{-1}, \dots, \widetilde{W}_j^{e_k}) \sim \mathrm{Dir}((n_0(k) - \widetilde{n}_0)/\kappa, \widetilde{n}_0/\kappa, 1/\kappa, \dots, 1/\kappa)$ such that

$$\sum_{i=0}^{e_k} W_{j,e_k,k}^i X_i = \widetilde{W}_j^{-1} X_0 + \sum_{i=0}^{e_k} \widetilde{W}_j^i X_i.$$

Next we are going to represent the Dirichlet random vector $\widetilde{W}$ by a stick breaking process (or, equivalently, represent via the generalized Dirichlet distribution)

$$\widetilde{W}_j^{-1} = \xi_j \qquad\qquad \xi_j \sim \mathrm{Beta}((n_0(k) - \widetilde{n}_0)/\kappa, (e_k + \widetilde{n}_0)/\kappa),$$

$$(\widetilde{W}_j^0, \dots, \widetilde{W}_j^{e_k}) = (1 - \xi_j) \cdot (\widehat{W}_j^0, \dots, \widehat{W}_j^{e_k}), \qquad \widehat{W}_j \sim \mathrm{Dir}(\widetilde{n}_0/\kappa, 1/\kappa, \dots, 1/\kappa),$$

---

[6]We can augment the space $\mathsf{X}$ with this additional point if needed

where $\xi_j$ and $\widehat{W}_j$ are independent. Therefore, we have the final decomposition

$$\sum_{i=0}^{e_k} W^i_{j,e_k,k} X_i - p_h V^\star_{h+1}(s,a) - \varepsilon L = \underbrace{\xi_j \big( r_0(H-1) - p_h V^\star_{h+1}(s,a) \big) - \varepsilon L}_{T_{\text{approx}}}$$

$$+ (1-\xi_j) \cdot \underbrace{\left( \sum_{i=0}^{e_k} \widehat{W}^i_j X_i - p_h V^\star_{h+1}(s,a) \right)}_{T_{\text{stoch}}}.$$

By independence of $\xi_j$ and $\widehat{W}_j$ we have

$$\mathbb{P}\left[ \sum_{i=0}^{e_k} W^i_{j,e_k,k} X_i \geq p_h V^\star_{h+1}(s,a) + \varepsilon L | \{X_i\}_{i=1}^{e_k} \right] \geq \mathbb{P}[T_{\text{approx}} \geq 0] \cdot \mathbb{P}[T_{\text{stoch}} \geq 0].$$

We split our problem to lower bound the two separate probabilities.

**Approximation error**  To deal with approximation error, we first of all notice that $p_h V^\star_{h+1}(s,a) \leq H - 1$, therefore we have

$$\mathbb{P}[T_{\text{approx}} \geq 0] = \mathbb{P}\left[ \xi_j \geq \frac{\varepsilon L}{H-1} \right].$$

Next we assume that $\varepsilon < (H-1)/L$, since $\xi_j \sim \text{Beta}((n_0(k) - \widetilde{n}_0)/\kappa, (e_k + \widetilde{n}_0)/\kappa)$, we may apply Alfers and Dinges [1984, Theorem 1.2"]

$$\mathbb{P}[T_{\text{approx}} \geq 0] \geq \Phi\Big( -\text{sign}(p-\mu) \cdot \sqrt{2\alpha \, \text{kl}(p,\mu)} \Big),$$

where $p = (n_0(k) - \widetilde{n}_0 - \kappa)/(e_k + \widetilde{n}_0 - \kappa)$ and $\mu = \varepsilon L/(H-1)$. Since $n_0(k) = \lceil \widetilde{n}_0 + \kappa + \frac{\varepsilon L}{H-1} \cdot (e_k + \widetilde{n}_0 + \kappa) \rceil$, we have $\mathbb{P}[T_{\text{approx}} \geq 0] \geq 1/2$.

**Stochastic error**  Since $X_0 = r_0(H-1)$ is an upper bound on $V$-function, and we have that the weight of the first atom $\alpha_0 \triangleq \widetilde{n}_0/\kappa - 1 = c_0 + \log_{17/16}(T) - 1$ for $c_0$ defined in (11).

Define a measure $\bar{\nu}_{e_k} = \frac{\widetilde{n}_0 - \kappa}{e_k + \widetilde{n}_0 - \kappa} \delta_{X_0} + \sum_{i=1}^{e_k} \frac{1}{e_k + n_0 - 1} \delta_{X_i}$. Since $p_h V^\star_{h+1}(s,a) \leq H - h - 1$, we can apply Lemma 10 with a fixed $\varepsilon = 1/2$ conditioned on independent random variables $X_i$

$$\mathbb{P}\left[ \sum_{i=0}^{e_k} \widehat{W}^i_j X_i \geq p_h V^\star_{h+1}(s,a) \mid \{X_i\}_{i=1}^{e_k} \right]$$

$$\geq \frac{1}{2} \left( 1 - \Phi\left( \sqrt{\frac{2(e_k + n_0 - \kappa) \, \mathcal{K}_{\text{inf}}\big( \bar{\nu}_{e_k}, p_h V^\star_{h+1}(s,a) \big)}{\kappa}} \right) \right),$$

where $\Phi$ is a CDF of a normal distribution. By Lemma 12 and the event $\mathcal{E}^\star(\delta, \varepsilon)$

$$(e_k + n_0 - \kappa) \, \mathcal{K}_{\text{inf}}\big( \bar{\nu}_{e_k}, p_h V^\star_{h+1}(s,a) \big) \leq e_k \, \mathcal{K}_{\text{inf}}\big( \widehat{\nu}_{e_k}, p_h V^\star_{h+1}(s,a) \big) \leq \beta^\star(\delta, T, \varepsilon),$$

where $\widehat{\nu}_{e_k} = \frac{1}{e_k} \sum_{i=1}^{e_k} \delta_{V^\star_{h+1}(F(s,a,\xi^{\ell i}_{h+1}))}$, and, as a corollary

$$\mathbb{P}\left[ \sum_{i=0}^{e_k} \widehat{W}^i_j X_i \geq p_h V^\star_{h+1}(s,a) \mid \mathcal{E}^\star(\delta, \varepsilon), \{X_i\}_{i=1}^{e_k} \right] \geq \frac{1}{2} \left( 1 - \Phi\left( \sqrt{\frac{2\beta^\star(\delta, T, \varepsilon)}{\kappa}} \right) \right).$$

By taking $\kappa = 2\beta^\star(\delta, T, \varepsilon)$ we have a constant probability of being optimistic for stochastic error

$$\mathbb{P}[T_{\text{stoch}} \geq 0 \mid \mathcal{E}^\star(\delta, \varepsilon)] \geq \frac{1 - \Phi(1)}{2}.$$

Overall, combining two lower bound for approximation and stochastic terms, we have

$$\mathbb{P}\left[ \sum_{i=0}^{e_k} W^i_{j,e_k,k} X_i \geq p_h V^\star_{h+1}(s,a) + \varepsilon L | \mathcal{E}^\star(\delta, \varepsilon) \right] \geq \frac{1 - \Phi(1)}{4} = \gamma.$$

Next, using a choice $J = \lceil (\log(2HT/\delta) + \log(N_\varepsilon))/\log(1/(1-\gamma)) \rceil = \lceil \tilde{c}_J \cdot (\log(2HT/\delta) + \log(N_\varepsilon)) \rceil$

$$\mathbb{P}\left[ \max_{j \in [J]} \left\{ \sum_{i=0}^{e_k} W^i_{j,e_k,k} X_i \right\} \geq p_h V^\star_{h+1}(s,a) + \varepsilon L |\mathcal{E}^\star(\delta,\varepsilon)| \right] \geq 1 - (1-\gamma)^J \geq 1 - \frac{\delta}{2N_\varepsilon HT}.$$

By a union bound we conclude the statement. $\qquad\square$

Next we provide a connection between $\mathcal{E}^{\mathrm{anticonc}}$ and $\mathcal{E}^{\mathrm{opt}}$.

**Proposition 5.** It holds $\mathcal{E}^{\mathrm{opt}} \subseteq \mathcal{E}^{\mathrm{anticonc}}$.

*Proof.* We proceed by a backward induction over $h$. Base of induction $h = H + 1$ is trivial. Fix state-action pair $(s,a)$ and let us call $(s',a')$ a center of the ball $\psi_\varepsilon(s,a)$ that is the ball where $(s,a)$ contains.

Next by the update formula for $\overline{Q}^t_h$, and Bellman equations

$$\overline{Q}^t_h(\psi_\varepsilon(s,a)) - Q^\star_h(s,a) = \max_{j \in [J]} \left\{ \sum_{i=0}^n W^i_{j,n}[r_h(s^{\ell^i}_h, a^{\ell^i}_h) - r_h(s',a')] \right.$$
$$\left. + \sum_{i=0}^n W^i_{j,n} \overline{V}^{\ell^i}_{h+1}(s^{\ell^i}_{h+1}) - p_h V^\star_{h+1}(s',a') \right\} + [Q^\star_h(s,a) - Q^\star_h(s',a')],$$

where $n = e_{k^t_h(B)}$ and we drop dependence on $k, t, h, s, a$ in $\ell^i$. By induction hypothesis we have $\overline{V}^{\ell^i}_{h+1}(s') \geq \overline{Q}^{\ell^i}_{h+1}(\psi_\varepsilon(s', \pi^\star(s'))) \geq Q^\star_{h+1}(s', \pi^\star(s')) = V^\star_{h+1}(s')$ for any $i$, thus combining it with Lipchitz continuity of reward function and $Q^\star$, and the value of $r_h(s^{\ell^0}, a^{\ell^0}) = r_0 > r_h(s,a)$,

$$\overline{Q}^t_h(\psi_\varepsilon(s,a)) - Q^\star_h(s,a) \geq \max_{j \in [J]} \left\{ W^0_{j,n} r_0(H-1) + \sum_{i=1}^n W^i_{j,n} V^\star_{h+1}(F_h(s^{\ell^i}_h, a^{\ell^i}_h, \xi^{\ell^i}_h)) \right\}$$
$$- p_h V^\star_{h+1}(s',a') - (L_r + L_V)\varepsilon.$$

Next we apply Lipschitz continuity of $F_h$ and $V^\star_{h+1}$ and obtain

$$\overline{Q}^t_h(\psi_\varepsilon(s,a)) - Q^\star_h(s,a) \geq \max_{j \in [J]} \left\{ W^0_{j,n} r_0(H-1) + \sum_{i=1}^n W^i_{j,n} V^\star_{h+1}(F_h(s', a', \xi^{\ell^i}_h)) \right\}$$
$$- p_h V^\star_{h+1}(s',a') - (L_r + L_V(1 + L_F))\varepsilon.$$

By the definition of event $\mathcal{E}^{\mathrm{anticonc}}$ we conclude the statement. $\qquad\square$

**Proposition 6** (Optimism). Define a constant $L = L_r + L_V(1 + L_F)$. Assume that $J = \lceil \tilde{c}_J \cdot (\log(2HT/\delta) + \log(N_\varepsilon)) \rceil$, $\kappa = 2\beta^\star(\delta, T, \varepsilon)$, $r_0 = 2$, and a prior count $n_0(k) = \lceil \tilde{n}_0 + \kappa + \frac{\varepsilon L}{H-1} \cdot (e_k + \tilde{n}_0 + \kappa) \rceil$ dependent on the stage $k$, where $\tilde{n}_0 = (c_0 + 1 + \log_{17/16}(2e_k)) \cdot \kappa$, $c_0$ is defined in (11), $\tilde{c}_J$ is defined in (12). Then $\mathbb{P}(\mathcal{E}^{\mathrm{opt}} \mid \mathcal{E}^\star(\delta,\varepsilon)) \geq 1 - \delta/2$.

### E.5  Regret Bounds

As in the tabular setting, we first connect our algorithm to the algorithm by Song and Sun [2019], using the following corollary. Define an event $\mathcal{G}'(\delta,\varepsilon) = \mathcal{G}(\delta,\varepsilon) \cap \mathcal{E}^{\mathrm{opt}}$.

Let us define the logarithmic term as follows

$$\beta^{\max}(\delta,\varepsilon) = \max\{\kappa, \tilde{n}_0/\kappa, \beta^B(\delta,\varepsilon), \beta(\delta,\varepsilon), \beta^{\mathrm{conc}}(\delta,\varepsilon)\}$$

that has dependence of order $\mathcal{O}(\log(TH/\delta) + \log N_\varepsilon)$.

**Corollary 2.** *Fix $\varepsilon \in (0, L_V/H)$ and assume conditions of Proposition 6. Let $t \in [T], h \in [H], B \in \mathcal{N}_\varepsilon$. Define $k = k^t_h(B)$ and let $\ell^1 < \ldots < \ell^{e_k}$ be a excursions of $(B,h)$ till the end of the previous stage. Then on the event $\mathcal{G}'(\delta)$ the following bound holds for $k \geq 0$ and any $(s,a) \in B$*

$$0 \leq \overline{Q}^t_h(B) - Q^\star_h(s,a) \leq \frac{1}{e_k} \sum_{i=1}^{e_k} [\overline{V}^{\ell^i}_{h+1}(s^{\ell^i}_{h+1}) - V^\star_{h+1}(s^{\ell^i}_{h+1})] + \mathcal{B}^t_h(k),$$

*where*

$$\mathcal{B}_h^t(k) = 121\mathrm{e}^2 \cdot \sqrt{\frac{H^2(\beta^{\max}(\delta,\varepsilon))^2}{e_k}} + 2401\mathrm{e} \cdot \frac{H(\beta^{\max}(\delta,\varepsilon))^4}{e_k} + 3(L_r + (1 + L_F)L_V)\varepsilon.$$

*Proof.* The lower bound follows from the definition of the event $\mathcal{E}^{\mathrm{opt}}$. For the upper bound we first apply the decomposition for $\overline{Q}_h^t(s,a)$ and the definition of event $\mathcal{E}^B(\delta,\varepsilon)$ from Lemma 6

$$\overline{Q}_h^t(B) = \max_{j \in [J]} \left\{ \sum_{i=0}^{e_k} W_{j,n}^i \left( r_h(s_h^{\ell^i}, a_h^{\ell^i}) + \overline{V}_{h+1}^{\ell^i}(s_{h+1}^{\ell^i}) \right) \right\}$$

$$\leq \frac{1}{e_k + n_0(k)} \sum_{i=1}^{e_k} \left( r_h(s_h^{\ell^i}, a_h^{\ell^i}) + \overline{V}_{h+1}^{\ell^i}(s_{h+1}^{\ell^i}) \right) + \frac{n_0(k) \cdot 2H}{e_k + n_0(k)}$$

$$+ 120\mathrm{e}^2 \sqrt{\frac{H^2 \kappa \beta^B(\delta,\varepsilon)}{e_k + n_0(k)}} + 2400\mathrm{e} \frac{H\kappa \log(n + n_0(k))(\beta^B(\delta,\varepsilon))^2}{e_k + n_0(k)}.$$

Additionally, by Bellman equations

$$Q_h^\star(s,a) = \frac{1}{e_k} \sum_{i=1}^{e_k} Q_h^\star(s_h^{\ell^i}, a_h^{\ell^i}) + \frac{1}{e_k} \sum_{i=1}^{e_k} \left( Q_h^\star(s,a) - Q_h^\star(s_h^{\ell^i}, a_h^{\ell^i}) \right)$$

$$\geq \frac{1}{e_k} \sum_{i=1}^{e_k} \left( r_h(s_h^{\ell^i}, a_h^{\ell^i}) + p_h V_{h+1}^\star(s_h^{\ell^i}, a_h^{\ell^i}) \right) - 2\varepsilon L_V.$$

Combining and using the fact that $n_0(k) \leq \frac{L\varepsilon}{H-1} \cdot (e_k + n_0(k)) + \widetilde{n}_0 + \kappa$ for $L = L_r + (1 + L_F)L_V$

$$\overline{Q}_h^t(s,a) - Q_h^\star(s,a) \leq \frac{1}{e_k} \sum_{i=1}^{e_k} \left[ \overline{V}_{h+1}^{\ell^i} - V_{h+1}^\star \right](s_{h+1}^{\ell^i}) + \frac{1}{e_k} \sum_{i=1}^{e_k} \left[ V_{h+1}^\star(s_{h+1}^{\ell^i}) - p_h V_{h+1}^\star(s_h^{\ell^i}, a_h^{\ell^i}) \right]$$

$$+ 120\mathrm{e}^2 \cdot \sqrt{\frac{H^2(\beta^{\max}(\delta,\varepsilon))^2}{e_k}} + (2400\mathrm{e} + 2)\frac{H(\beta^{\max}(\delta,\varepsilon))^4}{e_k} + 3L\varepsilon.$$

Finally, the applications of event $\mathcal{E}^{\mathrm{conc}}(\delta,\varepsilon)$ concludes the statement. $\qquad\square$

Let us define $\delta_h^t = \overline{V}_h^t(s_h^t) - V_h^{\pi^t}(s_h^t)$ and $\zeta_h^t = \overline{V}_h^t(s_h^t) - V_h^\star(s_h^t)$.

**Lemma 7.** *Assume conditions of Proposition 6. Then on event $\mathcal{G}'(\delta,\varepsilon) = \mathcal{G}(\delta,\varepsilon) \cap \mathcal{E}^{\mathrm{opt}}$, where $\mathcal{G}(\delta,\varepsilon)$ is defined in Lemma 6, the following upper bound on regret holds*

$$\mathfrak{R}^T \leq 2\mathrm{e}H \sum_{t=1}^T \sum_{h=1}^H \mathbb{1}\{N_h^t = 0\} + \sum_{t=1}^t \sum_{h=1}^H (1 + 1/H)^{H-h}\xi_h^t + \mathrm{e}\sum_{t=1}^T \sum_{h=1}^H \mathcal{B}_h^t.$$

*where $\xi_h^t = p_h[V_{h+1}^\star - V_{h+1}^{\pi^t}](s_h^t, a_h^t) - [V_{h+1}^\star - V_{h+1}^{\pi^t}](s_{h+1}^t)$ and $\mathcal{B}_h^t = \mathcal{B}_h^t(k_h^t(s_h^t, a_h^t)) \cdot \mathbb{1}\{k_h^t(s_h^t, a_h^t) \geq 0\}$ for $\mathcal{B}_h^t$ defined in Corollary 2.*

*Proof.* As in the tabular setting, we notice that on the event $\mathcal{E}^{\mathrm{opt}}$ we can upper bound the regret in terms of $\delta_1^t$.

$$\mathfrak{R}^T \leq \sum_{t=1}^T \delta_1^t. \tag{13}$$

Next we analyze $\delta_h^t$. Since $a_h^t = \arg\max_{a \in \mathcal{A}} \overline{Q}_h^t(\psi_\varepsilon(s_h^t, a))$, we can use Corollary 2 and Bellman equations in the following way

$$\delta_h^t = \overline{V}_h^t(s_h^t) - V_h^{\pi^t}(s_h^t) = \overline{Q}_h^t(B_h^t) - Q_h^{\pi^t}(s_h^t, a_h^t)$$

$$= \overline{Q}_h^t(B_h^t) - Q_h^\star(s_h^t, a_h^t) + Q_h^\star(s_h^t, a_h^t) - Q_h^{\pi^t}(s_h^t, a_h^t)$$

$$\leq r_0 H \mathbb{1}\{N_h^t = 0\} + \mathbb{1}\{N_h^t > 0\} \left( \frac{1}{N_h^t} \sum_{i=1}^{N_h^t} \zeta_{h+1}^{\ell_{t,h}^i} + \mathcal{B}_h^t(k_h^t) + p_h[V_{h+1}^\star - V_{h+1}^{\pi^t}](s_h^t, a_h^t) \right).$$

where $k_h^t = k_h^t(B_h^t)$, $N_h^t = e_{k_h^t}$, $\ell_{t,h}^i$ is an $i$-th visitation of the ball $B_h^t$ during an stage $k_h^t$, and additionally by a convention $0/0 = 0$.

Define $\xi_h^t = p_h[V_{h+1}^\star - V_{h+1}^{\pi^t}](s_h^t, a_h^t) - [V_{h+1}^\star - V_{h+1}^{\pi^t}](s_{h+1}^t)$ a martingale-difference sequence, and $\mathcal{B}_h^t = \mathcal{B}_h^t(k_h^t)\mathbb{1}\{N_h^t > 0\}$ then

$$\delta_h^t \leq r_0 H \mathbb{1}\{N_h^t = 0\} + \frac{\mathbb{1}\{N_h^t > 0\}}{N_h^t} \sum_{i=1}^{N_h^t} \zeta_{h+1}^{\ell_{t,h}^i} - \zeta_{h+1}^t + \delta_{h+1}^t + \xi_h^t + \mathcal{B}_h^t.$$

and, as a result

$$\sum_{t=1}^T \delta_h^t \leq r_0 H \sum_{t=1}^T \mathbb{1}\{N_h^t = 0\} + \sum_{t=1}^T \frac{\mathbb{1}\{N_h^t > 0\}}{N_h^t} \sum_{i=1}^{N_h^t} \zeta_{h+1}^{\ell_{t,h}^i}$$
$$- \sum_{t=1}^T \zeta_{h+1}^t + \sum_{t=1}^T \delta_{h+1}^t + \sum_{t=1}^T \xi_h^t + \sum_{t=1}^T \mathcal{B}_h^t.$$

For the second term we may repeat arguments as in the proof of Lemma 5 and obtain

$$\sum_{q=1}^T \zeta_{h+1}^q \cdot \sum_{t=1}^T \frac{\mathbb{1}\{k_h^t \geq 0\}}{N_h^t} \sum_{i=1}^{N_h^t} \mathbb{1}\{\ell_{t,h}^i = q\} \leq \left(1 + \frac{1}{H}\right) \sum_{q=1}^T \zeta_{h+1}^q.$$

After a simple algebraic manipulations and using the fact that $\zeta_h^t \leq \delta_h^t$

$$\sum_{t=1}^T \delta_h^t \leq H \sum_{t=1}^T \mathbb{1}\{N_h^t = 0\} + \sum_{t=1}^T (1 + 1/H)\zeta_{h+1}^t - \sum_{t=1}^T \zeta_{h+1}^t + \sum_{t=1}^T \delta_{h+1}^t + \sum_{t=1}^T \xi_h^t + \sum_{t=1}^T \mathcal{B}_h^t$$
$$\leq H \sum_{t=1}^T \mathbb{1}\{N_h^t = 0\} + \left(1 + \frac{1}{H}\right) \sum_{t=1}^T \delta_{h+1}^t + \sum_{t=1}^T \xi_h^t + \sum_{t=1}^T \mathcal{B}_h^t.$$

By rolling out the upper bound on regret (13) we have

$$\mathfrak{R}^T \leq 2eH \sum_{t=1}^T \sum_{h=1}^H \mathbb{1}\{N_h^t = 0\} + \sum_{t=1}^t \sum_{h=1}^H (1 + 1/H)^{H-h}\xi_h^t + e \sum_{t=1}^T \sum_{h=1}^H \mathcal{B}_h^t.$$

$\square$

*Proof of Theorem 2.* First, we notice that the event $\mathcal{G}'(\delta, \varepsilon)$ defined in Lemma 7, holds with probability at least $1 - \delta$ by Lemma 6 and Proposition 6. Thus, we may assume that $\mathcal{G}'(\delta, \varepsilon)$ holds for $\varepsilon > 0$ that we will specify later.

By Lemma 7

$$\mathfrak{R}^T \leq 2eH \sum_{t=1}^T \sum_{h=1}^H \mathbb{1}\{k_h^t = -1\} + \sum_{t=1}^t \sum_{h=1}^H (1 + 1/H)^{H-h}\xi_h^t + e \sum_{t=1}^T \sum_{h=1}^H \mathcal{B}_h^t.$$

The first term is upper bounded by $2eH^3 \cdot N_\varepsilon$, since there is no more than $H$ visits of each ball in $\varepsilon$-net before the update for the first stage. The second term is bounded by $\mathcal{O}(\sqrt{H^3 T \beta^{\max}(\delta, \varepsilon)})$ by a definition of the event $\mathcal{E}(\delta)$ in Lemma 6.

To analyze the last term, consider the following sum

$$\sum_{t=1}^T \sum_{h=1}^H \frac{\mathbb{1}\{e_{k_h^t(B_h^t)} > 0\}}{\sqrt{e_{k_h^t(B_h^t)}}} \leq \sum_{(B,h) \in \mathcal{N}_\varepsilon \times [H]} \sum_{k=0}^{k_h^T(B)} \frac{e_{k+1}}{\sqrt{e_k}},$$

where
$$e_k = \left\lfloor \left(1 + \frac{1}{H}\right)^k H \right\rfloor \Rightarrow \frac{e_{k+1}}{\sqrt{e_k}} \leq 2\sqrt{H}\left(1 + \frac{1}{H}\right)^{k/2},$$

therefore
$$\sum_{h=0}^{k_h^T(B)} \frac{e_{k+1}}{\sqrt{e_k}} \leq 4\sqrt{H}\frac{(1 + 1/H)^{(k_h^T(B)+1)/2}}{\sqrt{1 + 1/H} - 1} = 4H\sqrt{e^{k_h^T(B)+1}}. \tag{14}$$

Notice that
$$N_h^{T+1}(B) \geq \sum_{k=0}^{k_h^T(B)} e^k = H(e^{k_h^T(B)+1} - 1) \Rightarrow e^{k_h^T(B)+1} \leq \frac{N_h^{T+1}(B) + 1}{H}$$

thus from the Cauchy-Schwarz inequality
$$\sum_{t=1}^{T}\sum_{h=1}^{H} \frac{\mathbb{1}\{e_{k_h^t(B_h^t)} > 0\}}{\sqrt{e_{k_h^t(B_h^t)}}} \leq 4\sqrt{H} \sum_{(B,h)\in\mathcal{N}_\varepsilon\times[H]} \sqrt{N_h^{T+1}(B) + 1}$$
$$\leq 4\sqrt{SAH^2}\sqrt{\sum_{(B,h)} (N_h^{T+1}(B) + 1)} \leq 4\sqrt{H^3 T \cdot N_\varepsilon} + 4N_\varepsilon H^2.$$

By the similar arguments we have
$$\sum_{t=1}^{T}\sum_{h=1}^{H} \frac{\mathbb{1}\{e_{k_h^t(B_h^t)} > 0\}}{e_{k_h^t(B_h^t)}} \leq \mathcal{O}(HN_\varepsilon \log(T)).$$

Using this upper bound, we have for $L = L_r + (1 + L_F)L_V$
$$\sum_{t=1}^{T}\sum_{h=1}^{H} \mathcal{B}_h^t = \mathcal{O}\left(H\beta^{\max}(\delta,\varepsilon)\sum_{t=1}^{T}\sum_{h=1}^{H} \frac{\mathbb{1}\{e_{k_h^t(s_h^t,a_h^t)} > 0\}}{\sqrt{e_{k_h^t(s_h^t,a_h^t)}}}\right)$$
$$+ \mathcal{O}\left(H(\beta^{\max}(\delta,\varepsilon))^4 \sum_{t=1}^{T}\sum_{h=1}^{H} \frac{\mathbb{1}\{e_{k_h^t(s_h^t,a_h^t)} > 0\}}{\sqrt{e_{k_h^t(s_h^t,a_h^t)}}}\right)$$
$$+ \mathcal{O}(LTH\varepsilon)$$
$$\leq \mathcal{O}\left(\sqrt{H^5 T N_\varepsilon \cdot (\beta^{\max}(\delta,\varepsilon))^2} + H^3 N_\varepsilon(\beta^{\max}(\delta,\varepsilon))^4 + LTH\varepsilon\right).$$

Overall, for any fixed $\varepsilon > 0$ we have
$$\mathfrak{R}^T \leq \mathcal{O}\left(\sqrt{H^5 T N_\varepsilon \cdot (\beta^{\max}(\delta,\varepsilon))^2} + H^3 N_\varepsilon(\beta^{\max}(\delta,\varepsilon))^4 + LTH\varepsilon + \sqrt{H^3 T}\right).$$

Next we finally use that $\mathcal{S} \times \mathcal{A}$ have covering dimension $d_c$ that means $N_\varepsilon \leq C_N \cdot \varepsilon^{-d_c}$, thus our regret bound transforms as follows
$$\mathfrak{R}^T \leq \mathcal{O}\left(\sqrt{H^5 T C_N \varepsilon^{-d_c} \cdot (\log(TC_N H/\delta) + d_c \log(1/\varepsilon))^2}\right.$$
$$\left. + H^3 C_N \varepsilon^{-d_c}(\log(TC_N H/\delta) + d_c \log(1/\varepsilon))^4 + LTH\varepsilon\right).$$

By taking $\varepsilon = T^{-1/(d_c+2)}$ we conclude the statement

$\square$

# F  Adaptive `RandQL`

In this section we describe how to improve the dependence in our algorithm from covering dimension to zooming dimension, and describe all required notation.

## F.1  Additional Notation

In this section we introduce an additional notation that is needed for introducing an adaptive version of `RandQL` algorithm for metric spaces.

**Hierarchical partition**   Next, we define all required notation to describe an adaptive partition, as Sinclair et al. [2019, 2023]. Finally, we define the following general framework of hierarchical partition. Instead of balls, we will use a more general notion of regions that will induce a better structure from the computational point of view. We recall for any compact set $A \subseteq \mathcal{S} \times \mathcal{A}$ we call $\mathrm{diam}(A) = \max_{x,y \in A} \rho(x, y)$.

**Definition 6.** A hierarchical partition of $\mathcal{S} \times \mathcal{A}$ of a depth $d > 0$ is a collection of regions $\mathcal{P}_d$ and their centers such that

- Each region $B \in \mathcal{P}_d$ is of the form $\mathcal{S}(B) \times \mathcal{A}(B)$, where $\mathcal{S}(B) \subseteq \mathcal{S}, \mathcal{A}(B) \subseteq \mathcal{A}$;

- $\mathcal{P}_d$ is a cover of $\mathcal{S} \times \mathcal{A}$: $\bigcup_{B \in \mathcal{P}_d} B = \mathcal{S} \times \mathcal{A}$;

- For every $B \in \mathcal{P}_d$, we have $\mathrm{diam}(B) \leq d_{\max} \cdot 2^{-d}$;

- Let $B_1, B_2 \in \mathcal{P}_d$. If $B_1 \neq B_2$ then $\rho(\mathrm{center}(B_1), \mathrm{center}(B_2)) \geq d_{\max} \cdot 2^{-d}$;

- For any $B \in \mathcal{P}_d$, there exists a unique $A \in \mathcal{P}_{d-1}$ (called the parent of $B$) such that $B \subseteq A$.

and, for $d = 0$ we define it as $\mathcal{P}_0 = \{\mathcal{S} \times \mathcal{A}\}$.

We call the tree generated by the structure of $\mathcal{T} = \{\mathcal{P}_d\}_{d \geq 0}$ a tree of this hierarchical partition. The main example of this partition is the dyadic partition of $\mathcal{S} \times \mathcal{A}$ in the case of $\mathcal{S} = [0, 1]^{d_\mathcal{S}}, \mathcal{A} = [0, 1]^{d_\mathcal{A}}$ and the metric induced by the infinity norm $\rho((s, a), (s', a')) = \max\{\|s - s'\|_\infty, \|a - a'\|_\infty\}$. For examples we refer to [Sinclair et al., 2023].

## F.2  Algorithm

In this section we describe two algorithms: `Adaptive-RandQL` which is an adaptive metric counterpart of `RandQL`, and `Adaptive-Staged-RandQL` which is an adaptive metric counterpart of `Staged-RandQL`. First, we start from the notation and algorithmic parts that will be common for both algorithms.

Algorithms maintain an adaptive partition $\mathcal{P}_h^t$ of $\mathcal{S} \times \mathcal{A}$, that is a sub-tree of an (infinite) tree of the hierarchical partition $\mathcal{T} = \{\mathcal{P}_d\}_{d \geq 0}$. We initialize $\mathcal{P}_h^1 = \{\mathcal{P}_0\}$, and then we refine the tree $\mathcal{P}_h^t$ be adding new nodes that corresponding to nodes of $\mathcal{T}$. The leaf nodes of $\mathcal{P}_h^t$ represent the active balls, and for $B \in \mathcal{P}_h^t$ the set of its inactive parent balls is defined as $\{B' \in \mathcal{P}_h^t \mid B \subset B'\}$. For any $B \in \mathcal{P}_h^t$ we define $d(B)$ as a depth of $B$ in the tree under a convention $d(\mathcal{S} \times \mathcal{A}) = 0$.

Additionally, we need to define so-called *selection rule* and *splitting rule*. For any state $s \in \mathcal{S}$ we define the set of all relevant balls as $\mathcal{R}_h^t(s) = \{$ active $b \in \mathcal{P}_h^t \mid (s, a) \in B$ for some $a \in \mathcal{A}\}$. Then for the current state $s_h^t$ we define the current ball as $B_h^t = \arg\max_{B \in \mathcal{R}_h^t(s_h^t)} \overline{Q}_h^t(B)$ and the corresponding action as $a_h^t$. To define the splitting rule we maintain the counters $n_h^t(B)$ for all $B \in \mathcal{P}_h^t$ as a number of visits of a node $B$ and all its parent nodes. Then we will perform splitting of the current ball $B_h^t$ if $\sqrt{d_{\max}^2 / n_h^t(B_h^t)} \leq \mathrm{diam}(B_h^t)$. During splitting, we extend $\mathcal{P}_h^{t+1}$ by its child nodes in the hierarchical partition tree $\mathcal{T}$. For more details we refer to [Sinclair et al., 2023], up to small changes in notation. In particular, their constant $\tilde{C}$ is equal to $d_{\max}$ in our setting to make the construction exactly the same for both `Adaptive-RandQL` and `Adaptive-Staged-RandQL` algorithms.

---

**Algorithm 5** `Adaptive-RandQL`

1: **Input:** ensemble size $J$, number of prior transitions $n_0$, prior reward $r_0$.
2: **Initialize:** $\overline{Q}_h(B) = \widetilde{Q}_h^j(B) = r_0 H$, initialize counters $n_h(s, a) = 0$ for $h, s, a \in [H] \times \mathcal{S} \times \mathcal{A}$.

3: **for** $t \in [T]$ **do**
4:     **for** $h \in [H]$ **do**
5:         Compute $B_h = \arg\max_{B \in \mathcal{R}_h^t(s_h)} \overline{Q}_h(B)$ and play $a_h$ for $(s_h, a_h) = \text{center}(B_h)$;
6:         Observe reward and next state $s_{h+1} \sim p_h(s_h, a_h)$.
7:         Sample $\mathring{w}_j \sim \text{Beta}(n, n_0)$ for $n = n_h(B_h)$.
8:         Compute value $\overline{V}_{h+1}(s_{h+1}) = \max_{B \in \mathcal{R}_h^t(s_{h+1})} \overline{Q}_{h+1}(B)$.
9:         Build targets for all $j \in [J]$

$$\mathring{Q}_h^j = \mathring{w}_j[r_h(s_h, a_h) + \overline{V}_{h+1}(s_{h+1})] + (1 - \mathring{w}_j)r_0 H \,.$$

10:        Sample learning rates $w_j \sim \text{Beta}(H, n)$.
11:        Update ensemble $Q$-functions for all $j \in [J]$

$$\widetilde{Q}_h^j(B_h) := (1 - w_j)\widetilde{Q}_h^j(s_h, a_h) + w_j \mathring{Q}_h^j \,.$$

12:        Update policy $Q$-function $\overline{Q}_h(s_h, a_h) := \max_{j \in [J]} \widetilde{Q}_h^j(s_h, a_h)$.
13:        Update counters $n_h(B_h) := n_h(B_h) + 1$;
14:        If $\sqrt{d_{\max}^2/n_h(B_h)} \leq \text{diam}(B_h)$, then refine partition $B_h$ (see Sinclair et al. [2023]).
15:     **end for**
16: **end for**

---

`Adaptive-RandQL`      This algorithm is an adaptive metric version of `RandQL` algorithm. We recall that for $B \in \mathcal{P}_h^t$ we define $n_h^t(B) = \sum_{i=1}^{t-1} \mathbb{1}\{(B_h^i) \text{ is a parent of } B\}$ is the number of visits of the ball $B$ and its parent balls at step $h$ before episode $t$. We start by initializing the ensemble of Q-values, the policy Q-values, and values to an optimistic value $\widetilde{Q}_h^{t,j}(B) = \overline{Q}_h^1(B) = \overline{V}_h^1(B) = r_0 H$ for all $(j, h) \in [J] \times [H]$ and the unique ball in the partition $B = \mathcal{S} \times \mathcal{A}$ and $r_0 > 0$ some pseudo-rewards.

At episode $t$ we update the ensemble of Q-values as follows, denoting by $n = n_h^t(B)$ the count, $w_{j,n} \sim \text{Beta}(H, n)$ the independent learning rates,

$$\widetilde{Q}_h^{t+1,j}(B) = \begin{cases} (1 - w_{j,n})\widetilde{Q}_h^{t,j}(B) + w_{j,n}\mathring{Q}_h^{t,j}(s_h^t, a_h^t), & B = B_h^t \\ \widetilde{Q}_h^{t,j}(B) & \text{otherwise,} \end{cases}$$

where we defined the target $\mathring{Q}_h^{t,j}(s_h^t, a_h^t)$ as a mixture between the usual target and some prior target with mixture coefficient $\mathring{w}_{n,j} \sim \text{Beta}(n, n_0)$ and $n_0$ the number of prior samples,

$$\mathring{Q}_h^{t,j}(s_h^t, a_h^t) = \mathring{w}_{j,n}[r_h(s_h^t, a_h^t) + \overline{V}_{h+1}^t(s_{h+1}^t)] + (1 - \mathring{w}_{j,n})r_0 H \,.$$

For a discussion on prior re-injection we refer to Appendix B. The value function is computed on-flight by the rule $\overline{V}_h^t(s) = \max_{B \in \mathcal{R}_h^t} \overline{Q}_h^t(B)$.

The policy Q-values are obtained by taking the maximum among the ensemble of Q-values

$$\overline{Q}_h^{t+1}(B) = \max_{j \in [J]} \widetilde{Q}_h^{t+1,j}(B) \,.$$

The policy is then greedy with respect to the policy Q-values and selection rule $(s, \pi_h^{t+1}(s)) = \text{center}(B)$, where $B = \arg\max_{B \in \mathcal{R}_h^{t+1}} \overline{Q}_h^{t+1}(B)$. After the update of Q-values, algorithm verifies the splitting rule. If the splitting rule is triggered, then all new balls are defined by counter and Q-values of its parent. We notice that all Q-values could be efficiently computed on the nodes of the adaptive partition. The complete and detailed description is presented in Algorithm 6.

`Adaptive-Staged-RandQL`      The notation for this algorithm is very close to `Net-Staged-RandQL` and we describe only differences between them. The main difference is a way to compute value $\overline{V}_h^t(s) = \max_{B \in \mathcal{R}_h^t(s)} \overline{Q}_h^t(B)$ and policy $(s, \pi_h^t(s)) = \text{center}(B)$ for $B =$

---

**Algorithm 6** `Adaptive-Staged-RandQL`

---

1: **Input:** inflation coefficient $\kappa$, $J$ ensemble size, number of prior transitions $n_0(k)$, prior reward $r_0$.

2: **Initialize:** $\overline{Q}_h(B) = \widetilde{Q}_h^j(B) = r_0 H$, initialize counters $\widetilde{n}_h(B) = 0$ for $j, h, B \in [J] \times [H] \times \mathcal{N}_\varepsilon$, stage $q_h(B) = 0$.

3: **for** $t \in [T]$ **do**

4:     **for** $h \in [H]$ **do**

5:         Compute $B_h = \arg\max_{B \in \mathcal{R}_h^t(s_h)} \overline{Q}_h(B)$ and play $a_h$ for $(s_h, a_h) = \text{center}(B_h)$;

6:         Observe reward and next state $s_{h+1} \sim p_h(s_h, a_h)$.

7:         Sample learning rates $w_j \sim \text{Beta}(1/\kappa, (\widetilde{n} + n_0(q_h(B_h)))/\kappa)$ for $\widetilde{n} = \widetilde{n}_h(B_h)$.

8:         Compute value $\overline{V}_{h+1}(s_{h+1}) = \max_{B \in \mathcal{R}_h^t(s_{h+1})} \overline{Q}_{h+1}(B)$.

9:         Update temporary $Q$-values for all $j \in [J]$

$$\widetilde{Q}_h^j(B) := (1 - w_j)\widetilde{Q}_h^j(B) + w_j\big(r_h(s_h, a_h) + \overline{V}_{h+1}(s_{h+1})\big).$$

10:         Update counters $\widetilde{n}_h(B_h) := \widetilde{n}_h(B_h) + 1$ and $n_h(B_h) := n_h(B_h) + 1$.

11:         **if** $\widetilde{n}_h(B_h) = \lfloor(1 + 1/H)^q H\rfloor$ for $q = q_h(B_h)$ is the current stage **then**

12:             Update policy $Q$-values $\overline{Q}_h(B_h) := \max_{j \in [J]} \widetilde{Q}_h^j(B_h)$.

13:             Reset temporary $Q$-values $\widetilde{Q}_h^j(B_h) := r_0 H$.

14:             Reset counter $\widetilde{n}_h(B_h) := 0$ and change stage $k_h(B_h) := k_h(B_h) + 1$.

15:         **end if**

16:         If $\sqrt{d_{\max}^2/n_h(B_h)} \leq \text{diam}(B_h)$, then refine partition $B_h$ (see Sinclair et al. [2023]).

17:     **end for**

18: **end for**

---

$\arg\max_{B \in \mathcal{R}_h^t(s)} \overline{Q}_h^t(B)$. Additionally, all counters including temporary will move to the child nodes after splitting, as it performed in `Adaptive-RandQL`. The detailed description is presented in Algorithm 6.

## F.3 Regret Bound

In this section we state the regret bounds for `Adaptive-Staged-RandQL` and derive a proof. The given proof shares a lot of similarities with the proof of `Net-Staged-RandQL` in the first half and to the proof of `Adaptive-QL` by Sinclair et al. [2023] in the second half.

We fix $\delta \in (0, 1)$, and the number of posterior samples

$$J \triangleq \lceil \tilde{c}_J \cdot (\log(2C_N HT/\delta) + d_c \log_2(8T/d_{\max}))\rceil, \tag{15}$$

where $\tilde{c}_J = 1/\log(4/(3 + \Phi(1)))$ and $\Phi(\cdot)$ is the cumulative distribution function (CDF) of a normal distribution.

Additionally we select

$$n_0(k) = \left\lceil \widetilde{n}_0 + \kappa + \frac{L \cdot d_{\max}}{H - 1} \cdot \frac{e_k + \widetilde{n}_0 + \kappa}{\sqrt{He_k - k - H^2}} \right\rceil, \quad \widetilde{n}_0 = (c_0 + 1 + \log_{17/16}(T)) \cdot \kappa$$

where $c_0$ is an absolute constant defined in (5) (see Appendix D.3), $\kappa$ is the posterior inflation coefficient and $L = L_r + (1 + L_F)L_V$ is a constant. Next we restate the regret bound result for `Adaptive-Staged-RandQL` algorithm.

**Theorem** (Restatement of Theorem 3). *Consider a parameter $\delta \in (0, 1)$. Let $\kappa \triangleq 2(\log(8HC_N/\delta) + d_c \log_2(8T/d_{\max}) + 3\log(e\pi(2T + 1)))$, $r_0 \triangleq 2$. Then it holds for `Adaptive-Staged-RandQL`, with probability at least $1 - \delta$,*

$$\mathfrak{R}^T = \widetilde{\mathcal{O}}\left( LH^{3/2} \sum_{h=1}^{H} T^{\frac{d_{z,h}+1}{d_{z,h}+2}} \right),$$

*where $d_{z,h}$ is the step-$h$ zooming dimension and we ignore all multiplicative factors in the covering dimension $d_c$.*

*Proof.* We divide the proof to four main parts, a little bit different proof of `Staged-RandQL` and `Net-Staged-RandQL` since we also need to apply clipping techniques.

**Concentration events** We can define (almost) the same set of events as in Appendix E.3, where union bound over balls is taken over all the hierarchical partition tree up to depth $D$ that we define as $\mathcal{T}_D$.

$$\mathcal{E}^{\star}(\delta) \triangleq \left\{ \forall t \in \mathbb{N}, \forall h \in [H], \forall B \in \mathcal{T}_D, k = k_h^t(B), (s,a) = \text{center}(B) : \right.$$

$$\left. \mathcal{K}_{\inf}\left( \frac{1}{e_k} \sum_{i=1}^{e_k} \delta_{V_{h+1}^{\star}(F_h(s,a,\xi_{h+1}^{\ell^i}))}, p_h V_{h+1}^{\star}(s,a) \right) \leq \frac{\beta^{\star}(\delta, e_k, \varepsilon)}{e_k} \right\},$$

$$\mathcal{E}^B(\delta, T) \triangleq \left\{ \forall t \in [T], \forall h \in [H], \forall B \in \mathcal{T}_D, \forall j \in [J], k = k_h^t(B) : \right.$$

$$\left| \sum_{i=0}^{e_k} (W_{j,e_k,k}^i - \mathbb{E}[W_{j,e_k,k}^i]) \left( r_h(s_h^{\ell^i}, a_h^{\ell^i}) + \overline{V}_{h+1}^{\ell^i}(s_{h+1}^{\ell^i}) \right) \right|$$

$$\left. \leq 60\mathrm{e}^2 \sqrt{\frac{r_0^2 H^2 \kappa \beta^B(\delta, \varepsilon)}{e_k + n_0(k)}} + 1200\mathrm{e}\frac{r_0 H \kappa \log(e_k + n_0(k))(\beta^B(\delta, \varepsilon))^2}{e_k + n_0(k)} \right\},$$

$$\mathcal{E}^{\mathrm{conc}}(\delta, T) \triangleq \left\{ \forall t \in [T], \forall h \in [H], \forall B \in \mathcal{T}_D, k = k_h^t(B) : \right.$$

$$\left. \left| \frac{1}{e_k} \sum_{i=1}^{e_k} V_{h+1}^{\star}(s_{h+1}^{\ell_{k,h}^i(B)}) - p_h V_{h+1}^{\star}(s_h^{\ell_{k,h}^i(B)}, a_h^{\ell_{k,h}^i(B)}) \right| \leq \sqrt{\frac{2r_0^2 H^2 \beta^{\mathrm{conc}}(\delta, \varepsilon)}{e_k}} \right\},$$

$$\mathcal{E}(\delta) \triangleq \left\{ \sum_{t=1}^{T} \sum_{h=1}^{H} (1 + 3/H)^{H-h} \left| p_h[V_{h+1}^{\star} - V_{h+1}^{\pi_t}](s_h^t, a_h^t) - [V_{h+1}^{\star} - V_{h+1}^{\pi_t}](s_{h+1}^t) \right| \right.$$

$$\left. \leq 2\mathrm{e}^3 r_0 H \sqrt{2HT\beta(\delta)}. \right.$$

To apply the union bound argument, we have to bound the size of $\mathcal{T}_D$. First, we notice that relation between centers of balls in each layer $\mathcal{P}_d$ implies that there at least $|\mathcal{P}_d|$ non-intersecting balls of radius $d_{\max} \cdot 2^{-d-2}$. Thus, the size of this sub-tree could be bounded as

$$|\mathcal{T}_D| \leq \sum_{d=0}^{D} N_{d_{\max}2^{-d-2}} \leq C_N \sum_{d=0}^{D} \left(2^{d+2}/d_{\max}\right)^{d_c} \leq (8/d_{\max})^{d_c} C_N \cdot 2^{d_c \cdot D}.$$

using the relation between covering and packing numbers, see e.g. Lemma 4.2.8 by Vershynin [2018]. The only undefined quantity here is $D$, that can be upper-bounded given budget $T$. To do it, we apply Lemma B.2 by Sinclair et al. [2023] for any $B \in \mathcal{P}_h^t$

$$\left( \frac{d_{\max}}{2 \cdot \mathrm{diam}(B)} \right)^2 \leq n_h^t(B) \leq \left( \frac{d_{\max}}{\mathrm{diam}(B)} \right)^2. \tag{16}$$

Our goal is to find a value $D$ such that $\mathcal{P}_h^{T+1} \subseteq \mathcal{T}_D$ for any MDPs and correct interactions. To do it, we notice that it is equivalent to show that $\mathrm{diam}(B) \geq d_{\max} 2^{-D}$, that could be guaranteed since

$$\mathrm{diam}(B) \geq \frac{d_{\max}}{2\sqrt{n_h^{T+1}(B)}} \geq \frac{d_{\max}}{2T},$$

which implies that $D = 1 + \log_2(T)$ is enough. Finally, since for the value of interest

$$\log |\mathcal{T}_D| \leq d_c \log_2(T) + \log C_N + d_c \log(8/d_{\max}),$$

we can define the $\beta$-functions as follows follows

$$\beta^\star(\delta) \triangleq \log(8C_N H/\delta) + d_c \log_2(8T/d_{\max}) + 3\log(\mathrm{e}\pi(2n+1)),$$
$$\beta^B(\delta, T) \triangleq \log(8C_N H/\delta) + d_c \log_2(8T/d_{\max}) + \log(TJ),$$
$$\beta^{\mathrm{conc}}(\delta, T) \triangleq \log(8C_N H/\delta) + d_c \log_2(8T/d_{\max}) + \log(2T),$$
$$\beta(\delta) \triangleq \log(16C_N H/\delta) + d_c \log_2(8T/d_{\max}),$$

and following line-by-line the proof of Lemma 6, for an event $\mathcal{G}(\delta) = \mathcal{E}^\star(\delta) \cap \mathcal{E}^B(\delta, T) \cap \mathcal{E}^{\mathrm{conc}}(\delta, T) \cap \mathcal{E}(\delta)$ we have $\mathbb{P}(\mathcal{G}(\delta)) \geq 1 - \delta/2$.

**Optimism** Next, we state the required analog of Proposition 4. We can show that with probability at least $1 - \delta/2$ on the event $\mathcal{E}^\star(\delta)$ the following event

$$\mathcal{E}_{\mathrm{anticonc}} \triangleq \Bigg\{ \forall t \in [T]\, \forall h \in [H]\, \forall B \in \mathcal{T}_D : \text{for } k = k_h^t(B), (s, a) = \mathrm{center}(B) :$$

$$\max_{j \in [J]} \Bigg\{ W_{j,e_k,k}^0 r_0(H-1) + \sum_{i=1}^{e_k} W_{j,e_k,k}^i V_{h+1}^\star(F_h(s, a, \xi_h^{\ell^i})) \Bigg\} \geq p_h V_{h+1}^\star(s,a) + L \cdot \mathrm{diam}(B_h^t) \Bigg\}$$

under the choice $J = \lceil \tilde{c}_J \cdot (\log(2HT/\delta) + \log(|\mathcal{T}_D|)) \rceil$, $\kappa = 2\beta^\star(\delta, T)$, $r_0 = 2$, and a prior count

$$n_0(k) = \lceil \tilde{n}_0 + \kappa + \frac{L \cdot d_{\max}}{H-1} \cdot \frac{e_k + \tilde{n}_0 + \kappa}{\sqrt{He_k - k - H^2}} \rceil$$

dependent on the stage $k$, where $\tilde{n}_0 = (c_0 + 1 + \log_{17/16}(T)) \cdot \kappa$, $L = L_r + L_V(1 + L_F)$. In particular, the proof exactly the same as the proof of Proposition 4 for $\varepsilon$ dependent on $k$.

At the same time, it is possible to show that $\mathcal{E}_{\mathrm{anticonc}}$ implies

$$\mathcal{E}_{\mathrm{opt}} \triangleq \Big\{ \forall t \in [T], h \in [H], \forall B \in \mathcal{P}_h^t, \forall (s,a) \in B : \overline{Q}_h^t(B) \geq Q_h^\star(s,a) \Big\}. \qquad (17)$$

Indeed, in the proof of Proposition 5 we actively uses the bound $\rho((s_h^{\ell^i}, a_h^{\ell^i}), (s,a)) \leq \varepsilon$. In the adaptive setting, we have to, at first, use an upper bound $\rho((s_h^{\ell^i}, a_h^{\ell^i}), (s,a)) \leq \mathrm{diam}(B_h^{\ell^i})$ by a construction $B \subseteq B_h^{\ell^i}$, and then apply Lemma B.2 by Sinclair et al. [2023] by defining an upper bound

$$\mathrm{diam}(B_h^{\ell^i}) \leq \frac{d_{\max}}{\sqrt{n_h^{\ell^i}(B_h^{\ell^i})}} \leq \frac{d_{\max}}{\sqrt{\sum_{i=0}^{k-1} e_i}} \leq \frac{d_{\max}}{\sqrt{H\sum_{i=0}^{k-1}(1+1/H)^i - k}} \leq \frac{d_{\max}}{\sqrt{He_k - k - H^2}}$$

for $k = k_h^t(B)$ for a particular ball $B \in \mathcal{P}_h^t$ in the case $He_k - k - H^2 \geq 0$.

By combining event $\mathcal{E}_{\mathrm{opt}}$ and the event $\mathcal{E}^B(\delta)$ we can prove the same statement as Corollary 2.

Let $t \in [T], h \in [H], B \in \mathcal{P}_h^t$. Define $k = k_h^t(B)$ and let $\ell^1 < \ldots < \ell^{e_k}$ be a excursions of $(B, h)$ till the end of the previous stage. Then on the event $\mathcal{G}'(\delta) = \mathcal{G}(\delta) \cap \mathcal{E}_{\mathrm{opt}}$ the following bound holds for $k \geq 0$ and for any $(s,a) \in B$

$$0 \leq \overline{Q}_h^t(B) - Q_h^\star(s,a) \leq H\mathbb{1}\{He_k/2 \leq k+H^2\} + \frac{1}{e_k}\sum_{i=1}^{e_k} [\overline{V}_{h+1}^{\ell^i}(s_{h+1}^{\ell^i}) - V_{h+1}^\star(s_{h+1}^{\ell^i})] + \mathcal{B}_h^t, \ (18)$$

where

$$\mathcal{B}_h^t = 121\mathrm{e}^2 \cdot \sqrt{\frac{H^2(\beta^{\max}(\delta, T))^2}{e_k}} + 2401\mathrm{e} \cdot \frac{H(\beta^{\max}(\delta, T))^4}{e_k} + \frac{5L \cdot d_{\max}}{\sqrt{He_k}} \qquad (19)$$

where $k = k_h^t(B_h^t)$ and $\beta^{\max}(\delta, T) = \max\{\beta^\star(\delta, T), \beta^B(\delta), \beta^{\mathrm{conc}}(\delta), \beta(\delta)\}$. Also we can express this bound in terms of a diameter of $B_h^t$ as follows

$$\mathrm{diam}(B_h^t) \geq \frac{d_{\max}}{2\sqrt{n_h^t(B_h^t)}} \geq \frac{d_{\max}}{2\sqrt{\sum_{i=0}^k e_i}} \geq \frac{d_{\max}}{2\sqrt{H\sum_{i=0}^k(1+1/H)^i}} \geq \frac{d_{\max}}{2\sqrt{H}}$$

$$\geq \frac{d_{\max}}{2\sqrt{H^2(1+1/H)^{k+1}}} \geq \frac{d_{\max}}{2\sqrt{2He_k}},$$

thus

$$\frac{1}{\sqrt{He_k}} \leq \frac{3\mathrm{diam}(B_h^t)}{d_{\max}},$$

and we have

$$\begin{aligned}
\mathcal{B}_h^t &\leq 7566\mathrm{e}^2 H^{3/2}(\beta^{\max}(\delta,T))^4 \mathrm{diam}(B_h^t)/d_{\max} + 15L\mathrm{diam}(B_h^t) \\
&\leq \rho(H,\delta,L) \cdot \mathrm{diam}(B_h^t),
\end{aligned} \tag{20}$$

where we define $\rho(H,\delta,L) \triangleq 7566\mathrm{e}^2 H^{3/2}(\beta^{\max}(\delta,T))^4/d_{\max} + 15L$.

As a additional corollary, we have for all $t \in [T], h \in [H]$

$$\overline{V}_h^t(s) = \max_{B \in \mathcal{R}_h^t(s)} \overline{Q}_h^t(B) = \overline{Q}_h^t(B^\star) \geq Q_h^\star(s,\pi^\star(s)) = V_h^\star(s), \tag{21}$$

where $B^\star$ is a ball that contains a pair $(s,\pi^\star(s))$.

This upper and lower bound have the similar structure as Lemma D.2 by Sinclair et al. [2023] and the rest of the proof directly follows [Sinclair et al., 2023].

**Clipping techniques**  Next we introduce the required clipping techniques developed by Simchowitz and Jamieson [2019], Cao and Krishnamurthy [2020]. Definition 2 introduces the quantity $\mathrm{gap}_h(s,a) = V_h^\star(s) - Q_h^\star(s,a)$, and for any compact set $B \subseteq \mathcal{S} \times \mathcal{A}$ we define $\mathrm{gap}_h(B) = \min_{(s,a) \in B} \mathrm{gap}_h(s,a)$. Finally, we define clipping operator for any $\mu,\nu \in \mathbb{R}$

$$\mathrm{clip}(\mu|\nu) = \mu\mathbb{1}\{\mu \leq \nu\}. \tag{22}$$

In particular, this operator satisfies the following important property

**Lemma 8** (Lemma E.2. of Sinclair et al. [2023]). *Suppose that* $\mathrm{gap}_h(B) \leq \psi \leq \mu_1 + \mu_2$ *for any* $\psi, \mu_1, \mu_2$. *Then*

$$\psi \leq \mathrm{clip}\left[\mu_1 \left| \frac{\mathrm{gap}_h(B)}{H+1} \right.\right] + \left(1 + \frac{1}{H}\right)\mu_2$$

Now we apply this lemma to our update rules, producing a result similar to Lemma E.3 of Sinclair et al. [2023]. We notice that

$$\begin{aligned}
\mathrm{gap}_h(B_h^t) &\leq \mathrm{gap}_h(s_h^t, a_h^t) = V_h^\star(s_h^t) - Q_h^\star(s_h^t, a_h^t) \\
&\leq \overline{V}_h^t(s_h^t) - Q_h^\star(s_h^t, a_h^t) = \overline{Q}_h^t(B_h^t) - Q_h^\star(s_h^t, a_h^t).
\end{aligned}$$

Thus, denoting $\psi = \overline{Q}_h^t(B_h^t) - Q_h^\star(s_h^t, a_h^t)$ and, by (18),

$$\mu_1 = H\mathbb{1}\{He_{k_h^t}/2 > k_h^t + H^2\} + \mathcal{B}_h^t, \quad \mu_2 = \frac{1}{e_k}\sum_{i=1}^{e_k}[\overline{V}_{h+1}^{\ell^i}(s_{h+1}^{\ell^i}) - V_{h+1}^\star(s_{h+1}^{\ell^i})]$$

we apply Lemma 8 and obtain

$$\begin{aligned}
\overline{V}_h^t(s_h^t) - Q_h^\star(s_h^t, a_h^t) &\leq \mathrm{clip}\left[H\mathbb{1}\{He_{k_h^t}/2 \leq k_h^t + H^2\} + \mathcal{B}_h^t \left| \frac{\mathrm{gap}_h(B_h^t)}{H+1}\right.\right] \\
&\quad + \left(1 + \frac{1}{H}\right)\frac{1}{e_k}\sum_{i=1}^{e_k}[\overline{V}_{h+1}^{\ell^i}(s_{h+1}^{\ell^i}) - V_{h+1}^\star(s_{h+1}^{\ell^i})]
\end{aligned} \tag{23}$$

for $k_h^t = k_h^t(B_h^t)$ and $\mathcal{B}_h^t$ defined in (19).

**Regret decomposition**  The rest of the analysis we preform conditionally on event $\mathcal{G}'(\delta) = \mathcal{G}(\delta) \cap \mathcal{E}_{\mathrm{opt}}$ that holds with probability at least $1 - \delta$.

By defining $\delta_h^t = \overline{V}_h^t(s_h^t) - V^{\pi^t}(s_h^t)$ and $\zeta_h^t = \overline{V}_h^t(s_h^t) - V_h^\star(s_h^t)$ we have

$$\mathfrak{R}^T = \sum_{t=1}^T V_1^\star(s_1^t) - V_1^{\pi^t}(s_1^t) \leq \sum_{t=1}^T \delta_1^t,$$

and, at the same time, by Bellman equations

$$\delta_h^t = \overline{V}_h^t(s_h^t) - Q^{\pi_h^t}(s_h^t, a_h^t) = \overline{V}_h^t(s_h^t) - Q_h^\star(s_h^t, a_h^t) + Q_h^\star(s_h^t, a_h^t) - Q^{\pi^t}(s_h^t, a_h^t)$$
$$= \overline{V}_h^t(s_h^t) - Q_h^\star(s_h^t, a_h^t) + V_{h+1}^\star(s_{h+1}^t) - V_h^{\pi^t}(s_{h+1}^t) + \xi_h^t$$
$$= \overline{V}_h^t(s_h^t) - Q_h^\star(s_h^t, a_h^t) + \delta_{h+1}^t - \zeta_{h+1}^t + \xi_h^t,$$

where $\xi_h^t = p_h[V_{h+1}^\star - V_{h+1}^{\pi^t}](s_h^t, a_h^t) - [V_{h+1}^\star - V_{h+1}^{\pi^t}](s_{h+1}^t)$ is a martingale-difference sequence. By (23) we have

$$\sum_{t=1}^T \delta_h^t = \sum_{t=1}^T \overline{V}_h^t(s_h^t) - Q_h^\star(s_h^t, a_h^t) + \delta_{h+1}^t - \zeta_{h+1}^t + \xi_h^t$$
$$\leq \left(1 + \frac{1}{H}\right) \sum_{t=1}^T \frac{1}{e_{k_h^t}} \sum_{i=1}^{e_{k_h^t}} \zeta_{h+1}^{\ell_{k_h^t}^i} + \sum_{t=1}^T \delta_{h+1}^t - \sum_{t=1}^T \zeta_{h+1}^t + \sum_{t=1}^T \xi_h^t$$
$$+ \sum_{t=1}^T \mathrm{clip}\left[H\mathbb{1}\{He_{k_h^t}/2 > k_h^t + H^2\} + \mathcal{B}_h^t(k_h^t)\big| \frac{\mathrm{gap}_h(B_h^t)}{H+1}\right]$$

where $k_h^t = k_h^t(B_h^t)$. Repeating argument of Lemma 5 and Zhang et al. [2020]

$$\left(1 + \frac{1}{H}\right) \sum_{t=1}^T \frac{1}{e_{k_h^t}} \sum_{i=1}^{e_{k_h^t}} \zeta_{h+1}^{\ell_{k_h^t}^i} \leq \left(1 + \frac{1}{H}\right)^2 \sum_{t=1}^T \zeta_{h+1}^t \leq \left(1 + \frac{3}{H}\right) \sum_{t=1}^T \zeta_{h+1}^t.$$

Using an upper bound $\zeta_h^t \leq \delta_h^t$ we have for any $h \geq 1$

$$\sum_{t=1}^T \delta_h^t \leq \left(1 + \frac{3}{H}\right) \sum_{t=1}^T \delta_{h+1}^t + \sum_{t=1}^T \xi_h^t$$
$$+ \sum_{t=1}^T \mathrm{clip}\left[H\mathbb{1}\{He_{k_h^t}/2 \leq k_h^t + H^2\} + \mathcal{B}_h^t\big| \frac{\mathrm{gap}_h(B_h^t)}{H+1}\right],$$

and, rolling out starting with $h = 1$ we have the following regret decomposition

$$\mathfrak{R}^T \leq \mathrm{e}^3 \sum_{t=1}^T \sum_{h=1}^H H\mathbb{1}\{He_{k_h^t}/2 \leq k_h^t + H^2\} \qquad = (\mathbf{A})$$
$$+ \mathrm{e}^3 \sum_{t=1}^T \sum_{h=1}^H \mathrm{clip}\left[\mathcal{B}_h^t\big| \frac{\mathrm{gap}_h(B_h^t)}{H+1}\right] \qquad = (\mathbf{B})$$
$$+ \sum_{t=1}^T \sum_{h=1}^H (1 + 3/H)^{H-h} \xi_h^t. \qquad = (\mathbf{C})$$

**Term (A)** For this term we notice that for any fixed $h$ the following event

$$He_{k_h^t} \leq 2(k_h^t + H^2) \iff H\lfloor H(1 + 1/H)^{k_h^t}\rfloor \leq 2(k_h^t + H^2),$$

that is guaranteed if

$$(1 + 1/H)^{k_h^t} \leq 2T + 3 \iff k_h^t \log(1 + 1/H) \leq \log(2T/H^2 + 3).$$

Thus, indicator can be equal to 1 no more than $H\log(2T + 3)$ times for any $t \in [T]$. As a result,

$$(\mathbf{A}) \leq \mathrm{e}^2 H^3 \log(2T + 3).$$

**Term (B)** Let us rewrite this term using a definition of clipping operator and use the definition of near-optimal set (see Definition 3)

$$(\mathbf{B}) = e^3 \sum_{t=1}^{T} \sum_{h=1}^{H} \mathcal{B}_h^t \mathbb{1}\left\{(H+1)\mathcal{B}_h^t \geq \text{gap}_h(B_h^t)\right\} \leq e^3 \sum_{t=1}^{T} \sum_{h=1}^{H} \mathcal{B}_h^t \mathbb{1}\{\text{center}(B_h^t) \in Z_h^{\mathcal{B}_h^t}\}.$$

Next we consider the summation for a fixed $h$. Here we follow Theorem F.3 by Sinclair et al. [2023] and obtain

$$\sum_{t=1}^{T} \mathcal{B}_h^t \mathbb{1}\{\text{center}(B_h^t) \in Z_h^{\mathcal{B}_h^t}\} = \sum_{r} \sum_{B:\text{diam}(B)=r} \sum_{t:B_h^t=B} \mathcal{B}_h^t \mathbb{1}\{\text{center}(B) \in Z_h^{\mathcal{B}_h^t}\},$$

where we applied an additional rescaling by a function $\rho$ defined in (20).

Next we fix a constant $r_0 > 0$ and break a summation into two parts: $r \geq r_0$ and $r \leq r_0$.

1. Case $r \leq r_0$. In this situation we have can apply (20)

$$\sum_{r \leq r_0} \sum_{B:\text{diam}(B)=r} \sum_{t:B_h^t=B} \mathcal{B}_h^t \mathbb{1}\{\text{center}(B) \in Z_h^{\mathcal{B}_h^t}\}$$
$$= \mathcal{O}(Tr_0\rho(H,\delta,L)).$$

2. Case $r \geq r_0$. In this situation we also apply (20) under the indicator function

$$\sum_{r \geq r_0} \sum_{B:\text{diam}(B)=r} \sum_{t:B_h^t=B} \mathcal{B}_h^t \mathbb{1}\{\text{center}(B) \in Z_h^{\mathcal{B}_h^t}\}$$
$$\leq \sum_{r \geq r_0} \sum_{B:\text{diam}(B)=r\cdot\rho(H,\delta,L)} \mathbb{1}\{\text{center}(B) \in Z_h^{\rho(H,\delta,L)\cdot r}\} \sum_{t:B_h^t=B} \mathcal{B}_h^t.$$

To upper bound the last sum we repeat the argument of (14) and apply (16), using the fact that $\text{diam}(B) = r \cdot \rho(H,\delta,L)$

$$\sum_{t:B_h^t=B} \frac{1}{\sqrt{e_k}} \leq \sum_{k=0}^{k_h^T(B)} \frac{e_{k+1}}{\sqrt{e_k}} \leq 4H\sqrt{e^{k_h^T(B)+1}}$$
$$\leq 4\sqrt{H(n_h^{T+1}(B)+1)} \leq 4\sqrt{2H} \cdot \frac{d_{\max}}{\text{diam}(B)} = \frac{\sqrt{32H} \cdot d_{\max}}{r}.$$

As a result, we have by (19)

$$\sum_{t:B_h^t=B} \mathcal{B}_h^t \leq \frac{\sqrt{32H} \cdot d_{\max}}{r} \cdot \left(2522e^2 H(\beta^{\max}(\delta,T))^4 + 5Ld_{\max}/\sqrt{H}\right)$$

and

$$\sum_{r \geq r_0} \sum_{B:\text{diam}(B)=r} \sum_{t:B_h^t=B} \mathcal{B}_h^t \mathbb{1}\{\text{center}(B) \in Z_h^{\mathcal{B}_h^t}\}$$
$$= \mathcal{O}\left(\sum_{r \geq r_0} N_r(Z_h^{\rho(H,\delta,L)\cdot r}) \cdot \frac{H^{3/2} d_{\max}(\beta^{\max}(\delta,T))^4 + Ld_{\max}^2}{r}\right).$$

Finally, by an arbitrary choice of $r_0$ and a definition of zooming dimension with a scaling $\rho = \rho(H,\delta,L)$ (Definition 4)

$$(\mathbf{B}) = \mathcal{O}\left((H^{3/2} d_{\max}(\beta^{\max}(\delta,T))^4 + Ld_{\max}^2) \cdot \sum_{h=1}^{H} \inf_{\tilde{r}_0}\left\{Tr_0 + \sum_{r \geq r_0} \frac{C_{N,h}}{\tilde{r}^{d_{z,h}+1}}\right\}\right).$$

**Term (C)**   For this term we just apply definition of the main event $\mathcal{G}(\delta) \supseteq \mathcal{E}(\delta)$ and obtain

$$(\mathbf{C}) = \mathcal{O}\left(\sqrt{H^3 T \beta^{\max}(\delta, T)}\right).$$

**Final regret bound**   First, we notice that $\beta^{\max(\delta,T)} = \widetilde{\mathcal{O}}(d_c)$, therefore we have

$$\mathfrak{R}^T = \widetilde{\mathcal{O}}\left( H^3 d_c + (H^{3/2} d_c^4 + L) \sum_{h=1}^{H} \inf_{r_0 > 0} \left\{ T r_0 + \sum_{r \geq r_0} \frac{C_{N,h}}{r^{d_{z,h}+1}} \right\} + \sqrt{H^3 T d_c} \right).$$

Taking $r_0 = K^{-d_{z,h}+1/2}$ for each $h$ and summing the geometric series we conclude the statement.

$\square$

# G Deviation and Anti-Concentration Inequalities

## G.1 Deviation inequality for $\mathcal{K}_{\mathbf{inf}}$

For a measure $\nu \in \mathcal{P}([0,b])$ supported on a segment $[0,b]$ (equipped with a Borel $\sigma$-algebra) and a number $\mu \in [0,b]$ we recall the definition of the minimum Kullback-Leibler divergence

$$\mathcal{K}_{\text{inf}}(\nu, \mu) \triangleq \inf\{\text{KL}(\nu, \eta) : \eta \in \mathcal{P}([0,b]), \nu \ll \eta, \mathbb{E}_{X \sim \eta}[X] \geq \mu\}.$$

As the Kullback-Leibler divergence this quantity admits a variational formula.

**Lemma 9** (Lemma 18 by Garivier et al., 2018)**.** *For all $\nu \in \mathcal{P}([0,b])$, $u \in [0,b)$,*

$$\mathcal{K}_{inf}(\nu, u) = \max_{\lambda \in [0,1]} \mathbb{E}_{X \sim \nu}\left[\log\left(1 - \lambda \frac{X-u}{b-u}\right)\right],$$

*moreover if we denote by $\lambda^\star$ the value at which the above maximum is reached, then*

$$\mathbb{E}_{X \sim \nu}\left[\frac{1}{1 - \lambda^\star \frac{X-u}{b-u}}\right] \leq 1.$$

**Remark 2.** Contrary to Garivier et al. [2018] we allow that $u = 0$ but in this case Lemma 9 is trivially true, indeed

$$\mathcal{K}_{\text{inf}}(\nu, 0) = 0 = \max_{\lambda \in [0,1]} \mathbb{E}_{X \sim \nu}\left[\log\left(1 - \lambda \frac{X}{b}\right)\right].$$

Let $(X_t)_{t \in \mathbb{N}^\star}$ be i.i.d. samples from a measure $\nu$ supported on $[0,b]$. We denote by $\widehat{\nu}_n \in \mathcal{P}([0,b])$ the empirical measure $\widehat{\nu}_n = \sum_{i=1}^{n} \delta_{X_i}$, where $\delta_{X_i}$ is a Dirac measure on $X_i \in [0,b]$.

We are now ready to state the deviation inequality for the $\mathcal{K}_{\text{inf}}$ by Tiapkin et al. [2022b] which is a self-normalized version of Proposition 13 by Garivier et al. [2018]. Notice that this inequality is stated in terms of slightly less general definition of $\mathcal{K}_{\text{inf}}$, however, the proof remains completely the same.

**Theorem 4.** *For all $\nu \in \mathcal{P}([0,b])$ and for all $\delta \in [0,1]$,*

$$\mathbb{P}\big(\exists n \in \mathbb{N}^\star, \, n\, \mathcal{K}_{inf}(\widehat{\nu}_n, \mathbb{E}_{X \sim \nu}[X]) > \log(1/\delta) + 3\log(e\pi(1 + 2n))\big) \leq \delta.$$

## G.2 Anti-concentration Inequality for Dirichlet Weighted Sums

In this section we state anti-concentration inequality by Tiapkin et al. [2022a] in terms of slightly different definition of $\mathcal{K}_{\text{inf}}$.

$$c_0(\varepsilon) = \left(\frac{4}{\sqrt{\log(17/16)}} + 8 + \frac{49 \cdot 4\sqrt{6}}{9}\right)^2 \frac{2}{\pi \cdot \varepsilon^2} + \log_{17/16}\left(\frac{5}{32 \cdot \varepsilon^2}\right). \tag{24}$$

**Theorem 5** (Lower bound)**.** *For any $\alpha = (\alpha_0 + 1, \alpha_1, \ldots, \alpha_m) \in \mathbb{R}_{++}^{m+1}$ define $\overline{p} \in \Delta_m$ such that $\overline{p}(\ell) = \alpha_\ell/\overline{\alpha}, \ell = 0, \ldots, m$, where $\overline{\alpha} = \sum_{j=0}^{m} \alpha_j$. Let $\varepsilon \in (0,1)$. Assume that $\alpha_0 \geq c_0(\varepsilon) + \log_{17/16}(\overline{\alpha})$ for $c_0(\varepsilon)$ defined in (24), and $\overline{\alpha} \geq 2\alpha_0$. Then for any $f: \{0, \ldots, m\} \to [0, b_0]$ such that $f(0) = b_0$, $f(j) \leq b < b_0/2, j \in \{1, \ldots, m\}$ and $\mu \in (\overline{p}f, b_0)$*

$$\mathbb{P}_{w \sim \text{Dir}(\alpha)}[wf \geq \mu] \geq (1 - \varepsilon)\mathbb{P}_{g \sim \mathcal{N}(0,1)}\left[g \geq \sqrt{2\overline{\alpha}\, \mathcal{K}_{inf}\left(\sum_{i=0}^{m} \overline{p}(i) \cdot \delta_{f(i)}, \mu\right)}\right].$$

Next we formulate a simple corollary of Theorem 5, that slightly relaxes assumptions of this theorem under assumption $\mu < b \leq b_0/2$.

**Lemma 10.** *For any $\alpha = (\alpha_0 + 1, \alpha_1, \ldots, \alpha_m) \in \mathbb{R}_{++}^{m+1}$ define $\overline{p} \in \Delta_m$ such that $\overline{p}(\ell) = \alpha_\ell/\overline{\alpha}, \ell = 0, \ldots, m$, where $\overline{\alpha} = \sum_{j=0}^m \alpha_j$. Also define a measure $\overline{\nu} = \sum_{i=0}^m \overline{p}(i) \cdot \delta_{f(i)}$.*

*Let $\varepsilon \in (0,1)$. Assume that $\alpha_0 \geq c_0(\varepsilon) + \log_{17/16}(2(\overline{\alpha} - \alpha_0))$ for $c_0(\varepsilon)$ defined in (24). The for any $f \colon \{0, \ldots, m\} \to [0, b_0]$ such that $f(0) = b_0, f(j) \leq b \leq b_0/2, j \in [m]$, and any $\mu \in (0, b)$*

$$\mathbb{P}_{w \sim \mathrm{Dir}(\alpha)}[wf \geq \mu] \geq (1 - \varepsilon)\mathbb{P}_{g \sim \mathcal{N}(0,1)}\left[g \geq \sqrt{2\overline{\alpha}\,\mathcal{K}_{inf}(\overline{\nu}, \mu)}\right].$$

*Proof.* Assume that assumption $\overline{\alpha} \geq 2\alpha_0$ holds.

Then we show that the Theorem 5 also holds for $\mu \leq \overline{p}f$. First, we notice that for any $\gamma > 0$

$$\mathbb{P}_{w \sim \mathrm{Dir}(\alpha)}[wf \geq \mu] \geq \mathbb{P}_{w \sim \mathrm{Dir}(\alpha)}[wf \geq \overline{p}f + \gamma] \geq (1 - \varepsilon)\mathbb{P}_{g \sim \mathcal{N}(0,1)}\left[g \geq \sqrt{2\overline{\alpha}\,\mathcal{K}_{\inf}(\overline{\nu}, \overline{p}f + \gamma)}\right].$$

By continuity of $\mathcal{K}_{\inf}$ in its second argument (see Theorem 7 by Honda and Takemura [2010]) we can tend $\gamma$ to zero, and then use an equality $\mathcal{K}_{\inf}(\overline{\nu}, \overline{p}f) = \mathcal{K}_{\inf}(\overline{\nu}, \mu) = 0$.

Next, assume $\overline{\alpha} \leq 2\alpha_0$. In this case we have $\overline{p}f \geq b$, thus for any $0 \leq \mu \leq b$

$$\mathbb{P}_{w \sim \mathrm{Dir}(\alpha)}[wf \geq \mu] \geq \mathbb{P}_{\xi \sim \mathrm{Beta}(\alpha_0 + 1, \overline{\alpha} - \alpha_0)}[b_0\xi \geq \mu] \geq \mathbb{P}_{\xi \sim \mathrm{Beta}(\alpha_0 + 1, \overline{\alpha} - \alpha_0)}\left[\xi \geq \frac{1}{2}\right],$$

where we first apply a lower bound $f(j) \geq 0$ for all $j > 0$ and $f(0) = b_0$, and second apply a bound $\mu \leq b_0/2$. Here we may apply the result of Alfers and Dinges [1984, Theorem 1.2"] and obtain the following lower bound

$$\mathbb{P}_{w \sim \mathrm{Dir}(\alpha)}[wf \geq \mu] \geq \Phi\left(-\mathrm{sign}(\alpha_0/\overline{\alpha} - 1/2) \cdot \sqrt{2\overline{\alpha}\,\mathrm{kl}(\alpha_0/\overline{\alpha}, 1/2)}\right) \geq (1 - \varepsilon)\mathbb{P}_{g \sim \mathcal{N}(0,1)}[g \geq 0]$$

where we used $\alpha_0/\overline{\alpha} > 1/2$. $\square$

### G.3  Rosenthal-type inequality

In this section we state Rosenthal-type inequality for martingale differences by [Pinelis, 1994, Theorem 4.1]. The exact constants could be derived from the proof.

**Theorem 6.** *Let $X_1, \ldots, X_n$ be a martingale-difference sequence adapted to a filtration $\{\mathcal{F}_i\}_{i=1,\ldots,n}$: $\mathbb{E}[X_i|\mathcal{F}_i] = 0$. Define $\mathcal{V}_i = \mathbb{E}[X_i^2|\mathcal{F}_{i-1}]$. Then for any $p \geq 2$ the following holds*

$$\mathbb{E}^{1/p}\left[\left|\sum_{i=1}^n X_i\right|^p\right] \leq C_1 p^{1/2}\mathbb{E}^{1/p}\left[\left|\sum_{i=1}^n \mathcal{V}_i\right|^{p/2}\right] + 2C_2 p\mathbb{E}^{1/p}\left[\max_{i \in [n]}|X_i|^p\right],$$

*where $C_1 = 60\mathrm{e}, C_2 = 60$.*

Additionally, we need some additional lemma to use this inequality in our setting.

**Definition 7.** *A random variable $X$ is called sub-exponential with parameters $(\sigma^2, b)$ if the following tail condition holds for any $t > 0$*

$$\mathbb{P}[|X - \mathbb{E}[X]| \geq t] \leq 2\exp\left(-\frac{t^2}{2\sigma^2 + 2bt}\right).$$

By Theorem 1 of Skorski [2023] we have for any $\xi \in B(\alpha, \beta)$ with $\beta \geq \alpha$ and any $t > 0$

$$\mathbb{P}[|\xi - \mathbb{E}[\xi]| \geq t] \leq 2\exp\left(-\frac{t^2}{2(v + ct/3)}\right),$$

where

$$v = \frac{\alpha\beta}{(\alpha + \beta)^2(\alpha + \beta + 1)} \leq \frac{\alpha}{(\alpha + \beta)^2}, \quad c = \frac{2(\beta - \alpha)}{(\alpha + \beta)(\alpha + \beta + 2)} \leq \frac{2}{\alpha + \beta},$$

so $\xi$ is $(\alpha/(\alpha + \beta)^2, 2/(3(\alpha + \beta)))$ sub-exponential.

**Lemma 11.** *Let $X_1, \ldots, X_n$ be a sequence of centred $(\sigma^2, b)$ sub-exponential random variables, not necessarily independent. Then for any $p \geq 2$*

$$\mathbb{E}\left[\max_{\ell \in [n]} |X_\ell|^p\right] \leq \max\{\sqrt{8\sigma^2 \log n}, 8b \log n\}^p + \mathrm{e}(2\sigma)^p p^{p/2} + 2\mathrm{e}(8b)^p p^p.$$

*Proof.* By Fubini theorem we have for any $\eta \geq 0$: $\mathbb{E}[\eta^p] = p \int_0^\infty u^{p-1} \mathbb{P}[\eta \geq u] \mathrm{d}u$, thus for any $a > 0$ the following holds

$$\mathbb{E}\left[\max_{\ell \in [n]} |X_\ell|^p\right] = p \int_0^\infty u^{p-1} \mathbb{P}\left[\max_{\ell \in [n]} |X_\ell - \mathbb{E}[X_\ell]| \geq u\right] \mathrm{d}u$$

$$\leq a^p + p \int_a^\infty u^{p-1} \mathbb{P}[\exists \ell \in [n] : |X_\ell| \geq u] \mathrm{d}u$$

$$\leq a^p + 2p \int_a^\infty u^{p-1} n \exp\left(-\frac{u^2}{2(\sigma^2 + bu)}\right) \mathrm{d}u.$$

By selecting $a = \max\{\sqrt{8\sigma^2 \log n}, 8b \log n\}$ we have

$$n \exp\left(-\frac{u^2}{2(\sigma^2 + bu)}\right) \leq \exp\left(-\frac{u^2}{4(\sigma^2 + bu)}\right) \leq \exp\left(-\frac{u^2}{8\sigma^2}\right) + \exp\left(-\frac{u}{8b}\right)$$

for any $u \geq a$, thus

$$\mathbb{E}\left[\max_{\ell \in [n]} |X_\ell|^p\right] \leq \max\{\sqrt{8\sigma^2 \log n}, 8b \log n\}^p$$

$$+ 2p \int_a^\infty u^{p-1} \exp\left(-\frac{u^2}{8\sigma^2}\right) \mathrm{d}u + 2p \int_a^\infty u^{p-1} \exp\left(-\frac{u}{8b}\right) \mathrm{d}u$$

$$\leq \max\{\sqrt{8\sigma^2 \log n}, 8b \log n\}^p + p(2\sqrt{2}\sigma)^p \Gamma(p/2) + 2p(8b)^p \Gamma(p).$$

By the bounds on Gamma-function we have

$$p\Gamma(p/2) = \Gamma(p/2 + 1) \leq (p+1)^{(p+1)/2} 2^{-(p+1)/2} \mathrm{e}^{1-p/2} \leq \mathrm{e} p^{p/2} 2^{-p/2}$$

and $p\Gamma(p) = \Gamma(p+1) \leq (p+1/2)^{p+1/2} \mathrm{e}^{1-p} \leq \mathrm{e} p^p$ (see Guo et al. [2007]), thus

$$\mathbb{E}\left[\max_{\ell \in [n]} |X_\ell|^p\right] \leq \max\{\sqrt{8\sigma^2 \log n}, 8b \log n\}^p + \mathrm{e}(2\sigma)^p p^{p/2} + 2\mathrm{e}(8b)^p p^p.$$

$\square$

**Proposition 7.** *Let $W_1, \ldots, W_n$ be a sequence of Beta-distributed random variables $W_i \sim \mathrm{Beta}(1/\kappa, (n-1)/\kappa)$ for $\kappa > 0$. Let $\{\mathcal{F}_i\}_{i \in [n]}$ be a filtration such that $W_i$ is independent from $\mathcal{F}_{i-1}$ : $\mathbb{E}[W_i | \mathcal{F}_{i-1}] = \mathbb{E}[W_i]$, and $X_1, \ldots, X_n$ be a sequence of bounded predictable random variables: $\mathbb{E}[X_i | \mathcal{F}_{i-1}] = X_i, |X_i| \leq B$.*

Then with probability at least $1 - \delta$ the following holds

$$\left|\sum_{i=1}^n W_i X_i - \frac{1}{n}\sum_{i=1}^n X_i\right| \leq 60\mathrm{e}^2 B \sqrt{\frac{\kappa \log(1/\delta)}{n}} + 1200\mathrm{e}B \frac{\kappa \log(n) \log^2(1/\delta)}{n}$$

*Proof.* First we notice that $Z_i = (W_i - \mathbb{E}[W_i]) \cdot X_i$ forms a martingale-difference sequence: $\mathbb{E}[Z_i | \mathcal{F}_{i-1}] = 0$. Therefore, we can apply Theorem 6

$$\mathbb{E}^{1/p}\left[\left|\sum_{i=1}^n Z_i\right|^p\right] \leq 60\mathrm{e}\sqrt{p} \cdot \mathbb{E}^{1/p}\left[\left|\sum_{i=1}^n \mathcal{V}_i\right|^{p/2}\right] + 120p \cdot \mathbb{E}^{1/p}\left[\max_{i \in [n]} |Z_i|^p\right],$$

where $\mathcal{V}_i = \mathbb{E}[Z_i^2 | \mathcal{F}_{i-1}] = X_i^2 \mathrm{Var}(W_i)$. We can easily upper bound the variance of Beta-distributed random variable and obtain

$$\mathbb{E}^{1/p}\left[\left|\sum_{i=1}^n \mathcal{V}_i\right|^{p/2}\right] \leq \mathbb{E}^{1/p}\left[\left|\sum_{i=1}^n \frac{\kappa X_i^2}{n^2}\right|^{p/2}\right] \leq \sqrt{\frac{\kappa B^2}{n}}.$$

For the second term we apply Lemma 11 since $W_i$ are $(\kappa/n^2, 2\kappa/(3n))$-sub-exponential

$$\mathbb{E}^{1/p}\left[\max_{i\in[n]}|Z_i|^p\right] \leq B\left(\max\left\{\sqrt{\frac{8\kappa\log n}{n^2}}, \frac{16\kappa\log n}{3n}\right\} + \mathrm{e}^{1/p}\sqrt{\frac{\kappa}{n^2}}\sqrt{p} + (2\mathrm{e})^{1/p}\frac{16\kappa}{3n}\cdot p\right)$$

$$\leq 20B\kappa\cdot p\cdot\frac{\log n}{n}.$$

Therefore we have

$$\mathbb{E}^{1/p}\left[\left|\sum_{i=1}^n Z_i\right|^p\right] \leq 60\mathrm{e}\cdot p^{1/2}\sqrt{\frac{\kappa B^2}{n}} + 1200\cdot p^2\frac{B\kappa\cdot\log n}{n}.$$

Next we turn from moments to tails. By Markov inequality with $p = \log(1/\delta)$

$$\mathbb{P}\left[\left|\sum_{i=1}^n Z_i\right|\geq t\right] \leq \left(\frac{\mathbb{E}^{1/p}\left[|\sum_{i=1}^n Z_i|^p\right]}{t}\right)^p$$

$$\leq \left(\frac{60\mathrm{e}B\sqrt{\kappa\log(1/\delta)/n} + 1200\log^2(1/\delta)B\kappa\log(n)/n}{t}\right)^{\log(1/\delta)}.$$

Taking $t = 60\mathrm{e}^2 B\sqrt{\frac{\kappa\log(1/\delta)}{n}} + 1200\mathrm{e}B\frac{\kappa\log(n)\log^2(1/\delta)}{n}$ we conclude the statement. $\qquad\square$

## H   Technical Lemmas

**Lemma 12.** *Let $\nu \in \mathcal{P}([0,b])$ be a probability measure over the segment $[0,b]$ and let $\bar{\nu} = (1 - \alpha)\delta_{b_0} + \alpha\cdot\nu$ be a mixture between $\nu$ and a Dirac measure on $b_0 > b$. Then for any $\mu \in (0,b)$*

$$\mathcal{K}_{inf}(\bar{\nu},\mu) \leq (1-\alpha)\,\mathcal{K}_{inf}(\nu,\mu).$$

*Proof.* By a variational formula for $\mathcal{K}_{\mathrm{inf}}$ (see Lemma 9)

$$\mathcal{K}_{\mathrm{inf}}(\bar{\nu},\mu) = \max_{\lambda\in[0,1/(b_0-\mu)]}\mathbb{E}_{X\sim\bar{\nu}}[\log(1 - \lambda(X - \mu))].$$

Since $\bar{\nu}$ is a mixture, we have for any $\lambda \in [0, 1/(b_0 - \mu)]$

$$\mathbb{E}_{X\sim\bar{\nu}}[\log(1 - \lambda(X - \mu))] = (1-\alpha)\mathbb{E}_{X\sim\bar{\nu}}[\log(1 - \lambda(X - \mu))] + \alpha\log(1 - \lambda(b_0 - \mu)).$$

Notice that $\max_{\lambda>0}\log(1 - \lambda(b_0 - \mu)) = 0$. Thus, maximizing each term separately over $\lambda$, we have

$$\mathcal{K}_{\mathrm{inf}}(\bar{\nu},\mu) \leq (1-\alpha)\max_{\lambda\in[0,1/(b_0-\mu)]}\mathbb{E}_{X\sim\bar{\nu}}[\log(1 - \lambda(X - \mu))]$$

$$\leq (1-\alpha)\max_{\lambda\in[0,1/(b-\mu)]}\mathbb{E}_{X\sim\bar{\nu}}[\log(1 - \lambda(X - \mu))] = (1-\alpha)\,\mathcal{K}_{\mathrm{inf}}(\nu,\mu).$$

$\qquad\square$

## I   Experimental details

In this section we detail the experiments we conducted for tabular and non-tabular environments. For all experiments we used 2 CPUs (Intel Xeon CPU 2.20GHz), and no GPU was used. Each experiment took approximately one hour.

### I.1   Tabular experiments

In our initial experiment, we investigated a simple grid-world environment.

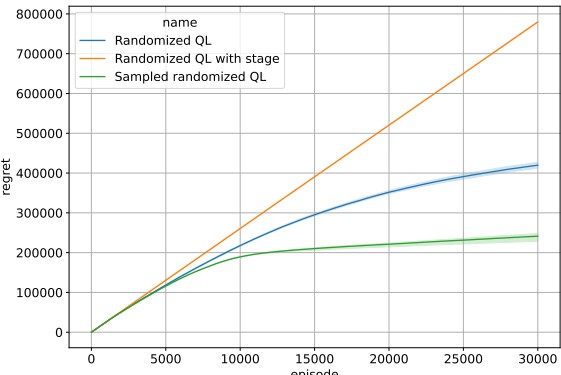

Figure 2: Regret curves of `RandQL`, `Staged-RandQL` and `Sampled-RandQL` on a grid-world environment with $100$ states and $4$ actions for $H = 50$ an transitions noise $0.2$. We show average over $4$ seeds.

**Environments**    For tabular experiments we use two environments.

The first one is a grid-world environment with $100$ states $(i, j) \in [10] \times [10]$ and $4$ actions (left, right, up and down). The horizon is set to $H = 50$. When taking an action, the agent moves in the corresponding direction with probability $1 - \varepsilon$, and moves to a neighbor state at random with probability $\varepsilon = 0.2$. The agent starts at position $(1, 1)$. The reward equals to $1$ at the state $(10, 10)$ and is zero elsewhere.

The second one is a chain environment described by Osband et al. [2016] with $L = 15$ states and $2$ actions (left or right). The horizon is equal to $30$, the probability of moving into wrong direction is equal to $0.1$. The agent starts in the leftmost state with reward $0.05$, also the largest reward is equal to $1$ is the rightmost state.

**Variations of randomized Q-learning**    First we compare the different variations of randomized Q-learning on grid-world environment. Precisely we consider:

- `RandQL` a randomized version of `OptQL`, detailed in Appendix B.
- `Staged-RandQL` a staged version of `RandQL`, described in Section 3.2.
- `Sampled-RandQL` a version of `RandQL` which samples one Q-value function in the ensemble to act, described in Appendix B.

For these algorithms we used the same parameters: posterior inflation $\kappa = 1.0$, $n_0 = 1/S$ prior sample (same as PSRL, see below), ensemble size $J = 10$. We use a similar ensemble size as the one used for the experiments with `OPSRL` by Tiapkin et al. [2022a]. For `Staged-RandQL` we use stage of sizes $\left((1 + 1/H)^k\right)_{k \geq 1}$ without the $H$ factor, in order to have several epochs per state-action pair even for few episodes.

The comparison is presented in Figure 2. We observe that `RandQL` and `Sampled-RandQL` behave similarly with slightly better performance for `Sampled-RandQL`. This is coherent with the experiment on the comparison between `OPSRL` and PSRL Tiapkin et al. [2022a] where the optimistic version performs worst than the fully randomized algorithm. We also note that even with the aggressive stage schedule, `Staged-RandQL` needs more episode to converge. We conclude that despite that stage simplifies the analysis, it artificially slows down the learning in practice.

To ease the comparison with the baselines, for the rest of the experiments we only use `RandQL` because of its similarity with `OptQL`.

**Baselines**    We compare `RandQL` algorithm to the following baselines:

- `OptQL` [Jin et al., 2018] a model-free optimistic Q-learning.

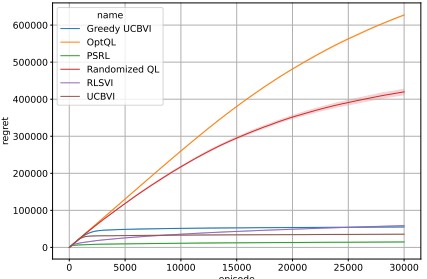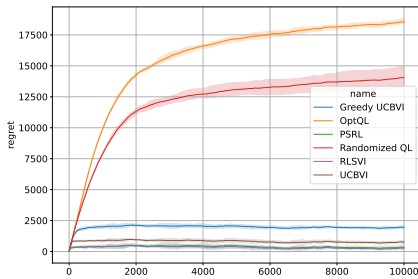

Figure 3: Regret curves of `RandQL` and several baselines in (left) a grid-world environment with 100 states and 4 actions for $H = 50$ an transitions noise 0.2, and (right) in a chain environment of length $L = 15$, 2 actions for $H = 30$ with transition noise 0.1: smaller is better. We show average and error bars over 4 seeds.

- `UCBVI` [Azar et al., 2017] a model-based optimistic dynamic programming.
- `Greedy-UCBVI` [Efroni et al., 2019] optimistic real-time dynamic programming.
- `PSRL` [Osband et al., 2013] model-based posterior sampling.
- `RLSVI` [Russo, 2019] model-based randomized dynamic programming.

The selection of parameters can have a significant impact on the empirical regrets of an algorithm. For example, adjusting the multiplicative constants in the bonus of `UCBVI` or the scale of the noise in `RLSVI` can result in vastly different regrets. To ensure a fair comparison between algorithms, we have made the following parameter choices:

- For bonus-based algorithm, `UCBVI`, `OptQL` we use simplified bonuses from an idealized Hoeffding inequality of the form

$$\beta_h^t(s,a) \triangleq \min\left(\sqrt{\frac{1}{n_h^t(s,a)}} + \frac{H-h+1}{n_h^t(s,a)}, H-h+1\right). \tag{25}$$

  As explained by Ménard et al. [2021], this bonus does not necessarily result in a true upper-confidence bound on the optimal Q-value. However, it is a valid upper-confidence bound for $n_h^t(s,a) = 0$ which is important in order to discover new state-action pairs.
- For `RLSVI` we use the variance of Gaussian noise equal to simplified Hoeffding bonuses described above in (25).
- For `PSRL`, we use a Dirichlet prior on the transition probability distribution with parameter $(1/S, \ldots, 1/S)$ and for the rewards a Beta prior with parameter $(1, 1)$. Note that since the reward $r$ is not necessarily in $\{0, 1\}$ we just sample a new randomized reward $r' \sim \text{Ber}(r)$ accordingly to a Bernoulli distribution of parameter $r$, to update the posterior, see Agrawal and Goyal [2013].

**Results**   Figure 3 shows the result of the experiments. Overall, we see that `RandQL` outperforms `OptQL` algorithm on tabular environment, but still degrades in comparison to model-based approaches, that is usual for model-free algorithms in tabular environments. Indeed, as explained by Ménard et al. [2021], using a model and backward induction allows new information to be more quickly propagated. For example `UCBVI` needs only one episode to propagate information about the last step $h = H$ to the first step $h = 1$ whereas `OptQL` or `RandQL` need at least $H$ episodes. But as counterpart, `RandQL` has a better time-complexity and space- complexity than model based algorithm, see Table 2.

### I.2   Non-tabular experiments

The second experiment was performed on a set of two dimensional continuous environments [Domingues et al., 2021a] with levels of increasing exploration difficulty.

| Algorithm | Time-complexity (per episode) | Space complexity |
|---|---|---|
| UCBVI [Azar et al., 2017] | | |
| PSRL [Osband et al., 2013] | $\widetilde{\mathcal{O}}(HS^2A)$ | $\widetilde{\mathcal{O}}(HS^2A)$ |
| RLSVI [Russo, 2019] | | |
| Greedy-UCBVI [Efroni et al., 2019] | $\widetilde{\mathcal{O}}(HSA)$ | |
| OptQL [Jin et al., 2018] | $\widetilde{\mathcal{O}}(H)$ | $\widetilde{\mathcal{O}}(HSA)$ |
| RandQL (this paper) | $\widetilde{\mathcal{O}}(H)$ | $\widetilde{\mathcal{O}}(HSA)$ |

Table 2: Time- and space- complexity of several tabular algorithms.

**Environment** We use a ball environment with the 2-dimensional unit Euclidean ball as state-space $\mathcal{S} = \{s \in \mathbb{R}^2, \|s\|_2 \leq 1\}$ and of horizon $H = 30$. The action space is a list of 2-dimensional vectors $\mathcal{A} = \{[0.0, 0.0], [-0.05, 0.0], [0.05, 0.0], [0.0, 0.05], [0.0, -0.05]\}$ that can be associated with the action of staying at the same place, moving left, right, up or down. Given a state $s_h$ and an action $a_h$ the next state is

$$s_{h+1} = \text{proj}_{\mathcal{S}}(s_h + a_h + \sigma z_h)$$

where $z_h \sim \mathcal{N}([0,0], I_2)$ is some independent Gaussian noise with zero mean and identity covariance matrix and $\text{proj}_B$ is the euclidean projection on the unit ball $\mathcal{S}$. The initial position $s_1 = \sigma_1 z_1$ with $z_1 \sim \mathcal{N}([0,0], I_2)$ and $\sigma_1 = 0.001$, is sampled at random from a Gaussian distribution. The reward function independent of the action and the step

$$r_h(s, a) = \max(0, 1 - \|s - s'\|/c)$$

where $s' = [0.5, 0.5] \in \mathcal{S}$ is the reward center and $c > 0$ is some smoothness parameter. We distinguish 3 levels by increasing exploration difficulty:

- Level 1, dense reward and small noise. The smoothness parameter is $c = 0.5 \cdot \sqrt{2} \approx 0.71$ and the transition standard deviation is $\sigma = 0.01$.
- Level 2, sparse reward and small noise. The smoothness parameter is $c = 0.2$ and the transition standard deviation is $\sigma = 0.01$.
- Level 3, sparse reward and large noise. The smoothness parameter is $c = 0.2$ and the transition standard deviation is $\sigma = 0.025$.

**RandQL algorithm** Among the different versions of RandQL for continuous state-action space, see Section 4, we pick the Adaptive-RandQL algorithm, described in Appendix F, as it is the closest version to the Adaptive-QL algorithm. It combines the RandQL algorithm and adaptive discretization. For Adaptive-RandQL we used an ensemble of size $J = 10 \approx \log(T)$, $\kappa = 10 \approx \log(T)$ and a prior number of samples of $n_0 = 0.33$. Note that we increased the number of prior samples in comparison to the tabular case as explained in Section 4.

**Baselines** We compare Adaptive-RandQL algorithm to the following baselines:

- Adaptive-QL [Sinclair et al., 2019, 2023], an adaptation of OptQL algorithm to continuous state-space thanks to adaptive discretization;
- Kernel-UCBVI [Domingues et al., 2021c], a kernel-based version of the UCBVI algorithm;
- DQN [Mnih et al., 2013], a deep RL algorithm;
- BootDQN [Osband and Van Roy, 2015], a deep RL algorithm with an additional exploration given by bootstraping several Q-networks;

For Adaptive-QL and Kernel-UCBVI baselines we employ the same simplified bonuses (25) used for the tabular experiments. For Kernel-UCBVI we used Gaussian kernel of bandwidth $0.025$ and the representative states technique, with 300 representative states, described by Domingues et al. [2021c].

For DQN and BootDQN we use as netwrok a 2-layer multilayer perceptron (MLP) with hidden layer size equals to $64$. For exploration, DQN utilizes $\varepsilon$-greedy exploration with coefficient annealing from $1.0$ to $0.1$ during the first $10,000$ steps. For BootDQN we use ensemble of 10 heads and do not use $\varepsilon$-greedy exploration.

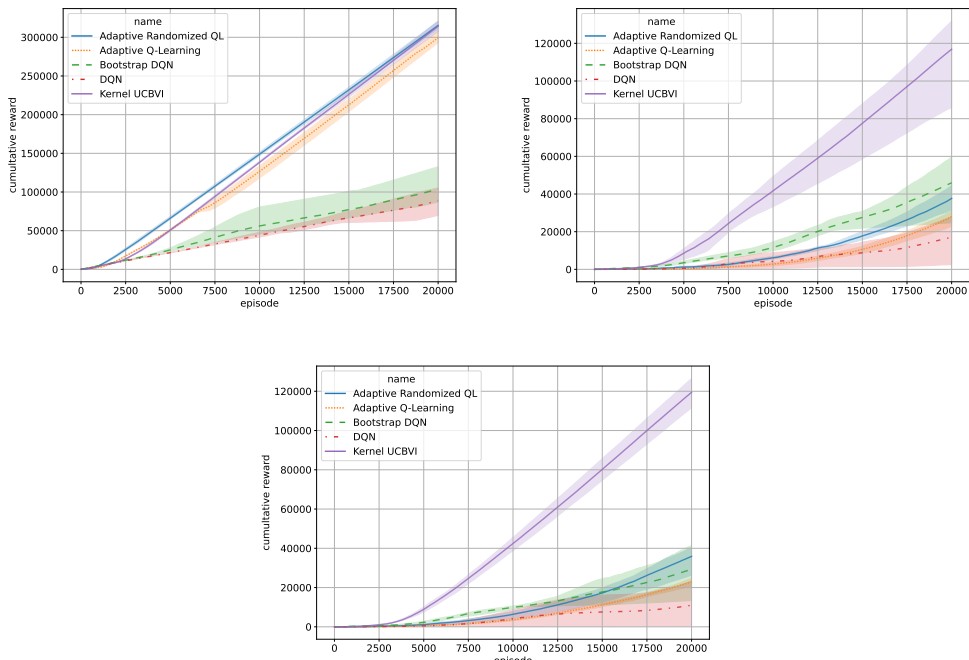

Figure 4: Cumulative rewards (higher is better) of `Adaptive-RandQL` and several baselines in ball environments with increasing exploration difficulty: Upper Left displays Level 1, Upper Right shows Level 2, Down shows Level 3. We show average and error bars over 4 seeds.

| Algorithm | Episode time (second) |
|---|---|
| Adaptive-RandQL | $5.780e^{-02}$ |
| Adaptive-QL | $4.213e^{-02}$ |
| Kernel-UCBVI | $1.523e^{-01}$ |

Table 3: Average time of one episode in second (averaged over 20000 episodes).

**Results** Figure 4 shows the results of non-tabular experiments. Overall, we see that `Adaptive-RandQL` outperforms `Adaptive-QL` in all environments, especially in the sparse reward setting. However, we see that model-based algorithm is much more sample efficient than model-free algorithm, as it was shown by Domingues et al. [2021c]. This is connected to low dimension of the presented environment, where the difference in theoretical regret bounds is not so large. However, this performance come at the price of 3-times larger time complexity, see Table 3.

Regarding the comparison to neural-network based algorithms, we see that approaches based on adaptive discretization always outperforms `DQN` and `BootDQN` on an environment with non-sparse rewards. We connect this phenomenon to the fact that neural network algorithms are solving two problems at the same time: exploration and optimization, whereas discretization-based approaches solve only exploration problem.

In the setup of sparse rewards it turns out that neural network-based approaches are competitive with `Adaptive-QL` and `Adaptive-RandQL`. Notably, `DQN` shows itself as the worst one, whereas `Adaptive-RandQL` and `BootDQN` show similar performance, additionally justifying exploration effect of ensemble learning and randomized exploration.

