# OpenReview forum: "Model-free Posterior Sampling via Learning Rate Randomization"
_NeurIPS.cc/2023/Conference — NeurIPS 2023 poster_

### Official Review · Reviewer_XC7g · 2023-06-14

**Soundness:** 3 good
**Presentation:** 3 good
**Contribution:** 4 excellent
**Rating:** 6
**Confidence:** 3

**Summary:**

The paper introduces Randomized Q-learning (RandQL), a novel model-free algorithm for regret minimization in episodic MDPs. Leveraging a new idea of learning rate randomization, two respect algorithms are proposed for tabular and non-tabular metric space settings, achieving near-optimal regret upper bound. Proof-of-concept experiments are conducted.

**Strengths:**

- The concept of learning rate randomization is very interesting, which enables the algorithm to explore without bonuses.

- Near-optimal regret upper bounds are obtained, showing the efficiency of proposed algorithms.

- Empirical results show that the algorithm outperforms the existing model-free approach (OptQL) and support their theoretical results.

**Weaknesses:**

- The experiments are too simple. Although the theoretical contribution of this paper has outweighed the limited empirical verification and made this paper relatively solid, conducting additional experiments on more complex environments would be advantageous and could further enhance the overall research.

- I was unable to find an explicit explanation of the intuition of the value of $J$. It would be helpful to understand the implications if $J$ is too small or too large. I noticed that you set $J$ to be of logarithmic order, but a high-level explanation of this choice would be beneficial.

- The policy $Q$ value $\overline{Q}_h(s_h,a_h)$ is computed by taking the maximum of $J$ temporary $Q$-values. Will it incur a large estimation error? Recall that in double Q-learning, one common trick is to take a minimum of two networks in order to reduce the estimation error. Although I am not sure how this issue affects your theoretical analysis, this seems a real problem in practice.

- The Net-Staged-RandQL replies on the exact computation of covering, which may be considered a drawback. However, it is worth noting that many existing works also depend on this computation, suggesting that it could be an acceptable requirement.

**Questions:**

While most questions are shown in the "weaknesses" section, there are a few additional smaller questions:

- In Line 141, shouldn't $\overline{V}_h^{n+1}$ be computed from $\overline{Q}^{n+1}_h$? Is it a typo?

- What is the intuition of setting $r_0=2$ (e.g., in line 210)?

**Limitations:**

No negative societal impact was identified.

---

> ### Author Rebuttal · Authors · 2023-08-08
>
> We are very thankful to Reviewer  XC7g for detailed reading and informative feedback. Additionally, we are very thankful for pointing out the misprints. In the sequel, we address the particular questions raised in the review.
>
> - “Intuition of the value of J”.
>
> This value of J is used in the proof to provide "global" optimism in the sense that $\bar{Q}^t_h(s,a) >= Q^*_h(s,a)$ for any $s,a,h$ given "local" optimism for any particular state-action pair. The learning rate randomization technique gives precisely the required "local" optimism.
>
> This idea mathematically is described in the proof of Proposition 1 in Appendix D.3: by the properties of the Dirichlet distribution, we can obtain "local" optimism with positive probability described in Equation (7). Then we use ensembling to show that this "local" optimism holds for all s,a,h with high probability, ensuring "global" optimism by induction. The chosen value of $J$ is minimal to ensure global optimism given the local one. It could be possible to have global optimism even with $ J=1 $, but it requires a non-trivial extension of the Dirichlet anti-concentration results, which we leave for future work.
>
> - “Taking maximum over an ensemble leads to overestimation”.
>
> As usual in optimistic algorithms, the maximum allows us to balance between optimism for ensuring exploration and over-estimation error. In other words, the agent should overestimate, but not too much.
> Regarding comparison with Double Q-learning, the problem of overestimation due to error accumulation becomes very important in the functional approximation setup since it is uncontrollable and could lead to extensive exploration even in environments where it is not needed. In the tabular and metric setting, controlling and reducing over-estimation error is possible due to the proper choice of learning rates. A similar idea of using ensembles to induce additional overestimation was successfully used in BayesUCBDQN algorithm, see Tiapkin et al. (2022).
>
> Tiapkin, Daniil, et al. "From Dirichlet to Rubin: Optimistic exploration in RL without bonuses." International Conference on Machine Learning. PMLR, 2022.
>
> - “Exact computation of coverings”.
>
> We want to stress that we study Net-Staged-RandQL because it is the simplest algorithm for metric spaces, allowing us to obtain (in general) optimal regret bounds in the metric space setting. Additionally, even for this algorithm, it is possible to use not exact covering but its greedy approximation, see Song and Sun [2019]. The price for this inexactness will not change the asymptotic regret.
>
> Additionally, in Appendix F, we provided the main part of the proof for an adaptive algorithm that does not rely so hard on computation of covering for all space but only for regions of interest.
>
> - “Intuition under setting $r_0 = 2$”.
>
> Instead of 2, one could use any number strictly greater than one at the price of increasing a constant in the definition of a number of prior samples $n_0$. This additional reward gives a large enough prior for the maximum value till enough data is collected. In particular, this reward ensures that even if we collected 0 value many times, the corresponding state-action pair is still considered good enough. This helps to make exploration in the first stages when it is crucial.

---

> > ### Comment · Reviewer_XC7g · 2023-08-17
> >
> > Thanks for the detailed response. It addressed my concerns, and I decide to maintain my score.

---

### Official Review · Reviewer_4Am8 · 2023-07-01

**Soundness:** 4 excellent
**Presentation:** 3 good
**Contribution:** 2 fair
**Rating:** 6
**Confidence:** 4

**Summary:**

This work aims to develop computationally efficient, posterior sampling-style, model-free RL algorithms. They propose a family of algorithms based on the idea of learning rate randomization, which they show are able to provably learn and mimic to some degree the updates of posterior sampling. In the tabular setting, they show that the regret of this approach scales with $O(\sqrt{H^5 SAT})$. They also extend the approach to RL in metric spaces, and show that their algorithm is provably efficient in that setting as well.

**Strengths:**

1. The algorithmic technique (exploration via learning rate randomization) presented here is interesting and to my knowledge novel. A common technique in RL is to add bonuses to the value function estimates to induce exploration via optimism. A related approach inspired from posterior sampling is to randomize the rewards in training, which has been shown to induce exploration and is provably efficient. Rather than adding bonuses or randomizing rewards, this paper instead induces exploration by updating the $Q$-function at each step with a random learning rate with carefully chosen distribution. They show that the resulting $Q$ function has a form very similar to that which would be generated by posterior sampling, and that that the algorithm is provably efficient.
2. To my knowledge there are no existing works which propose provably efficient posterior sampling-inspired algorithms for RL in metric spaces, so the results in this setting are the first of their kind.
3. The technique of learning rate randomization proposed here seems like it could be implemented in a straightforward way in deep RL algorithms, and may therefore have practical implications (though this is not explored in this work---some experiments are given, but these are restricted to gridworld environments).
4. Experimental results demonstrate that on gridworld MDPs the approach proposed here performs better than OptQL (but worse than model-based algorithms).

**Weaknesses:**

1. This work is motivated by noting that “a provably tractable model-free posterior sampling algorithm has remained a challenge”. However, I’m not entirely sure this is correct. The RLSVI algorithm of (Russo, 2019) is inspired by posterior sampling, is computationally efficient, and is model free (in that it never requires computing or storing an estimate of the transition dynamics). While I do think the algorithmic techniques are interesting and the extension to the metric RL setting is novel, it’s not clear to me what the contributions of this paper are beyond that.
2. The guarantee obtained in the metric RL setting does not scale with the zooming dimension, but instead scales with the ambient dimension. While this is optimal in the worst-case, in general one would hope to see a scaling in the zooming dimension for learning in metric spaces (as is shown, for example, in (Sinclair et al., 2022)).
3. Furthermore, in the metric RL setting the proposed approach is not in general computationally efficient (as the computational complexity scales with $N_\epsilon$, which in general could be exponentially large in the dimension). In contrast, the metric RL algorithm of (Sinclair et al., 2022) is computationally efficient. Both this and the previous issue arise because the algorithm applies a fixed rather than adaptive discretization technique.
4. The assumptions required in the metric RL setting (in particular Assumptions 2 and 3) are somewhat stronger than what has been required in previous work on metric RL (Sinclair et al., 2022). I do not believe the extra assumptions are that restrictive, but some more discussion on them would be helpful.
5. It would greatly strengthen this paper if experiments could be provided using learning rate randomization on deep RL benchmarks, and illustrating that this approach is able to effectively induce exploration in such settings.
6. It would also be helpful to include the algorithm of (Russo, 2019) in the experimental evaluation of Section 5.

**Questions:**

Please clarify the contributions of this paper and why the algorithm of (Russo, 2019) does not satisfy the desiderate (in the tabular setting).

**Limitations:**

Yes.

---

> ### Author Rebuttal · Authors · 2023-08-08
>
> We are very thankful to Reviewer 4Am8 for detailed reading and informative feedback. In the sequel we answer the questions raised in the review.
>
> - "This work is motivated by noting that “a provably tractable model-free posterior sampling algorithm has remained a challenge”. However, I’m not entirely sure this is correct", "Please clarify the contributions of this paper and why the algorithm of (Russo, 2019) does not satisfy the desiderate (in the tabular setting)."
>
> Regarding this question, we refer to the separate comment submitted to all reviewers.
>
> - “Scaling with a zooming dimension instead of a covering dimension”.
>
> We have discussed the instance-dependent regret bounds scaling with a zooming dimension in the remark after Theorem 2 called “Adaptive discretization” and refer further to Appendix F.
>
> The presented adaptive version of the algorithms is used in the experiments for metric spaces, and we have provided a regret analysis for this algorithm that scales in zooming dimension instead of covering dimension, see Theorem 3. The proof is only sketched since it mostly reuses the same arguments as in Sinclair et al. 2022 given that optimism and upper bound on the over-estimation error are in the same form.
>
> We provide a detailed analysis only for a net-based algorithm because it is much simpler and shows all theoretical insights and differences with tabular settings.
>
> - “Computational efficiency of a net-based algorithm”.
>
> We totally agree and also think that the worst-case algorithm that uses epsilon-nets could be prohibitively expensive from the point of view of both time and space complexity, and it is exactly the reason why we use an adaptive algorithm in the experiments.
>
> - “Assumption 2 and 3 are strong”.
>
> Thank you for your assumption question; we will discuss Assumptions 2 and 3.
>
> About Assumption 2: this assumption holds for any well-behaved transition kernel over metric spaces, as shown in Theorem 1.3.6 by Douc et al. (2018), so it could be taken for free.
>
> Regarding Assumption 3, this assumption is stronger than the one required in the previous work by Sinclair et al. (2022); see Lemma 1 and Lemma 2. However, in compact metric spaces, we are unaware of any MDPs with Lipschitz continuous transition kernel that fails to satisfy Assumption 3.
>
> - It would greatly strengthen this paper if experiments could be provided using learning rate randomization on deep RL benchmarks, and illustrating that this approach is able to effectively induce exploration in such settings.
>
> Providing experiments on deep RL experiments could be a precious contribution. However, due to a lack of time, we leave the extension of our algorithm to the DQN setting for future work.
> Additionally, let us mention that using a random learning rate appeared in deep learning literature with another goal rather than exploration, see Biler et al. 2020.
>
> Blier, Léonard, Pierre Wolinski, and Yann Ollivier. "Learning with random learning rates." Machine Learning and Knowledge Discovery in Databases: European Conference, ECML PKDD 2019, Würzburg, Germany, September 16–20, 2019, Proceedings, Part II. Springer International Publishing, 2020.
>
> - “Empirical comparison with RLSVI”
>
> We added a numerical comparison with the RLSVI algorithm by Russo (2019), see new figures in the general comment.

---

> > ### Comment · Reviewer_4Am8 · 2023-08-14
> > **Reply to rebuttal**
> >
> > I would like to thank the authors for their detailed response to my questions. I believe the majority of my concerns have been addressed and will raise my score accordingly. In particular, I agree with the author's assessment of RLSVI, and that it is not truly model-free in the sense considered in this work. Furthermore, I had previously missed the bounds scaling with the zooming dimension given in Appendix F. I would encourage the authors to include these bounds in the main body for the final version, as space permits (and provide a full proof of the results).

---

### Official Review · Reviewer_xW74 · 2023-07-06

**Soundness:** 3 good
**Presentation:** 3 good
**Contribution:** 2 fair
**Rating:** 5
**Confidence:** 2

**Summary:**

The paper introduces a novel algorithm for regret minimization in episodic Markov Decision Processes. The proposed algorithm, RandQL, achieves optimistic exploration without using bonuses, relying on learning rate randomization. The authors provide theoretical analysis of the algorithm's regret bound and demonstrate its effectiveness in experiments.

**Strengths:**

1. The paper presents a novel algorithm that achieves optimistic exploration with learning rate randomization.
2. Both theoretical analysis of the algorithm and experimental results are provided with discussions to compare with baseline methods.

**Weaknesses:**

1. Lack of baseline comparison: using ensembles of Q values is not new, for example, bootstrapped DQN [1]. It would be an essential baseline method to show the potential difference in learning rate randomization compared with other ensemble methods.
2. The experimental section would benefit from a more complex setting other than the grid world, for example, the chain environment in [1] which requires some “deep” exploration.

[1] Deep Exploration via Bootstrapped DQN. Ian Osband, Charles Blundell, Alexander Pritzel, Benjamin Van Roy

**Questions:**

1. If the learning rate randomization is to approximate the posterior distribution, how is it different from other ensemble methods like bootstrapped DQN[1], which also uses multiple Q values to estimate the posterior distribution?
2. The proposed method does not necessarily need to be restricted in the tabular case, is it possible to extend it to more practical scenarios since its advantage also comes from computation?

---

> ### Author Rebuttal · Authors · 2023-08-08
>
> We are very thankful to Reviewer xW74 for detailed reading and informative feedback. In the sequel we answer the particular questions raised in the review.
>
> - “Lack of baseline comparison: using ensembles of Q values is not new, for example, bootstrapped DQN [1]. It would be an essential baseline method to show the potential difference in learning rate randomization compared with other ensemble methods.”.
>
> The main difference between our approach and the approach of bootstrap DQN lies in the assumptions. We provide theoretical guarantees for our method in the tabular and metric space setups. In contrast, providing guarantees for bootstrap DQN and other neural-network-based algorithms remains an open problem unless strong assumptions are imposed. The goals are also different in our work and the bootstrap DQN paper, where the authors aim to construct an efficient algorithm in the Deep RL setting. In contrast, our primary goal is to provide a theoretically grounded model-free version of the PSRL algorithm.
>
> We agree that ensemble learning is not novel and do not claim it contributes to our work. However, additional sources of randomization are needed to apply this idea since more than the simple effect of different initializations is required, especially if one wants to deal with a stochastic environment.
>
> Additionally, bootstrap DQN trains the Q-head of a neural network with dependent targets, which breaks the analogy with a usual posterior sampling algorithm where one needs to sample completely independent new heads at each episode. Our work validates that the dependencies between targets and different posterior samples do not break the theoretical efficiency.
>
> In our algorithm, we use the ensemble learning idea as a  technique to provide high-probability optimism. In particular, learning rate randomization gives "local" optimism that is a positive probability to sample an upper-bound on V-value for a fixed state-action-step triple—then given "local" optimism, ensembling learning allows us to transmit it to "global" optimism that is indeed used in the proof of regret bound. Mathematically, this idea is described in the proof of Proposition 1 in Appendix D.3. Properties of Dirichlet distribution give "local" optimism in equation (7). However, "global" optimism requires taking a maximum over the ensemble.
>
> Regarding the experimental comparison, we do not provide a comparison with bootstrap DQN because we consider only algorithms with provable guarantees. Still, we compare with PSRL algorithm, the tabular inspiration of the bootstrap DQN.
>
> - “The experimental section would benefit from a more complex setting other than the grid world, for example, the chain environment in [1] which requires some “deep” exploration.”.
>
> We will add experiments for the chain environment. In particular, this experiment also validates the good exploration properties of RandQL compared to OptQL.
>
> - “If the learning rate randomization is to approximate the posterior distribution, how is it different from other ensemble method?”
>
> Theoretical guarantees are the main difference between our and bootstrap DQN methods. Bootstrap DQN highly relies on the properties of different random neural network initializations that give enough randomness to emulate the posterior distribution. However, more than this idea is needed from a theoretical point of view, even in the simple setting of tabular RL. It thus requires additional randomization, for example, the presented learning rate randomization. Additionally, the initial theoretical justification of the bootstrap DQN assumes independence between different Q-networks. This independence does not hold in practice, whereas our algorithm carefully handles this critical issue.
>
> - “The proposed method does not necessarily need to be restricted in the tabular case, is it possible to extend it to more practical scenarios since its advantage also comes from computation?”
>
> It is possible to extend our algorithm beyond the tabular setting by using a random learning rate of SGD during neural network training in the DQN-type algorithm. The variance of this random learning rate should decay over time to ensure that we will not leave a good policy in the end. Additionally, we note that using a random learning rate appeared in deep learning literature with another goal rather than exploration, see Biler et al. 2020. We leave the problem of a DQN extension of RandQL to future work.
>
> Blier, Léonard, Pierre Wolinski, and Yann Ollivier. "Learning with random learning rates." Machine Learning and Knowledge Discovery in Databases: European Conference, ECML PKDD 2019, Würzburg, Germany, September 16–20, 2019, Proceedings, Part II. Springer International Publishing, 2020.

---

> > ### Comment · Reviewer_xW74 · 2023-08-17
> >
> > Thanks for the reply. I still think that comparison with Bootstrapped DQN could be beneficial since 1) PSRL is a model-based algorithm but both Bootstrapped DQN and the proposed method are model-free, and 2) it could be informative to see the difference between two different ensemble-based methods. It is not for comparing with theoretically guaranteed methods, but to see the empirical evidence of how the proposed ensemble method behaves differently compared to Bootstrapped DQN.

---

> > > ### Author Response · Authors · 2023-08-21
> > > **Comparison with Bootstrap DQN**
> > >
> > > We have conducted preliminary experiments with DQN and Bootstrap DQN on a two-dimensional continuous ball environment. With the permission of AC, we attach a link to these results: https://anonymous.4open.science/r/randql_additional_figure-8026/pball_lvl1_randql.pdf.
> > >
> > > In this preliminary experiment the metric space algorithms such as Adaptive RandQL, Adaptive QL and Kernel UCBVI outperforms the neural network based algorithms. The explanations for this phenomenon are simple: while the metric space algorithms are aiming to solve only the exploration problem, the neural-network based algorithms also have to deal with optimization issues that result in slower convergence. Furthermore the metric space algorithms are provided with a good metric whereas the neural-network based algorithms have to learn it. A similar experiment could be found in (Sinclair, 2022), where adaptive metric-space algorithms outperform PPO for small-dimensional problems. We also note that this preliminary experiment is performed on a relatively easy to explore reward function that we call Level 1 in Section I.2 and the situation might be different in harder to explore cases.
> > >
> > > Another takeaway of the experiment is that the difference between DQN and Bootstrap DQN is quite similar to the difference between Adaptive QL and Adaptive RandQL. This shows that the randomization techniques equipped with ensemble learning improves the exploration properties of metric based and neural-network based algorithms in a consistent way.
> > >
> > > We will perform experiments for the metric space environments for more difficult reward functions (Level 2 and Level 3) and add them in the corresponding section as well as an additional discussion.
> > >
> > > Sinclair, S. R., Banerjee, S., & Yu, C. L. (2022). Adaptive discretization in online reinforcement learning. Operations Research.

---

### Official Review · Reviewer_axbs · 2023-07-14

**Soundness:** 4 excellent
**Presentation:** 4 excellent
**Contribution:** 3 good
**Rating:** 6
**Confidence:** 4

**Summary:**

The paper titled Model-free Posterior Sampling via Learning Rate Randomization proposes a novel model-free algorithm called Randomized Q-learning (RandQL) for regret minimization in episodic Markov Decision Processes (MDPs). The algorithm introduces the concept of learning rate randomization to achieve optimistic exploration without using bonuses. The performance of RandQL is analyzed in both tabular and non-tabular metric space settings, and it outperforms existing approaches in baseline exploration environments.

The authors address the challenge of sample complexity in model-free algorithms where the agent has no access to a simulator. They propose RandQL, a tractable model-free algorithm that achieves exploration without using bonuses. RandQL updates an ensemble of Q-values using Q-learning with Beta distributed step-sizes, introducing noise similar to the posterior sampling approach. The ensemble of Q-values can be seen as posterior samples, and RandQL chooses among these samples in an optimistic fashion. The authors analyze the regret bounds of RandQL in both tabular and metric state-action spaces.

The paper highlights that Bayesian-based exploration techniques have shown superior empirical performance compared to bonus-based exploration, but most theoretical studies have focused on model-based algorithms. The authors aim to extend the posterior sampling approach to a provably efficient model-free algorithm that matches the empirical performance of its model-based counterparts. Previous attempts at model-free posterior sampling algorithms have not been computationally tractable, making this a challenge to overcome.

**Strengths:**

1. The idea of relating learning rate randomization to PSRL Dirichlet sampling is interesting.

2. The first computationally tractable model-free (optimistic) posterior sampling algorithm with regret guarantees in tabular setting and metric RL setting.

3. Experiments in finite and continuous MDPs that show that RandQL is competitive with model-based and model-free baselines while keeping a low time-complexity.

**Weaknesses:**

Although the author claims the algorithm do not require bonus, the learning rate randomization still needs (s, a) counting. Counting is not easy to generalise to practical RL with function approximation.

**Questions:**

1. RLSVI related algorithm also do not require bonus but can be can be easily compatible with complex function approximation. Also it utilise all history data in previous theoretical analysis (which can be thought as a definition of model-based), it still only track the optimal value function. Could you give more detailed comparison to this line of work in both computation and sample complexity consideration.

2. See weakness.

I could consider increasing score if the questions are properly answered.

---

> ### Author Rebuttal · Authors · 2023-08-08
>
> We are very thankful to Reviewer axbs for detailed reading and informative feedback. In the sequel we answer the particular questions raised in the review.
>
> - “Although the author claims the algorithm do not require bonus, the learning rate randomization still needs (s, a) counting. Counting is not easy to generalise to practical RL with function approximation”.
>
>
> We agree that we need the counters to define the learning rate with the required properties in the theoretical analysis. However, we would like to insist on two points.
> First, using these counters comes from the learning rate of the vanilla Q-learning algorithm rather than exploration. This choice of learning rates aims to forget past targets with a required speed.
> Second, using counters is only needed to define appropriate parameters of the underlying Beta distribution, and it could be done using other (practical) considerations. For example, one can implement a DQN-type algorithm with a random learning rate whose expectation is the same without this randomization and with variance decaying by some schedule. Note that the same trick cannot work with a direct count-based exploration since it requires counters for exploration.
>
>
>
> - “RLSVI related algorithm also do not require bonus but can be can be easily compatible with complex function approximation. Also it utilise all history data in previous theoretical analysis (which can be thought as a definition of model-based), it still only track the optimal value function. Could you give more detailed comparison to this line of work in both computation and sample complexity consideration.”
>
> The general comparison with the RLSVI algorithm is provided separately. Additionally, our algorithm also tracks only the optimal value function. Still, it uses the randomness from the Bayesian model on the transitions, leading to the Dirichlet process-like posterior on the value rather than the Gaussian posterior in RLSVI. It makes the extension of RandQL to the function approximation setting a promising direction for further research.
> Additionally, a small probability of the appearance of large magnitude gradients could help remove a primacy bias, see Nikishin et al. (2022).
>
> Nikishin, Evgenii, et al. "The primacy bias in deep reinforcement learning." International conference on machine learning. PMLR, 2022.

---

> > ### Comment · Reviewer_axbs · 2023-08-14
> > **Reply to the rebuttals**
> >
> > Thank you for the response. I would keep my score.

---

### Official Review · Reviewer_eHBt · 2023-07-16

**Soundness:** 3 good
**Presentation:** 3 good
**Contribution:** 2 fair
**Rating:** 5
**Confidence:** 3

**Summary:**

This paper introduces a new model-free Q-learning algorithm that utilizes a randomized learning rate. The effectiveness of this algorithm is theoretically validated by demonstrating a promising regret bound of $\widetilde{O}(\sqrt{H^{5} SA T})$ under the tabular setting. Furthermore, a theoretical bound is established for the metric state-action space in this paper.

**Strengths:**


- Introduces a new model-free algorithm with random exploration.
- Provides provable guarantees under both tabular and non-tabular settings.



**Weaknesses:**

The research significance is not well addressed. What are the benefits of eliminating the exploration bonus? What are the promising parts (for future work to develop new methods and apply to practical tasks) compared with alternative methods (e.g., RLSVI, UCB)?

**Questions:**

1. The idea of using multiple ensembles and maximizing to incentivize exploration has been mentioned in previous works. See e.g., [R1, R2] (the reference of [R1] is missing).  The reviewer wants to know the difference and whether this is a general technique that theorists and practitioners should follow in the future.

2. Why is RLSVI viewed as a model-based algorithm in the introduction? What are the computational benefits compared with RLSVI?


[R1] Ishfaq, Haque, et al. "Randomized exploration in reinforcement learning with general value function approximation." *International Conference on Machine Learning*. PMLR, 2021.

[R2] Xiong, Zhihan, et al. "Near-Optimal Randomized Exploration for Tabular Markov Decision Processes." *Advances in Neural Information Processing Systems* 35 (2022): 6358-6371.



**Limitations:**

Limitations:

- The claim in the introduction that "Although empirical evidence suggests that model-based algorithms are more sample efficient than model-free algorithms..." is somewhat misleading. In the field of deep RL, there is no definitive conclusion yet. Please consider revising this argument.

- The vanilla Q-learning algorithm is rarely used in practice. Practitioners typically utilize the target Q-learning algorithm. Please refer to [R3] for further discussion. It is encouraged to investigate this algorithm with a randomized learning rate.

[R3] Zanette, Andrea, and Martin Wainwright. "Stabilizing Q-learning with Linear Architectures for Provable Efficient Learning." *International Conference on Machine Learning*. PMLR, 2022.

---

> ### Author Rebuttal · Authors · 2023-08-08
>
> We are very thankful to Reviewer eHBt for detailed reading and informative feedback. We will add all missing references. In the sequel we answer the questions raised in the review.
>
> - “What are the benefits of eliminating the exploration bonus?  What are the promising parts compared to alternative methods?”
>
> The main benefit of eliminating the exploration bonuses is that it allows for a straightforward generalization to large-scale problems. While implementing bonuses requires maintaining counters for all state-action pairs and using them directly, our algorithm needs counters only in the distribution of the learning rates, which could be done more efficiently. Also, we must emphasize that these counters are used not for exploration but to forget previous targets efficiently, as was done in optimistic Q-learning, see Jin et al. (2018).
>
> Regarding the present work, our main interest is in proposing a simple mechanism that ensures provably efficient exploration.
> As compared to RLVSI, our method provides an alternative way for randomization that automatically adapts to variance (without the need to estimate it); see general comment on RLSVI.
> UCB-type methods usually use highly over-optimistic bonuses (see Osband and Van Roy, 2017), leading to poor empirical performance compared to randomized model-based algorithms such as PSRL or RLSVI. The fairest comparison of RandQL could be done with the OptQL method since both algorithms are model-free, and RandQL outperformed the OptQL algorithm in all numerical experiments presented.
>
>
>
>
> Jin, Chi, et al. "Is Q-learning provably efficient?." Advances in neural information processing systems 31 (2018).
> Osband, Ian, and Benjamin Van Roy. "Why is posterior sampling better than optimism for reinforcement learning?." International conference on machine learning. PMLR, 2017.
>
>
> - “Idea of using multiple ensembles is not novel”.
>
> We would like to stress that the main novel idea presented in our paper is to use learning rate randomization, whereas the ensemble learning idea is a technical tool used to provide high-probability optimism.
>
> Learning rate randomization aims to obtain "local" optimism to obtain Q-value with a positive probability to be optimistic for one particular state-action-step triple. However, this property is not enough to provide a proper regret bound since "global" optimism is needed, that is, optimism with positive probability for all state-action pairs simultaneously.
>
> From the theoretical point of view, using ensembles allows one to move from a "local" optimism for one state-action-step triple with positive probability to a "global" optimism for the initial value with high probability. Mathematically, this idea is described in the proof of Proposition 1 in Appendix D.3. Learning rate randomization gives "local" optimism in Equation 7. However, "global" optimism requires taking a maximum over the ensemble.
> We conjecture that ensembles are not necessary to achieve sublinear regret bounds for this algorithm, and it serves only as a technical instrument. Still, it requires studying the properties of Dirichlet distribution and weighted sums of it more deeply.
>
>
>
> - The claim in the introduction that "Although empirical evidence suggests that model-based algorithms are more sample efficient than model-free algorithms..." is somewhat misleading. In the field of deep RL, there is no definitive conclusion yet. Please consider revising this argument.
>
> In the tabular setting this argument holds as it is demonstrated in our experiments see also Menard et al. (2021). However, we totally agree that there is no consensus yet on the question of sample efficiency beyond the tabular setting, for instance in deep RL setting. We will nuance this argument in the main text and focus on the tabular setting.
>
> Ménard, Pierre, et al. "UCB Momentum Q-learning: Correcting the bias without forgetting." International Conference on Machine Learning. PMLR, 2021.
>
>
> - “The vanilla Q-learning algorithm is rarely used in practice. Practitioners typically utilize the target Q-learning algorithm. Please refer to [R3] for further discussion. It is encouraged to investigate this algorithm with a randomized learning rate.”
>
> We totally agree that the extension of our algorithm to the functional approximation setting in the spirit of [R3] is an interesting direction for future work.
>
> Additionally, the spirit of the target network idea with delayed estimates for the Q-value is already used in the Staged-RandQL algorithm: the Q-value in the update formula after line 194 is delayed. We will discuss this point and a reference [R3].

---

> > ### Comment · Reviewer_eHBt · 2023-08-16
> >
> > Thank you for your response and the detailed explanation. Most of my concerns have been well-addressed. The difference and contribution compared with RLSVI is clear. However, I am not fully convinced that this idea can be easily extended to the practical deep reinforcement learning (RL) setting in future work, despite recognizing its potential based on the tabular experiments. As a result, I have decided to maintain my current stance on the matter.

---

### Official Review · Reviewer_PYAU · 2023-07-27

**Soundness:** 3 good
**Presentation:** 3 good
**Contribution:** 3 good
**Rating:** 6
**Confidence:** 3

**Summary:**

The paper proposes a new algorithm, Randomized Q-learning (RandQL), for regret minimization in episodic Markov Decision Processes (MDPs). RandQL is the first manageable model-free posterior sampling-based algorithm. The authors study RandQL's performance in both tabular and non-tabular metric space settings. In tabular MDPs, RandQL gives a regret bound of $O(\sqrt{H^5SAT})$, where $H$ is the planning horizon, $S$ the number of states, $A$ the number of actions, and $T$ the number of episodes. RandQL has a regret bound of order $O(\sqrt{H^5T^{(d_c+1)/(d_c+2)}})$, where $d_c$ is the covering dimension, in a metric state-action space. Interestingly, RandQL achieves optimistic exploration without bonuses and uses a new idea of learning rate randomization. The authors also show that RandQL does better than current methods on baseline exploration environments in their empirical study.

**Strengths:**

1. The introduction of learning rate randomization is novel and interesting, that lets RandQL achieve optimistic exploration without bonuses. This indeed reduce the memory cost.

2. RandQL does well in both tabular and non-tabular metric space settings. In both cases, the authors show that it gives good regret bounds. That's a pretty solid result.


**Weaknesses:**

In the experiments, the performance gap between model-free algorithms and model-based algorithms seems too large. It would be better to provide stronger evidence in experiments of the benefits of RandQL such as lower computation cost, lower memory cost, or possibly has a closer performance in a more complex environment compared to model-based methods.

**Questions:**

Can authors discuss why the randomized learning rate can work in a high-level idea? For example, is it possible that this randomization brings some optimistic?

---

> ### Author Rebuttal · Authors · 2023-08-08
>
> We are very thankful to Reviewer PYAU for detailed reading and informative feedback. In the following, we answer the concerns raised in the review.
>
> - “In the experiments, the performance gap between model-free algorithms and model-based algorithms seems too large. It would be better to provide stronger evidence in experiments of the benefits of RandQL”.
>
>
> The main advantage of model-free algorithms in general (and RandQL in particular) is their smaller time and space complexities. For example, the space complexity of UCBVI, PSRL and Greedy-UCBVI is S-times larger than the space complexity of the randomized Q-learning. We present this comparison in Table 2 of Appendix I due to space constraints.
>
> - High-level idea of the randomized learning rate.
>
>
> The  learning rate randomization aims to ensure that a (temporary) Q-value estimate is optimistic with positive probability, almost independent of the sampled data, which is crucial for the proper bounding of regrets in the randomized exploration algorithms.

---

> > ### Comment · Reviewer_PYAU · 2023-08-13
> >
> > Thanks for the author's response. The comments have addressed my concerns and I decide to maintain my score.

---

### Author Rebuttal · Authors · 2023-08-08

## Model-based vs model-free
First, we would like to stress that in the tabular setting, according to the Definition 1 by Jin et al. 2018, RLSVI is a model-based algorithm since it needs a space  complexity of order $O(S^2AH)$ independently of $T$, large enough to store an estimate of the MDP (empirical mean reward and empirical transition probabilities). In more general settings, RLSVI algorithms could be implemented in a “model-free fashion” by recording all the observed transitions. However, this implementation approach comes with a trade-off: it requires storing all $T$ transitions and solving a regression problem with sample size $O(T)$. Moreover, one can also argue that these stored transitions form a non-parametric model of the environment, classifying RLSVI as a model-based algorithm.

Similarly, the LSVI-PHE algorithm by Ishfaq et al. (2021) has a memory complexity of order $O(T)$ (to perform linear regressions with different targets at episode $k$, they need to keep in memory all previous transitions) even when the state space is small. In particular, it makes this algorithm model-based in a tabular setting in terms of the required memory complexity if $T$ is large enough (see Definition 1 by Jin et al. 2018). In contrast, the computational complexity of our algorithm doesn’t depend on $T$. To the best of our knowledge, there exists no RLSVI type algorithm avoiding this issue.

## Differences between RLSVI and RandQL

The primary ideological distinction between RLSVI-type algorithms and RandQL lies in the Bayesian model they adopt. RLSVI seeks to construct a Gaussian posterior for the Q-value, whereas RandQL, as a PSRL-based algorithm, approximates the Dirichlet posterior on the transitions in a model free way. Specifically, the Dirichlet Bayesian model on transition probabilities induces a Dirichlet process posterior for the $Q$ value, see Remark 1 in Appendix E.4. This leads to a significantly more expressive distribution family than the Gaussian model employed by RLSVI.

In particular, this expressive representation leads to one interesting feature of RandQL that is not present in RLSVI: the variance adaptivity. In particular, if we consider the expression in Equation 1, the direct computation of the variance of this estimate with respect to random Dirichlet weights yields the empirical variance of the V-value up to a bias that comes from the prior.

To illustrate the importance of this point, it is worth noting that Xiong et al. (2022) needed to adjust by hand the variance of the Gaussian noise based on the variance of a $V$-value estimate like for the case  of Bernstein-type  bonuses. In contrast, the noise generated by our algorithm automatically adjusts to the variance of the V-values. A potential improvement in the regret bound resulting from this variance adaptivity  is a promising research direction.

As  to the differences between sample complexity and computational costs, the regret bound of RandQL algorithm is of order $\widetilde{O}(\sqrt{H^5 SAT})$ whereas a basic RLSVI algorithm, without a single seed randomization technique and variance adjustment (Xiong, 2022),  has regret of order $\widetilde{O}(\sqrt{H^4 S^2 A T})$ (Agrawal et al, 2021).

From the point of view of computational complexity, RandQL has $\widetilde{O}(H)$ per-episode time complexity and $\widetilde{O}(HSA)$ space complexity that outperforms $O(HS^2A)$ time complexity and $O(HS^2A)$ space complexity of  RLSVI-type algorithms.

## Additional figures

We attach a PDF with all additional figures, that includes
- Additional experiments on Chain environment  by Osband et al. (2016) with $L = 15$ states and 2 actions (left or right). The horizon is equal to $30$, the probability of moving into wrong direction is equal to 0.1. The agent starts in the leftmost state with reward 0.05, also the largest reward is equal to 1 is the rightmost state.
- Also we add comparison with RLSVI algorithm in tabular experiment with a grid-world environment.

### References

Russo, Daniel. "Worst-case regret bounds for exploration via randomized value functions." Advances in Neural Information Processing Systems 32 (2019).

Xiong, Zhihan, et al. "Near-Optimal Randomized Exploration for Tabular Markov Decision Processes." Advances in Neural Information Processing Systems 35 (2022): 6358-6371.

Jin, Chi, et al. "Is Q-learning provably efficient?." Advances in neural information processing systems 31 (2018).

Ishfaq, Haque, et al. "Randomized exploration in reinforcement learning with general value function approximation." International Conference on Machine Learning. PMLR, 2021.

Agrawal, Priyank, Jinglin Chen, and Nan Jiang. "Improved worst-case regret bounds for randomized least-squares value iteration." Proceedings of the AAAI Conference on Artificial Intelligence. Vol. 35. No. 8. 2021.

Osband, Ian, et al. "Deep exploration via bootstrapped DQN." Advances in neural information processing systems 29 (2016).

---

### Decision · Program_Chairs · 2023-09-21

**Decision:**

Accept (poster)

**Comment:**

The paper introduces an intriguing randomization strategy for the learning rate, aimed at enhancing the efficiency of model-free posterior sampling in Reinforcement Learning (RL). Both theoretical and experimental evidence are provided. All reviewers have expressed a favorable view of the paper, and the authors have satisfactorily addressed questions raised during the rebuttal period.